# Fast Convergence of Natural Gradient Descent for Over-parameterized Physics-Informed Neural Networks

**Xianliang Xu[4], Wang Kong[2], Jiaheng Mao[1], Zhongyi Huang[4] & Ye Li[1,3]** [*]

[1]College of Computer Science and Technology, Nanjing University of Aeronautics and Astronautics
[2]School of Mathematics, Nanjing University of Aeronautics and Astronautics
[3]MIIT Key Laboratory of Pattern Analysis and Machine Intelligence, Nanjing
[4]Department of Mathematical Sciences, Tsinghua University
`xuxl19@mails.tsinghua.edu.cn`, `zhongyih@tsinghua.edu.cn`,
`{wkong,1901620583,yeli20}@nuaa.edu.cn`

## Abstract

In the context of over-parameterization, there is a line of work demonstrating that randomly initialized (stochastic) gradient descent (GD) converges to a globally optimal solution at a linear convergence rate for the quadratic loss function. However, the convergence rate of GD for training two-layer neural networks exhibits poor dependence on the sample size and the Gram matrix, leading to a slow training process. In this paper, we show that for training two-layer ReLU$^3$ Physics-Informed Neural Networks (PINNs), the learning rate can be improved from the smallest eigenvalue of the limiting Gram matrix to the reciprocal of the largest eigenvalue, implying that GD actually enjoys a faster convergence rate. Despite such improvements, the convergence rate is still tied to the least eigenvalue of the Gram matrix, leading to slow convergence. We then develop the positive definiteness of Gram matrices with general smooth activation functions and provide the convergence analysis of natural gradient descent (NGD) in training two-layer PINNs, demonstrating that the maximal learning rate can be $\mathcal{O}(1)$ and at this rate, the convergence rate is independent of the Gram matrix. In particular, for smooth activation functions, the convergence rate of NGD is quadratic. Numerical experiments are conducted to verify our theoretical results.

## 1 Introduction

In recent years, neural networks have achieved remarkable breakthroughs in the fields of image recognition He et al. (2016), natural language processing Devlin et al. (2018), reinforcement learning Silver et al. (2016), and so on. Moreover, due to the flexibility and scalability of neural networks, researchers are paying much attention in exploring new methods involving neural networks for handling problems in scientific computing, especially in the area of solving partial differential equations (PDEs) Müller & Zeinhofer (2023); Raissi et al. (2019); Yu et al. (2018); Zang et al. (2020); Siegel et al. (2023). Among them, the most representative approach is Physics-Informed Neural Networks (PINNs) Raissi et al. (2019). In the framework of PINNs, one incorporate PDE constraints into the loss function and train the neural network with it. With the use of automatic differentiation, the neural network can be efficiently trained by first-order or second-order optimizers.

First-order methods, such as gradient descent (GD) and stochastic gradient descent (SGD), are widely used in optimizing neural networks as they only calculate the gradient, making them computationally efficient. In addition to first-order methods, there has been significant interest in utilizing second-order optimization methods to accelerate training. These methods have proven to be applicable not only to regression problems, as demonstrated in Martens & Grosse (2015), but also to problems related to PDEs, as shown in Müller & Zeinhofer (2023); Raissi et al. (2019).

---

[*]Corresponding author: Ye Li <yeli20@nuaa.edu.cn>

As for the convergence aspect of the optimization methods, it has been shown that gradient descent algorithm can even achieve zero training loss under the setting of over-parameterization, which refers to a situation where a model has more parameters than necessary to fit the data Du et al. (2018; 2019); Allen-Zhu et al. (2019a;b); Arora et al. (2019); Li & Liang (2018); Zou et al. (2020); Cao & Gu (2019). These works are based on the idea of neural tangent kernel (NTK)Jacot et al. (2018), which shows that training multi-layer fully-connected neural networks via gradient descent is equivalent to performing a certain kernel method as the width of every layer goes to infinity. Despite these attractive convergence results, the learning rate depends on the sample size and the Gram matrix, so it needs to be sufficiently small to guarantee convergence in practice. However, doing so results in a slow training process. In contrast to first-order methods, the second-order method natural gradient descent (NGD) has been shown to enjoy fast convergence for the $L^2$ regression problems as demonstrated in Zhang et al. (2019); Cai et al. (2019), and PINN problems as in Müller & Zeinhofer (2023); Guzmán-Cordero et al. (2025). However, the convergence of NGD in the context of training PINNs is still an open problem. In this paper, we demonstrate that when training PINNs, NGD indeed enjoys a faster convergence rate.

## 1.1 CONTRIBUTIONS

The main contributions of our work are summarized as follows:

- For the PINNs, we simultaneously improve both the learning rate $\eta$ of gradient descent and the requirement for the width $m$. The improvements rely on a new recursion formula for gradient descent. Specifically, our analysis yields a different step-size criterion $\eta = \mathcal{O}(1/\lambda_{\max})$, which empirically permits larger practical learning rates than the $\mathcal{O}(\lambda_0)$ requirement in Gao et al. (2023), see Remark 3.8. The requirement for the width $m$, i.e. $m = \widetilde{\Omega}\left(\frac{(n_1+n_2)^2}{\lambda_0^4 \delta^3}\right)$, can be improved to $m = \widetilde{\Omega}\left(\frac{1}{\lambda_0^4}(\log(\frac{n_1+n_2}{\delta}))\right)$, where $\widetilde{\Omega}$ indicates that some terms involving $\log(m)$ are omitted.

- We present a framework for demonstrating the positive definiteness of Gram matrices for a variety of commonly used smooth activation functions, including the logistic function, softplus function, hyperbolic tangent function, and others. This conclusion is not only applicable to the PDE we have considered but can also be naturally extended to other forms of PDEs.

- We provide the convergence results for natural gradient descent (NGD) in training over-parameterized two-layer PINNs with ReLU$^3$ activation functions and smooth activation functions. Due to the distinct optimization dynamics of NGD compared to GD, the learning rate can be $\mathcal{O}(1)$. Consequently, the convergence rate is independent of $n$ and $\lambda_0$, leading to faster convergence. Moreover, when the activation function is smooth, NGD can achieve a quadratic convergence rate.

## 1.2 RELATED WORKS

**First-order optimizers.** There are mainly two approaches to studying the optimization of neural networks and understanding why first-order methods can find a global minimum. One approach is to analyze the optimization landscape, as demonstrated in Jin et al. (2017); Ge et al. (2015). It has been shown that gradient descent can find a global minimum in polynomial time if the optimization landscape possesses certain favorable geometric properties. However, some unrealistic assumptions in these works make it challenging to generalize the findings to practical neural networks. Another approach to understand the optimization of neural networks is by analyzing the optimization dynamics of first-order methods. For the two-layer ReLU neural networks, as shown in Du et al. (2018), randomly initialized gradient descent converges to a globally optimal solution at a linear rate, provided that the width $m$ is sufficiently large and no two inputs are parallel. Later, these results were extended to deep fully-connected feedforward neural networks and ResNet with smooth activation functions Du et al. (2019). Results for both shallow and deep neural networks depend on the stability of the Gram matrices throughout the training process, which is crucial for convergence to the global minimum. In addition to regression and classification problems, Gao et al. (2023) demonstrated the convergence of the gradient descent for two-layer PINNs through a similar analysis of optimization dynamics. However, both Du et al. (2018) and Gao et al. (2023) require a sufficiently small learning

rate and a large enough network width to achieve convergence. In this work, we conduct a refined full-batch convergent analysis of the over-parameterized PINN regime for GD and NGD, building upon Gao et al. (2023). There're contemporaneous work analysis concentrate on stochastic Jin & Wu (2025) and non-overparameterized Nießen & Müller (2025) settings.

**Second-order optimizers.** Although second-order methods possess better convergence rate, they are rarely used in training deep neural networks due to the prohibitive computational cost. As a variant of the Gauss-Newton method, natural gradient descent (NGD) is more efficient in practice. Meanwhile, as shown in Zhang et al. (2019) and Cai et al. (2019), NGD also enjoys faster convergence rate for the $L^2$ regression problems compared to gradient descent. Müller & Zeinhofer (2023) proposed energy natural gradient descent for PINNs and deep Ritz method, demonstrating experimentally that this method yields solutions that are more accurate than those obtained through GD, Adam or BFGS. After observing the ill-conditioned loss landscape of PINNs, Rathore et al. (2024) introduced a novel second-order optimizer, NysNewtonCG (NNCG), showing that NNCG can significantly improve the solution returned by Adam+L-BFGS. Moreover, under the assumption that the PŁ$^\star$-condition holds, Rathore et al. (2024) demonstrated that the convergence rate of their algorithm is independent of the condition number, which is similar with our result. However, although the PŁ$^\star$-condition holds for over-parameterized neural networks in the context of regression problems Liu et al. (2022), it remains unclear whether this condition holds for PINNs. De Ryck et al. (2024) showed that operator-preconditioning analysis establishes convergence for linearized PINN problems. In this paper, we provide the convergence analysis for NGD in training two-layer PINNs with ReLU$^3$ activation functions or smooth activation functions, showing that it indeed converges at a faster rate.

## 1.3 NOTATIONS

We denote $[n] = \{1, 2, \cdots, n\}$ for $n \in \mathbb{N}$. Given a set $S$, we denote the uniform distribution on $S$ by $\mathrm{Unif}\{S\}$. We use $I\{E\}$ to denote the indicator function of the event $E$. For two positive functions $f_1(n)$ and $f_2(n)$, we use $f_1(n) = \mathcal{O}(f_2(n))$, $f_2(n) = \Omega(f_1(n))$ or $f_1(n) \lesssim f_2(n)$ to represent $f_1(n) \leq C f_2(n)$, where $C$ is a universal constant $C$. A universal constant means a constant independent of any variables. Throughout the paper, we use boldface to denote vectors. Given $x_1, \cdots, x_d \in \mathbb{R}$, we use $(x_1, \cdots, x_d)$ or $[x_1, \cdots, x_d]$ to denote a row vector with $i$-th component $x_i$ for $i \in [d]$ and then $(x_1, \cdots, x_d)^T \in \mathbb{R}^d$ is a column vector.

## 1.4 ORGANIZATION OF THIS PAPER

In Section 2, we provide the problem setup for training two-layer PINNs. We then present the improved convergence results of gradient descent for PINNs in Section 3. In Section 4, we analyze the convergence of natural gradient descent in training two-layer PINNs with ReLU$^3$ activation functions and smooth activation functions. In Section 5, we conduct experiments to verify the theoretical results. The limitations are briefly discussed in Section 6 and we conclude in Section 7. All the detailed proofs and experiments are provided in the Appendix for readability and brevity.

## 2 PROBLEM SETUP

In this section, we consider the same setup as Gao et al. (2023), focusing on the PDE with the following form.

$$\begin{cases} \dfrac{\partial u}{\partial x_0}(\boldsymbol{x}) - \sum_{i=1}^{d} \dfrac{\partial^2 u}{\partial x_i^2}(\boldsymbol{x}) = f(\boldsymbol{x}), \ \boldsymbol{x} \in (0, T) \times \Omega, \\ u(\boldsymbol{x}) = g(\boldsymbol{x}), \ \boldsymbol{x} \in \{0\} \times \Omega \cup [0, T] \times \partial\Omega, \end{cases} \tag{1}$$

where $\Omega \subset \mathbb{R}^d$ is an open and bounded domain, $\boldsymbol{x} = (x_0, x_1, \cdots, x_d)^T \in \mathbb{R}^{d+1}$ and $x_0 \in [0, T]$ is the time variable. In the following, we assume that $\|\boldsymbol{x}\|_2 \leq 1$ for $\boldsymbol{x} \in [0, T] \times \bar{\Omega}$ and $f, g$ are bounded continuous functions.

Moreover, we consider a two-layer neural network of the following form.

$$\phi(\boldsymbol{x}; \boldsymbol{w}, \boldsymbol{a}) = \frac{1}{\sqrt{m}} \sum_{r=1}^{m} a_r \sigma(\boldsymbol{w}_r^T \tilde{\boldsymbol{x}}), \tag{2}$$

where $\boldsymbol{w} = (\boldsymbol{w}_1^T, \cdots, \boldsymbol{w}_m^T)^T \in \mathbb{R}^{m(d+2)}$, $\boldsymbol{a} = (a_1, \cdots, a_m)^T \in \mathbb{R}^m$ and for $r \in [m]$, $\boldsymbol{w}_r \in \mathbb{R}^{d+2}$ is the weight vector of the first layer, $a_r$ is the output weight and $\sigma(\cdot)$ is the ReLU$^3$ activation function. Here, $\tilde{\boldsymbol{x}} = (\boldsymbol{x}^T, 1)^T \in \mathbb{R}^{d+2}$ is the augmented vector from $\boldsymbol{x}$ and in the following, we write $\boldsymbol{x}$ for $\tilde{\boldsymbol{x}}$ for brevity.

Similar to that for the $L^2$ regression problems, we initialize the first layer vector $\boldsymbol{w}_r(0) \sim \mathcal{N}(\boldsymbol{0}, \boldsymbol{I})$, output weight $a_r \sim \text{Unif}(\{-1, 1\})$ for $r \in [m]$ and fix the output weights. In the framework of PINNs, given training samples $\{\boldsymbol{x}_p\}_{p=1}^{n_1}$ and $\{\boldsymbol{y}_j\}_{j=1}^{n_2}$ that are from interior and boundary respectively, we denote $s_p(\boldsymbol{w})$ and $h_j(\boldsymbol{w})$ by

$$s_p(\boldsymbol{w}) = \frac{1}{\sqrt{n_1}} \left( \frac{\partial \phi}{\partial x_0}(\boldsymbol{x}_p; \boldsymbol{w}) - \sum_{i=1}^d \frac{\partial^2 \phi}{\partial x_i^2}(\boldsymbol{x}_p; \boldsymbol{w}) - f(\boldsymbol{x}_p) \right) \tag{3}$$

and

$$h_j(\boldsymbol{w}) = \frac{1}{\sqrt{n_2}}(\phi(\boldsymbol{y}_j; \boldsymbol{w}) - g(\boldsymbol{y}_j)). \tag{4}$$

Then the empirical loss function can be written as

$$L(\boldsymbol{w}) = \frac{1}{2} \left( \|\boldsymbol{s}(\boldsymbol{w})\|_2^2 + \|\boldsymbol{h}(\boldsymbol{w})\|_2^2 \right), \tag{5}$$

where $\boldsymbol{s}(\boldsymbol{w}) = (s_1(\boldsymbol{w}), \cdots, s_{n_1}(\boldsymbol{w}))^T \in \mathbb{R}^{n_1}$ and $\boldsymbol{h}(\boldsymbol{w}) = (h_1(\boldsymbol{w}), \cdots, h_{n_2}(\boldsymbol{w}))^T \in \mathbb{R}^{n_2}$.

The gradient descent updates the hidden weights by the following formulations:

$$\boldsymbol{w}_r(k+1) = \boldsymbol{w}_r(k) - \eta \frac{\partial L(\boldsymbol{w}(k))}{\partial \boldsymbol{w}_r} \tag{6}$$

for all $r \in [m]$ and $k \in \mathbb{N}$, where $\eta > 0$ is the learning rate. The Gram matrix $\boldsymbol{H}(\boldsymbol{w})$ is defined as $\boldsymbol{H}(\boldsymbol{w}) = \boldsymbol{J}\boldsymbol{J}^T$, where

$$\boldsymbol{J} := \left( \frac{\partial s_1(\boldsymbol{w})}{\partial \boldsymbol{w}}, \cdots, \frac{\partial s_{n_1}(\boldsymbol{w})}{\partial \boldsymbol{w}}, \frac{\partial h_1(\boldsymbol{w})}{\partial \boldsymbol{w}}, \cdots, \frac{\partial h_{n_2}(\boldsymbol{w})}{\partial \boldsymbol{w}} \right)^T. \tag{7}$$

## 3 IMPROVED RESULTS OF GD FOR TWO-LAYER PINNS

To simplify the analysis, we make the following assumptions on the training data.

**Assumption 3.1.** For $p \in [n_1]$ and $j \in [n_2]$, $\|\boldsymbol{x}_p\|_2 \leq \sqrt{2}, \|\boldsymbol{y}_j\|_2 \leq \sqrt{2}$, where all inputs have been augmented.

**Assumption 3.2.** No two samples in $\{\boldsymbol{x}_p\}_{p=1}^{n_1} \cup \{\boldsymbol{y}_j\}_{j=1}^{n_2}$ are parallel. This is guaranteed because augmenting $\boldsymbol{x}$ with $(\boldsymbol{x}, 1)$ ensures all samples are distinct.

Under Assumption 3.2, Lemma 3.3 in Gao et al. (2023) implies that the Gram matrix $\boldsymbol{H}^\infty := \mathbb{E}_{\boldsymbol{w} \sim \mathcal{N}(\boldsymbol{0}, \boldsymbol{I})}[\boldsymbol{H}(\boldsymbol{w})]$ is strictly positive definite and we let $\lambda_0 = \lambda_{min}(\boldsymbol{H}^\infty)$. Similar to the case of the regression problem in Du et al. (2018), $\boldsymbol{H}^\infty$ plays an important role in the optimization process. Specifically, under over-parameterization and random initialization, we have two facts that (1) at initialization $\|\boldsymbol{H}(0) - \boldsymbol{H}^\infty\|_2 = \mathcal{O}(1/\sqrt{m})$ and (2) for any iteration $k \in \mathbb{N}$, $\|\boldsymbol{H}(k) - \boldsymbol{H}(0)\|_2 = \mathcal{O}(1/\sqrt{m})$. The following two lemmas can be used to verify these two facts, which are crucial in the convergence analysis.

**Lemma 3.3.** If $m = \Omega\left( \frac{d^4}{\lambda_0^2} \log\left( \frac{n_1+n_2}{\delta} \right) \right)$, we have that with probability at least $1 - \delta$, $\|\boldsymbol{H}(0) - \boldsymbol{H}^\infty\|_2 \leq \frac{\lambda_0}{4}$ and $\lambda_{min}(\boldsymbol{H}(0)) \geq \frac{3}{4}\lambda_0$.

**Remark 3.4.** Under the premise of deriving the same conclusion as our Lemma 3.3, the Lemma 3.5 in Gao et al. (2023) requires that $m = \tilde{\Omega}\left( \frac{(n_1+n_2)^4}{(n_1 n_2)^2 \lambda_0^2} \left( \log(\frac{1}{\delta}) \right)^7 \right)$, where some terms involving $\log(m)$ are omitted. In contrast, on one hand, our conclusion is independent up to logarithmic factors in $n_1 + n_2$, and on the other hand, our conclusion exhibits a clear dependence on $d$. Moreover, the method in Gao et al. (2023) involves truncating the Gaussian distribution and then applying Hoeffding's inequality, which is quite complicated. In contrast, we utilize the concentration inequality for sub-Weibull random variables, which serves as a simple framework for this class of problems.

**Lemma 3.5.** *Let $R \in (0, 1]$, if $\boldsymbol{w}_1(0), \cdots, \boldsymbol{w}_m(0)$ are i.i.d. generated from $\mathcal{N}(\boldsymbol{0}, \boldsymbol{I})$, then with probability at least $1 - \delta - n_1 e^{-mR}$, the following holds. For any set of weight vectors $\boldsymbol{w}_1, \cdots, \boldsymbol{w}_m \in \mathbb{R}^{d+1}$ that satisfy $\|\boldsymbol{w}_r - \boldsymbol{w}_r(0)\|_2 < R$ for any $r \in [m]$, then*

$$\|\boldsymbol{H}(\boldsymbol{w}) - \boldsymbol{H}(0)\|_F < CM^2 R, \tag{8}$$

*where $M = 2(d+2) \log(2m(d+2)/\delta)$ and $C$ is a universal constant.*

**Remark 3.6.** The Lemma 3.6 in Gao et al. (2023) shows that when $\|\boldsymbol{w}_r - \boldsymbol{w}_r(0)\|_2 \leq R = \tilde{\mathcal{O}}\left(\frac{\lambda_0 \delta}{(n_1 + n_2)(\log m)^3}\right)$ holds for all $r \in [m]$, then $\|\boldsymbol{H}(\boldsymbol{w}) - \boldsymbol{H}(0)\|_2 \leq \frac{\lambda_0}{4}$. In contrast, our Lemma 3.5 only requires $R = \mathcal{O}\left(\frac{\lambda_0}{d^2 (\log(m/\delta))^2}\right)$ to reach same result.

For the $L^2$ regression problem, as shown in Du et al. (2018), the convergence of gradient descent requires that the learning rate $\eta = \mathcal{O}(\lambda_0/n^2)$, where $n$ is the sample size of the regression problem. It is evident that this requirement on the learning rate is difficult to satisfy in practical scenarios, since $\lambda_0$ is unknown and $n^2$ is too large . For PINNs, Gao et al. (2023) follows the methodology of Du et al. (2018), thus inheriting similarly stringent requirements on the learning rate. By investigating a new decomposition method for the residual, we arrive at our main result.

**Theorem 3.7.** *Under Assumption 3.1 and Assumption 3.2, if we set the number of hidden nodes*

$$m = \Omega \left( \frac{d^8}{\lambda_0^4} \log^6 \left( \frac{md}{\delta} \right) \log \left( \frac{n_1 + n_2}{\delta} \right) \right)$$

*and the learning rate $\eta = \mathcal{O}\left(\frac{1}{\|\boldsymbol{H}^\infty\|_2}\right)$, then with probability at least $1 - \delta$ over the random initialization, the gradient descent algorithm satisfies*

$$L(k) \leq \left( 1 - \frac{\eta \lambda_0}{2} \right)^k L(0) \tag{9}$$

*for all $k \in \mathbb{N}$.*

**Remark 3.8.** It may be confusing that Gao et al. (2023) has used the same method in Du et al. (2018), yet it only requires $\eta = \mathcal{O}(\lambda_0)$. Actually, it is because that the loss function of PINN has been normalized. If we let $n_1 = n_2 = n$ and $\widetilde{\boldsymbol{H}}^\infty$ be the Gram matrix induced by unnormalized loss function of PINN, then $\lambda_{min}(\boldsymbol{H}^\infty) = \lambda_{min}(\widetilde{\boldsymbol{H}}^\infty)/n$, leading to the convergence rate similar to that of regression problem. At this point, due to the normalization of loss function, $\|\boldsymbol{H}^\infty\|_2 = \lambda_{max}(\boldsymbol{H}^\infty)$ can be bounded by the trace of $\boldsymbol{H}^\infty$, which is an explicit constant that is independent of the sample size $n_1, n_2$. As $\lambda_{min}$ depends on the sample size, it is expected that our $\eta = \mathcal{O}(1/\lambda_{max})$ is an improvement over $\eta = \mathcal{O}(\lambda_{min})$ in Gao et al. (2023). A practical computation for 1D Poisson equation is $\lambda_{min} = 3.47 \times 10^{-11}$ and $1/\lambda_{\max} = 1/(1.73 \times 10^4) = 5.78 \times 10^{-5}$, suggesting that our analysis indeed improves the learning rate requirements.

# 4 CONVERGENCE OF NGD FOR TWO-LAYER PINNS

Although we have improved the learning rate of gradient descent for PINNs, it may still be necessary to set the learning rates to be sufficiently small. Because, although $\boldsymbol{tr}(\boldsymbol{H}^\infty)$ is an explicit constant, it depends on the form of the PDE. However, the loss function of PINNs has a much worse conditioning due to the appearance of the PDE operator. So the ill-conditioning occurs when we move from regression to PINNs, brings strict restrictions for the learning rate of gradient descent for PINNs. This is a central motivation for second-order and natural gradient methods. Moreover, the convergence rate $1 - \frac{\eta \lambda_0}{2}$ also depends on $\lambda_0$, which depends on the sample size and may be extremely small. Zhang et al. (2019) and Cai et al. (2019) have provided the convergence results for natural gradient descent (NGD) in training over-parameterized two-layer neural networks for $L^2$ regression problems. They showed that the maximal learning rate can be $\mathcal{O}(1)$ and the convergence rate is independent of $\lambda_0$, which result in a faster convergence rate. However, the situation in PINNs is significantly different from regression due to the presence of derivative terms from the partial differential equations, which complicates the analysis. Müller & Zeinhofer (2023) and Guzmán-Cordero et al. (2025) studied the energy natural gradient descent (ENGD) for PINNs with practical Woodbury, momentum,

and randomization techniques, demonstrated highly accurate solutions empirically. A theoretical convergence analysis, however, has not yet been established in these works. We note that the NGD in Zhang et al. (2019) and the ENGD in Müller & Zeinhofer (2023) coincide up to the choice of the Moore–Penrose pseudoinverse or the use of the Woodbury matrix identity; therefore, we do not distinguish between them in this work. In the section, we conduct the convergence analysis of NGD for PINNs and demonstrate that it results in a faster convergence rate for PINNs compared to gradient descent.

In this section, we consider the same setup as described in Section 2. During the training process, we fix the output weight $\boldsymbol{a}$ and update the hidden weights via NGD. The optimization objective is the empirical loss function presented in (5), which is defined as follows:

$$L(\boldsymbol{w}) = \frac{1}{2} \left( \|\boldsymbol{s}(\boldsymbol{w})\|_2^2 + \|\boldsymbol{h}(\boldsymbol{w})\|_2^2 \right), \tag{10}$$

The NGD gives the following update rule:

$$\boldsymbol{w}(k+1) = \boldsymbol{w}(k) - \eta \boldsymbol{J}(k)^T \left( \boldsymbol{J}(k)\boldsymbol{J}(k)^T \right)^{-1} \begin{pmatrix} \boldsymbol{s}(k) \\ \boldsymbol{h}(k) \end{pmatrix}, \tag{11}$$

where

$$\boldsymbol{J}(k) = \left( \boldsymbol{J}_1(k)^T, \cdots, \boldsymbol{J}_{n_1+n_2}(k)^T \right)^T \in \mathbb{R}^{(n_1+n_2) \times m(d+2)}$$

is the Jacobian matrix for the whole dataset and $\eta > 0$ is the learning rate. Specifically, for $p \in [n_1]$,

$$\boldsymbol{J}_p(k) = \left[ \left( \frac{\partial s_p(k)}{\partial \boldsymbol{w}_1} \right)^T, \cdots, \left( \frac{\partial s_p(k)}{\partial \boldsymbol{w}_m} \right)^T \right] \in \mathbb{R}^{1 \times m(d+2)} \tag{12}$$

and for $j \in [n_2]$,

$$\boldsymbol{J}_{n_1+j}(k) = \left[ \left( \frac{\partial h_j(k)}{\partial \boldsymbol{w}_1} \right)^T, \cdots, \left( \frac{\partial h_j(k)}{\partial \boldsymbol{w}_m} \right)^T \right] \in \mathbb{R}^{1 \times m(d+2)}. \tag{13}$$

**Remark 4.1.** We note that Zhang et al. (2019) and Cai et al. (2019) have independently and concurrently established the convergence of NGD in the context of regression problems. The difference lies in the fact that Zhang et al. (2019) focused on ReLU activation functions, whereas Cai et al. (2019) considered smooth activation functions and consistently set the learning rate to 1. Here, following Zhang et al. (2019), we refer to this approach as NGD. In Cai et al. (2019), the authors derived this method based on NTK kernel regression and termed it the Gram-Gauss-Newton (GGN) method. The extension of NGD convergence from regression to PINNs is challenging because of the complexity of the PDE residual loss.

**Remark 4.2.** The classical Gauss-Newton method Bonfanti et al. (2024) is given by $\boldsymbol{w}(k+1) = \boldsymbol{w}(k) - \left( \boldsymbol{J}(k)^T \boldsymbol{J}(k) \right)^{-1} \boldsymbol{J}(k)^T \begin{pmatrix} \boldsymbol{s}(k) \\ \boldsymbol{h}(k) \end{pmatrix}$. Although this formula looks different from the NGD update (11), the two coincide when $\eta = 1$ at the level of the Moore–Penrose pseudoinverse: $\boldsymbol{J}(k)^+ = \left( \boldsymbol{J}(k)^T \boldsymbol{J}(k) \right)^{-1} \boldsymbol{J}(k)^T = \boldsymbol{J}(k)^T \left( \boldsymbol{J}(k)\boldsymbol{J}(k)^T \right)^{-1}$. However, this equivalence is only algebraic. In practice the two updates behave differently because $\boldsymbol{J}(k) \in \mathbb{R}^{(n_1+n_2) \times m(d+2)}$ is highly rectangular and never invertible strictly, and different pseudoinverse representations apply in row-dependent or column-dependent cases. The computational cost are also different, as pointed in Guzmán-Cordero et al. (2025) with Woodbury's Identity. The NGD's formula (11) in this work, originally adopted from Zhang et al. (2019), also coincides with the energy natural gradient descent (ENGD) proposed in Müller & Zeinhofer (2023); Guzmán-Cordero et al. (2025) once the Moore–Penrose inverse or Woodbury identity is applied. A crucial distinction arises in the over-parameterized regime. The Gauss-Newton Gram matrix $\boldsymbol{J}(k)^T \boldsymbol{J}(k) \in \mathbb{R}^{m(d+2) \times m(d+2)}$ becomes extremely high-dimensional and typically singular as $m$ grows, while the NGD $\boldsymbol{J}(k)\boldsymbol{J}(k)^T \in \mathbb{R}^{(n_1+n_2) \times (n_1+n_2)}$ won't. This difference is key for both practical scalability and numerical stability.

For the activation function of the two-layer neural network

$$\phi(\boldsymbol{x}; \boldsymbol{w}, \boldsymbol{a}) = \frac{1}{\sqrt{m}} \sum_{r=1}^{m} a_r \sigma(\boldsymbol{w}_r^T \boldsymbol{x}), \tag{14}$$

we consider settings where $\sigma(\cdot)$ is either the ReLU$^3$ activation function or a smooth activation function satisfying the following assumption.

**Assumption 4.3.** There exists a constant $c > 0$ such that $\sup_{z \in \mathbb{R}} |\sigma^{(3)}(z)| \leq c$ and for any $z, z^{'} \in \mathbb{R}$,

$$|\sigma^{(k)}(z) - \sigma^{(k)}(z^{'})| \leq c|z - z^{'}|, \tag{15}$$

where $k \in \{0, 1, 2, 3\}$. Moreover, $\sigma(\cdot)$ is analytic and is not a polynomial function. We also assume that for any positive integer $n \geq 2$, $\lim_{x \to +\infty} \sigma^{(n)}(x)/\phi(x) = c_n \neq 0$, where the function $\phi(\cdot)$ needs

$$\lim_{x \to +\infty} \phi(x) = 0, \lim_{x \to +\infty} x \cdot \frac{\phi(bx)}{\phi(x)} = 0$$

holding for any constant $b > 1$.

**Lemma 4.4.** *If no two samples in $\{\boldsymbol{x}_p\}_{p=1}^{n_1} \cup \{\boldsymbol{y}_j\}_{j=1}^{n_2}$ are parallel, then the Gram matrix $\boldsymbol{H}^\infty$ is strictly positive definite for activation functions that satisfy Assumption 4.3, i.e., $\lambda_0 := \lambda_{min}(\boldsymbol{H}^\infty) > 0$.*

**Remark 4.5.** Assumption 4.3 holds for various commonly used activation functions, including logistic function $\sigma(z) = 1/(1 + e^{-z})$ (with $\phi(z) = e^{-z}$), softplus function $\sigma(z) = \log(1 + e^z)$ (with $\phi(z) = e^{-z}$), hyperbolic tangent function $\sigma(z) = (e^z - e^{-z})/(e^z + e^{-z})$ (with $\phi(z) = e^{-2z}$), swish function $\sigma(z) = z/(1 + e^{-z})$ (with $\phi(z) = ze^{-z}$) and others.

Unlike the approach for gradient descent, Zhang et al. (2019) focus on the change of the Jacobian matrix for NGD rather than the Gram matrix. Roughly speaking, they show that when $\|\boldsymbol{w} - \boldsymbol{w}(0)\|_2$ is small, then $\|\boldsymbol{J}(\boldsymbol{w}) - \boldsymbol{J}(0)\|_2$ is also proportionally small. However, this approach is not applicable to PINNs, because the loss function involves derivatives. Roughly speaking, the stability considered in Zhang et al. (2019) is more global in nature, whereas ours is local. In fact, the PINN loss includes first- and second-order derivatives of the neural output (see Eq. (3)–(4)), so each Jacobian block $\partial s_p/\partial w_r$ and $\partial h_j/\partial w_r$ contains higher-order derivatives of the activation and of the weights. Consequently, even a small perturbation in weights may cause large variations in the derivatives, violating the Lipschitz-type condition required by Zhang et al. (2019). Since the subsequent conclusions require the boundedness of local weights, we do not use this stability. Moreover, from the Theorem 1 in Zhang et al. (2019), we can see that this stability imposes additional constraints on the learning rate. Therefore, we instead focus on the stability of $\boldsymbol{J}(\boldsymbol{w})$ with respect to each individual weight vector $\boldsymbol{w}_r$ in the following Lemma, which provides a more targeted approach.

**Lemma 4.6.** *Let $R \in (0, 1]$, if $\boldsymbol{w}_1(0), \cdots, \boldsymbol{w}_m(0)$ are i.i.d. generated $\mathcal{N}(\boldsymbol{0}, \boldsymbol{I})$, then with probability at least $1 - P(\delta, m, R)$ the following holds. For any set of weight vectors $\boldsymbol{w}_1, \cdots, \boldsymbol{w}_m \in \mathbb{R}^{d+2}$ that satisfy for any $r \in [m]$, $\|\boldsymbol{w}_r - \boldsymbol{w}_r(0)\|_2 < R$, then*

*(1) when $\sigma(\cdot)$ is the ReLU$^3$ activation function, we have that*

$$\|\boldsymbol{J}(\boldsymbol{w}) - \boldsymbol{J}(0)\|_2 \leq CM\sqrt{R}, \tag{16}$$

*where $C$ is a universal constant, $M = 2(d + 2)\log(2m(d + 2)/\delta)$ and*

$$P(\delta, m, R) = \delta + n_1 e^{-mR}; \tag{17}$$

*(2) when $\sigma(\cdot)$ is the smooth activation function satisfies Assumption 4.3, we have that*

$$\|\boldsymbol{J}(\boldsymbol{w}) - \boldsymbol{J}(0)\|_2 \leq CdR \tag{18}$$

*for $m \geq \log^2(1/\delta)$, where $C$ is a universal constant and $P(\delta, m, R) = \delta$.*

With the stability of Jacobian matrix, we can derive the following convergence results.

**Theorem 4.7.** *Let $L(k) = L(\boldsymbol{w}(k))$, then the following conclusions hold.*

*(1) When $\sigma(\cdot)$ is the ReLU$^3$ activation function, under Assumption 3.2, we set*

$$m = \Omega\left(\frac{1}{(1-\eta)^2}\frac{d^8}{\lambda_0^4}\log^6\left(\frac{md}{\delta}\right)\log\left(\frac{n_1 + n_2}{\delta}\right)\right)$$

*and $\eta \in (0, 1)$, then with probability at least $1 - \delta$ over the random initialization for all $k \in \mathbb{N}$*

$$L(k) \leq (1 - \eta)^k L(0). \tag{19}$$

*(2) When $\sigma(\cdot)$ is the smooth activation function satisfies Assumption 4.3, under Assumption 3.2, we set*

$$m = \Omega\left(\frac{1}{1-\eta}\frac{d^6}{\lambda_0^3}\log^2\left(\frac{md}{\delta}\right)\log\left(\frac{n_1+n_2}{\delta}\right)\right)$$

*and $\eta \in (0,1)$, then with probability at least $1-\delta$ over the random initialization for all $k \in \mathbb{N}$*

$$L(k) \leq (1-\eta)^k L(0). \tag{20}$$

In Theorem 4.7, the requirements of $m$ with ReLU$^3$ and smooth activation functions exhibit different dependencies on $\lambda_0$ and $d$. The discrepancy is primarily due to the distinct formulations presented in (16) and (18) of Lemma 4.5.

**Remark 4.8.** We first compare our results with those of NGD for $L^2$ regression problems. Given that the convergence results are the same, our focus shifts to examining the necessary conditions for the width $m$. As demonstrated in Zhang et al. (2019) and Cai et al. (2019), it is required that $m = \Omega\left(\frac{n^4}{\lambda_0^4\delta^3}\right)$ for ReLU activation function and $m = \Omega\left(\max\left\{\frac{n^4}{\lambda_0^4}, \frac{n^2 d\log(n/\delta)}{\lambda_0^2}\right\}\right)$ for smooth activation function. Clearly, our result has a worse dependence on $d$, which is inevitable due to the involvement of derivatives in the loss function. Moreover, our requirement for $m$ appears to be almost independent of $n$, primarily because our loss function has been normalized. With smooth activation functions, in addition to the dependence on $d$, Theorem 4.7 (2) only requires that $m = \Omega(\lambda_0^{-3})$. However, Cai et al. (2019) demands a more stringent condition, requiring that $m = \Omega(\lambda_0^{-4})$.

Comparing with our results in Section 3, the requirement for $m$ in Theorem 4.7 (1) is the same as in Theorem 3.8, when we make $\eta$ less close to 1. On the other hand, since $\eta = \mathcal{O}(1)$ and the convergence rate only depends on $\eta$, NGD can lead to faster convergence than GD.

Note that as $\eta$ approaches 1, the width $m$ tends to infinity, thus, the convergence results in Theorem 4.7 become vacuous. In fact, when $\eta = 1$, NGD can enjoy a second-order convergence rate even though $m$ is finite, provided that $\sigma(\cdot)$ satisfies Assumption 4.3.

**Corollary 4.9.** *Under Assumption 3.2 and Assumption 4.3, set $\eta = 1$ and*

$$m = \Omega\left(\frac{d^6}{\lambda_0^3}\log^2\left(\frac{md}{\delta}\right)\log\left(\frac{n_1+n_2}{\delta}\right)\right),$$

*then with probability at least $1-\delta$, we have*

$$\left\|\begin{pmatrix}\boldsymbol{s}(t+1)\\\boldsymbol{h}(t+1)\end{pmatrix}\right\|_2 \leq \frac{CB^4}{\sqrt{m\lambda_0^3}}\left\|\begin{pmatrix}\boldsymbol{s}(t)\\\boldsymbol{h}(t)\end{pmatrix}\right\|_2^2$$

*for all $t \in \mathbb{N}$, where $C$ is a universal constant and $B = \sqrt{2(d+2)\log(2m(d+2)/\delta)}+1$. Moreover, we can get a second order convergence result for regression problems with smooth activation functions as follows.*

$$\|\boldsymbol{y}-\boldsymbol{u}(t+1)\|_2 \lesssim \frac{n^{3/2}}{\sqrt{m\lambda_0^3}}\|\boldsymbol{y}-\boldsymbol{u}(t)\|_2^2.$$

Instead of inducing on the convergence rate of the empirical loss function, as shown in Condition 1, we perform induction on the movements of the hidden weights as follows.

**Condition 1.** At the $t$-th iteration, we have $\|\boldsymbol{w}_r(t)\|_2 \leq B$ and

$$\|\boldsymbol{w}_r(t)-\boldsymbol{w}_r(0)\|_2 \leq \frac{CB^2\sqrt{L(0)}}{\sqrt{m}\lambda_0} := R'$$

for all $r \in [m]$, where $C$ is a universal constant and $B = \sqrt{2(d+2)\log\left(\frac{2m(d+2)}{\delta}\right)}+1$.

With Condition 1, we can directly derive the following convergence rate of the empirical loss function.

**Corollary 4.10.** *If Condition 1 holds for $t = 0, \cdots, k$ and $R' \leq R$ and $R'' \lesssim \sqrt{1-\eta}\sqrt{\lambda_0}$, then*

$$L(t) \leq (1-\eta)^t L(0),$$

*holds for $t = 0, \cdots, k$, where $R$ is the constant in Lemma 4.5 and $R'' = CM\sqrt{R}$ is in (16) when $\sigma$ is the ReLU$^3$ activation function, $R'' = CdR$ is in (18) when $\sigma$ satisfies Assumption 4.3.*

## 5 EXPERIMENTAL RESULTS

We conduct a comparative evaluation of the NGD against existing optimizers for PINN training with respect to accuracy, computational efficiency, and alignment with theoretical analysis. The experimental implementations are listed in the Appendix A.

Table 1: Relative $L^2$-error of Different Optimizers.

|  | SGD | Adam | L-BFGS | **NGD** |
|---|---|---|---|---|
| 1D Poisson | 1.28e-01 ± 4.31e-02 | 6.46e-02 ± 1.43e-02 | 2.63e-04 ± 8.95e-05 | **1.67e-05** ± 9.07e-06 |
| 2D Poisson | 1.45e-01 ± 7.34e-02 | 5.32e-03 ± 9.79e-04 | 3.17e-03 ± 8.66e-04 | **1.12e-04** ± 6.99e-05 |
| 1D Heat | 5.43e-01 ± 9.98e-02 | 6.91e-03 ± 1.31e-03 | 4.98e-03 ± 1.83e-03 | **3.42e-04** ± 7.52e-05 |
| 2D Helmholtz | 8.48e+00 ± 6.37e+00 | 1.06e+00 ± 8.11e-01 | 3.35e+00 ± 1.94e+00 | **6.67e-03** ± 1.89e-03 |
| 10D Poisson | 1.35e-02 ± 8.17e-03 | 3.15e-03 ± 8.93e-04 | nan | **9.91e-04** ± 1.47e-04 |

**Comparison to Existing Optimizers.** We report the relative $L^2$-error of the NGD optimizer to the commonly used first order optimizers (the SGD optimizer, the Adam optimizer) and second order optimizer (the L-BFGS optimizer) in Table 1. Here 'nan' means the training loss becomes infinity. We see that NGD performs best on all five equations.

**Learning Rate Study.** We report the behavior of convergence at different learning rates, showing the strong robustness of the NGD method to hyperparameter selection. Table 2 demonstrates that, unlike SGD and Adam which demand small learning rates for convergence, the NGD maintains stable convergence across a wide range of learning rates without notable accuracy deterioration. This characteristic, which markedly outperforms conventional optimization approaches, clearly illustrates the strong robustness of the NGD method to hyperparameter selection.

Table 2: Relative $L^2$-error Comparison Across Different Learning Rates $\eta$.

| learning rate $\eta$ | 1.0 | 0.5 | 0.1 | 0.05 | 0.01 | 0.005 | 0.001 |
|---|---|---|---|---|---|---|---|
| SGD | nan | nan | nan | nan | 1.19e-02 | 6.91e-02 | 7.36e-02 |
| Adam | 1.01e+00 | 1.00e+00 | 1.00e+00 | 1.01e+00 | 1.64e-02 | 3.25e-02 | 1.49e-02 |
| **NGD** | **1.97e-03** | **1.18e-03** | **3.24e-04** | **1.87e-04** | **1.12e-04** | **1.22e-04** | **1.68e-04** |

**Network Width Study.** A comparative analysis of the model performance is performed with progressively increasing network widths. Table 3 demonstrates that increasing network width leads to significant accuracy improvements. This trend validates that wider architectures exhibit enhanced function approximation capabilities.

Table 3: Relative $L^2$-error Comparison Across Different Network Width $m$ for NGD.

| $m$ | 20 | 40 | 80 | 160 | 320 | 640 | 1280 | 2560 |
|---|---|---|---|---|---|---|---|---|
| error | 1.59-03 | 7.21e-04 | 5.18e-04 | 3.8e-04 | 3.08e-04 | 2.76e-04 | 1.78e-04 | 7.05e-05 |

**Fast Convergence Study.** We report the training loss convergence results for different optimizers. We train SGD and Adam for 10000/20000 epochs with learning rate 1e-3, and the NGD for 100/200 epochs with learning rate 0.1. Figure 1 empirically demonstrates that the NGD converges much faster than commonly used SGD and Adam optimizers, which is consistent with our theoretical analysis equation (9) in Theorem 3.7 and equation (20) in Theorem 4.7.

**Empirical Convergence Rates Study.** We continue to report the empirical convergence rates of the NGD in different equations. We compare the empirical training loss curves of the NGD when $\eta = 0.1$ with the theoretical linear rates in our main Theorems 4.7. The theoretical decay follows $L(k) \approx C(1 - \eta)^k$, and the fitted experimental decay is $L(k) \approx \mathcal{O}(k^{-1.55})$ for 1D Poisson equation, $\mathcal{O}(k^{-1.92})$ for 1D Heat equation and $\mathcal{O}(k^{-1.13})$ for 2D Poisson equation in Figure 2. For the heat equation, convergence initially exceeds the predicted rate and later slows markedly. This is consistent

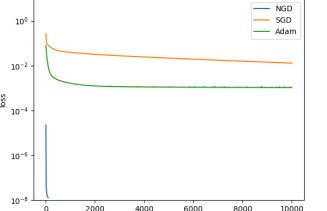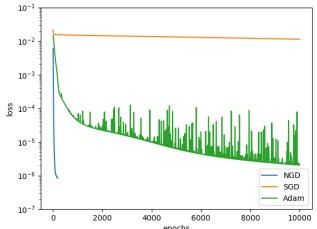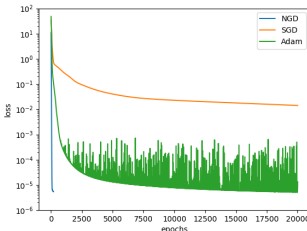

Figure 1: Training Loss Decay Comparison for 1D Poisson (left), 1D Heat (middle) and 2D Poisson (right) Equations.

with known NTK decay and multi-phase behaviors in PINNs. Generally, the empirical loss of NGD roughly follows the predicted linear regime in early iterations, before entering a slower phase usually observed across all optimizers.

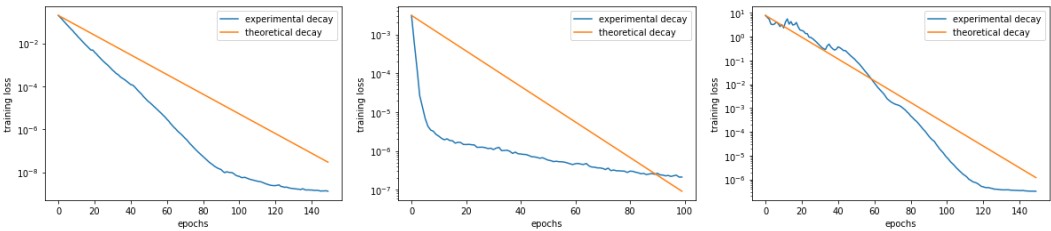

Figure 2: The Experimantal Training Loss Decay and Theoretical Decay (Theorem 4.7) for 1D Poisson (left), 1D Heat (middle) and 2D Poisson (right) Equations.

## 6 LIMITATIONS

The computational cost of NGD is mainly on the $(J \cdot J^\top)^{-1}$ with the Jacobian matrix $J$ is of size $n \times p$, where $n = n_1 + n_2$ is the training data size and $p = m(d + 2)$ is the number of trainable parameters. So NGD will be quite expensive for large amount of training data.As a result, several cost-effective variants have been proposed, such as K-FAC Martens & Grosse (2015); Dangel et al. (2024; 2025), ENGD Müller & Zeinhofer (2023); Guzmán-Cordero et al. (2025) and mini-batch NGD. We only proved the convergence results for the full-batch NGD in this paper, and it would be interesting to investigate the convergence of these methods for PINNs in future works. On the other hand, while the over-parameterized assumption enables the use of NTK stability for proving global convergence, the practical guarantee for arbitrary sampled projected gradient descent without assumption on the network size Nießen & Müller (2025) address a different framework, and the NGD analysis without over-parameterized assumption represent an interesting complementary direction.

## 7 CONCLUSION AND OUTLOOK

In this paper, we have improved the conditions required for the convergence of gradient descent for PINNs, showing that gradient descent actually achieves a better convergence rate. Furthermore, we demonstrate that natural gradient descent can find the global optima of two-layer PINNs with ReLU[3] or smooth activation functions for a class of second-order linear PDEs. Compared to gradient descent, natural gradient descent exhibits a faster convergence rate and its maximal learning rate is $\mathcal{O}(1)$. In conclusion, the NGD offers three key advantages: 1) more relaxed learning rate requirements; 2) faster convergence rates independent of $\lambda_0$; 3) superior empirical performance. Additionally, extending the convergence analysis to deep neural networks, stochastic version of NGD, and studying the generalization error of trained PINNs are important directions for future research.

ETHICS STATEMENT

We acknowledge that all authors have read and commit to adhering to the ICLR Code of Ethics.

REPRODUCIBILITY STATEMENT

To facilitate reproducibility, we have taken the following steps: (1) Source code and configuration files for all key experiments are provided as supplementary material. (2) All theoretical claims are accompanied by full proofs (in the Appendix) and assumptions are clearly stated. (3) All hyperparameters used, neural network architecture details are provided either in the main text or in the Appendix.

ACKNOWLEDGMENTS

Z.Y.H. was supported by NSFC Project No. 12025104, W. Kong was supported by the NSFC Project No. 12201299, Y.L. was supported by NSFC Project No. 62106103 and Fundamental Research Funds for the Central Universities No. ILF240021A25.

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

## A EXPERIMENTAL IMPLEMENTATION

In this section, we provide several examples to demonstrate the superiority of the natural gradient descent(NGD) approach. The configurations used in these examples are listed in Table 4. We report the relative $L^2$-error of the NGD optimizer to the commonly used first order optimizers (the SGD optimizer, the Adam optimizer) and second order optimizer (the L-BFGS optimizer) in Table 1. The relative $L^2$-error is defined as follows:

$$relative \ L^2 \ error = \frac{\sqrt{\sum_{i=1}^{N} |\hat{u}(\mathbf{x_i}) - u_{\text{ref}}(\mathbf{x_i})|^2}}{\sqrt{\sum_{i=1}^{N} |u_{\text{ref}}(\mathbf{x_i})|^2}}, \tag{21}$$

where $\hat{u}$ denotes the predicted solution and $u_{\text{ref}}$ represents the reference solution. To show the generalization ability of NGD, we should note that the testing collocation points $\{\mathbf{x_i}\}_{i=1}^{N}$ are different from the training samples $\{\boldsymbol{x}_p\}_{p=1}^{n_1}$ and $\{\boldsymbol{y}_j\}_{j=1}^{n_2}$.

Table 4: Configurations of Different Equations.

|  | $N_f$ | $N_b$ | batch size | hidden layers | hidden neurons | activation function |
|---|---|---|---|---|---|---|
| 1D Poisson | 500 | 2 | 100 | 1 | 128 | $\tanh(\cdot)$ |
| 2D Poisson | 1,000 | 200 | 100 | 1 | 128 | $\tanh(\cdot)$ |
| 1D Heat | 1,000 | 200 | 100 | 1 | 128 | $\tanh(\cdot)$ |
| 2D Helmholtz | 1,000 | 200 | 100 | 1 | 128 | $\tanh(\cdot)$ |
| 10D Poisson | 10,000 | 1,000 | 100 | 1 | 128 | $\tanh(\cdot)$ |

### A.1 1D POISSON EQUATION

First, we begin with a toy example of the 1D Poisson equation to display the performance of the NGD method. The equation is defined in the domain $\Omega = [0, \pi]$,

$$\begin{cases} -u''(x) = f(x), & x \in \Omega, \\ u(x) = 0, & x \in \partial\Omega. \end{cases} \tag{22}$$

The true solution is set as $u(x) = \sin(x)$, which allows us to derive the corresponding force term $f(x) = \sin(x)$. We randomly sample $N_f = 500$ points in the domain $\Omega$. For the neural network architecture, we employ a single hidden layer model with 128 units and $\tanh(\cdot)$ activation functions across all computations. The NGD optimizer is trained for 100 epochs, while the LBFGS optimizer is run for 1 epoch with a maximum of 500 iterations per epoch. All other optimizers are run for 10,000 epochs for comprehensive comparison. The relative $L^2$-error is $1.67e - 05$ for the NGD optimizer. Figure 3 shows the predicted solution for the 1D Poisson equation alongside the reference solution. The prediction is in excellent agreement with the reference solution, highlighting the superior performance of the NGD method. Figure 4 depicts the loss decay during the training process, we can see that the NGD method achieves a quite small loss at the very beginning.

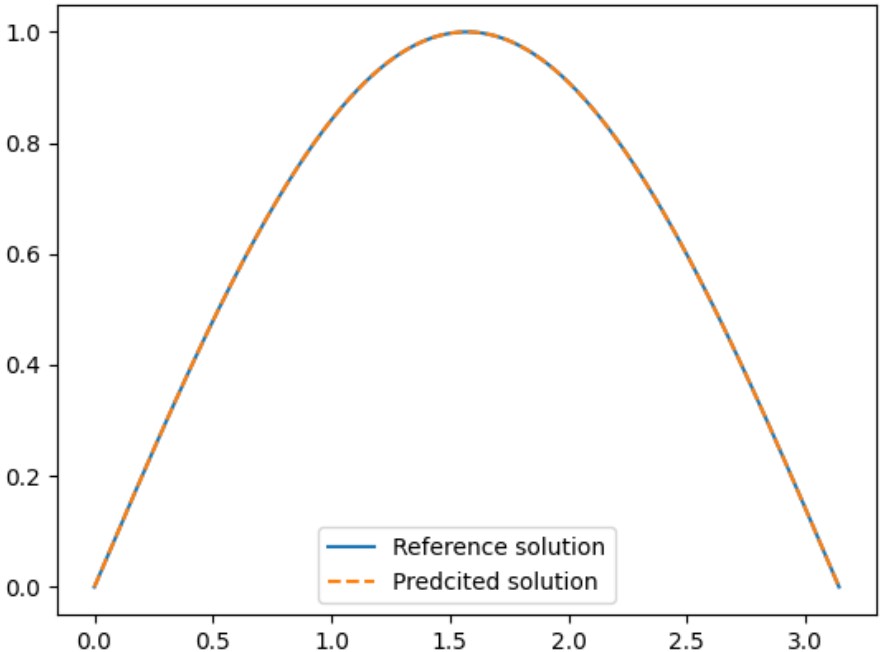

Figure 3: Reference Solution and Predicted Solution for the 1D Poisson Equation.

## A.2 2D POISSON EQUATION

We consider a 2D Poisson equation in the domain $\Omega = [0,1] \times [0,1]$,

$$\begin{cases} -\frac{\partial^2 u}{\partial x^2} - \frac{\partial^2 u}{\partial y^2} = f(x,y), & (x,y) \in \Omega, \\ u(x,y) = 0, & (x,y) \in \partial\Omega. \end{cases} \tag{23}$$

The true solution is given by $u(x,y) = \sin(\pi x)\sin(\pi y)$, and the force term $f(x,y) = 2\pi^2 \sin(\pi x)\sin(\pi y)$ is consequently derived.

We sample $N_b = 200$ random points on the boundary $\partial\Omega$ and $N_f = 1,000$ random points within the domain $\Omega$. We employ a single hidden layer model with 128 units and $\tanh(\cdot)$ activation functions across all computations. We run the NGD method for 200 epochs, while the L-BFGS method is trained for 1 epoch with a maximum of $5,000$ iterations per epoch. All other optimization methods are trained for $20,000$ epochs. The resulting relative $L^2$-error is $1.12e - 04$. Figure 5 illustrates the prediction of the 2D Poisson equation, along with the exact solution and the absolute error between them. It is clear that the predicted solution closely matches the reference solution, further demonstrating the superior performance of the NGD method. Figure 6 shows the loss decay during training, demonstrating that the NGD method converges significantly faster than other optimization methods.

## A.3 1D HEAT EQUATION

We consider the 1D heat equation

$$\begin{cases} \frac{\partial u(t,x)}{\partial t} = \frac{1}{4}\frac{\partial^2 u(t,x)}{\partial x^2}, & (t,x) \in [0,1]^2, \\ u(0,x) = \sin(\pi x), & x \in [0,1], \\ u(t,x) = 0, & (t,x) \in [0,1] \times \{0,1\}. \end{cases} \tag{24}$$

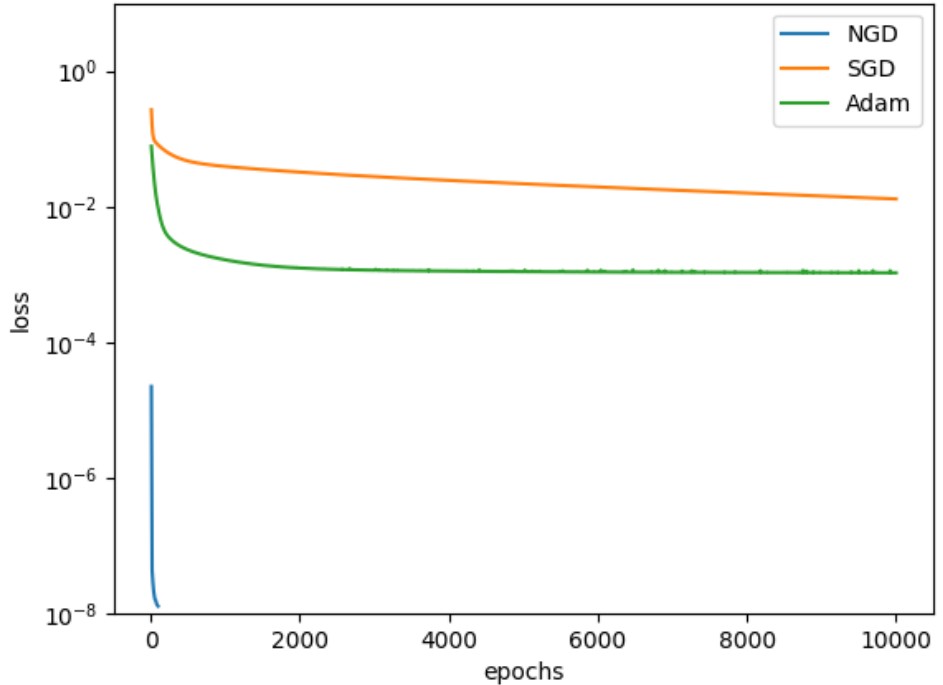

Figure 4: Loss Decay for the 1D Poisson Equation.

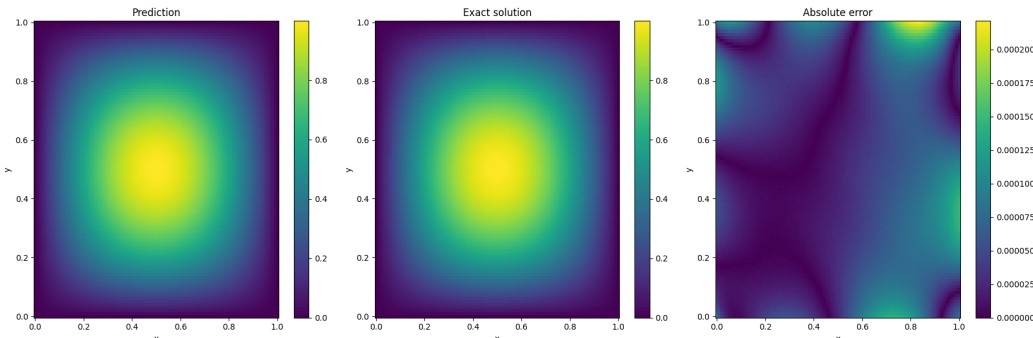

Figure 5: NGD Prediction and Analysis for the 2D Poisson Equation.

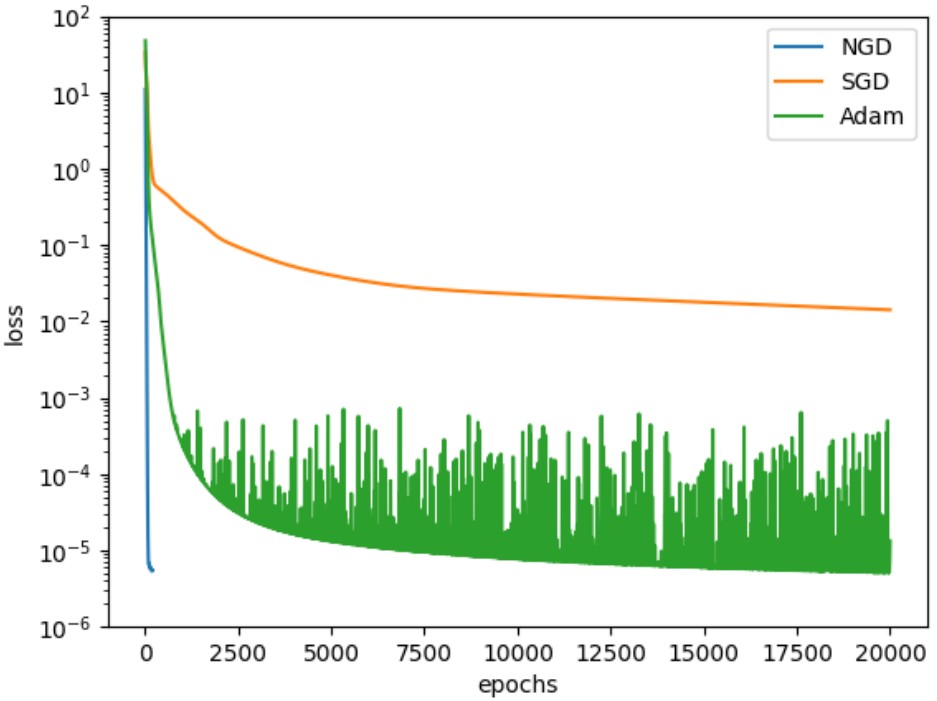

Figure 6: Loss Decay for the 2D Poisson Equation.

The reference solution is analytically defined by $u(t, x) = \exp(-\frac{\pi^2 t}{4}) \sin(\pi x)$. We generate $N_b = 200$ random sampling points for the boundary and initial conditions and $N_f = 1,000$ random points in the domain $\Omega$ to evaluate the PDE residual. The neural network used for all computations consists of 1 hidden layer, each containing 128 neurons with $\tanh(\cdot)$ activation functions. To train the model, we run the NGD method for 200 epochs and the L-BFGS method for 1 epoch with a maximum of $5,000$ iterations per epoch, and other optimizers are trained for $10,000$ epochs. The resulting relative $L^2$-error is $3.42e - 04$. Figure 7 provides a visual comparison between the predicted and exact solutions for the 1D heat equation, along with the corresponding absolute error distribution. The high degree of accuracy in the predicted solution demonstrates the effectiveness of the NGD method, showing its ability to capture the solution with remarkable precision. Figure 8 shows the loss curve over the course of training for the 1D heat equation. Notably, the NGD method rapidly reduces the loss, reaching a low value in the training process, demonstrating its efficiency in optimization.

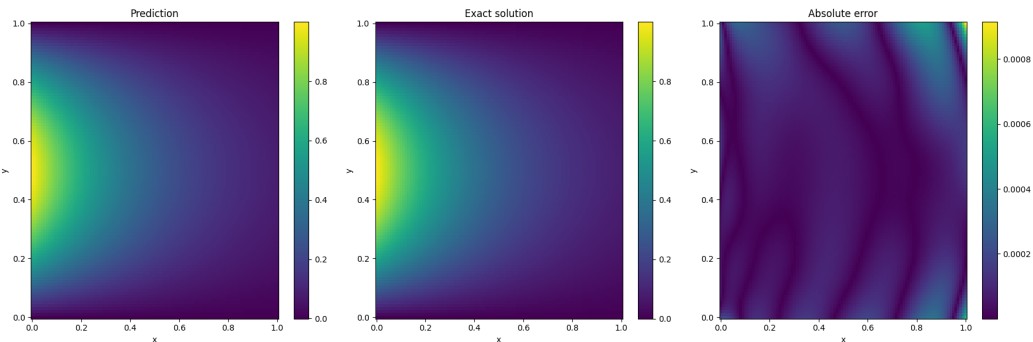

Figure 7: NGD Prediction and Analysis for the 1D Heat Equation.

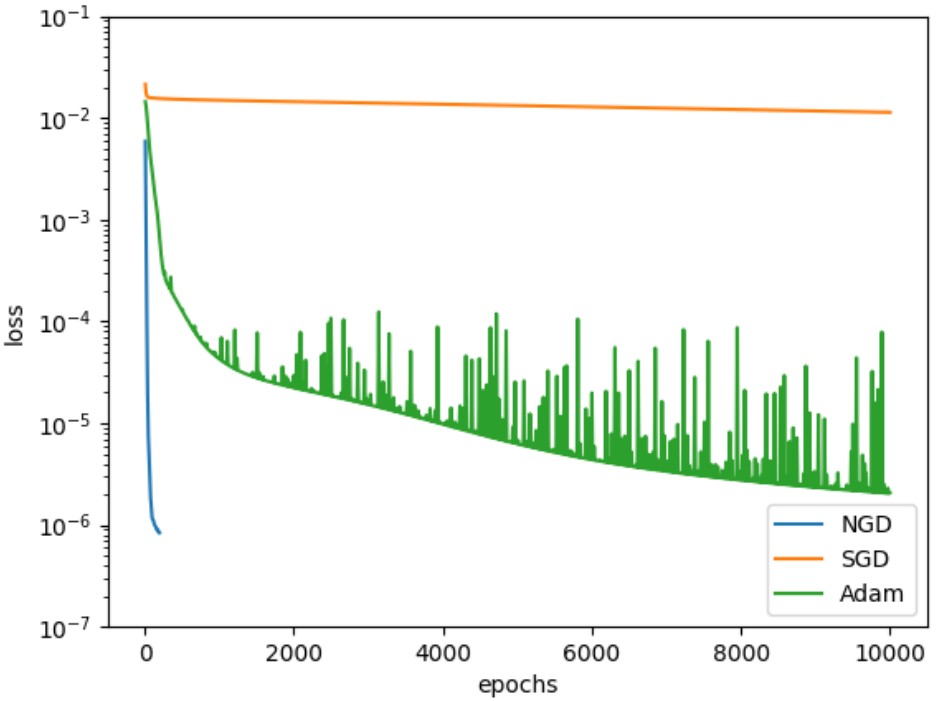

Figure 8: Loss Decay for the 1D Heat Equation.

## A.4    2D HELMHOLTZ EQUATION

We deal with the 2D helmholtz equation on the domain $\Omega = [0, 1] \times [0, 1]$ given by

$$\begin{cases} \frac{\partial^2 u}{\partial x^2} + \frac{\partial^2 u}{\partial y^2} + k^2 u(x, y) = f(x, y), & (x, y) \in \Omega, \\ u(x, y) = 0, & (x, y) \in \partial\Omega. \end{cases} \qquad (25)$$

The reference solution for $k = 4$ is $u(x, y) = \sin(\pi x) \sin(4\pi y)$, and the force term $f(x, y)$ can be easily computed. To evaluate the performance of the NGD approach on the 2D Helmholtz equation, we generate $N_b = 200$ random boundary points on $\partial\Omega$ and $N_f = 1,000$ random points inside the domain $\Omega$. The neural network employed consists of 1 hidden layers with 128 neurons per layer, utilizing $\tanh(\cdot)$ activation functions. Training is carried out for 200 epochs using the NGD method and 1 epoch with a maximum of $5,000$ iterations for L-BFGS. All other optimizers are run for $20,000$ epochs for comparison. The computed relative $L^2$-error is $6.67e - 03$, which is 3 orders of magnitude lower than those of the remaining optimizers. Figure 9 illustrates the predicted solution along with the exact reference solution and the absolute error distribution. The results indicate that the NGD method effectively captures the oscillatory nature of the Helmholtz equation, achieving a high level of accuracy. Figure 10 shows the evolution of the loss function during training for the 2D Helmholtz equation. In particular, the NGD method demonstrates rapid convergence, achieving a low loss value at the end of the training process.

## A.5    10D POISSON EQUATION

We conduct experiments to show that the NGD can also perform better than SGD, Adam, and L-BFGS for high-dimensional PDEs, despite that all optimizers become more challenging as the dimensionality of PDEs increases. As an example in higher dimensions, we consider again the Poisson equation in 10 spatial dimensions

$$\begin{cases} -\Delta u = f(x), & x \in \Omega = [0, 1]^{10}, \\ u(x) = \sum_{k=1}^{10} \sin(\pi x_k), & x \in \partial\Omega. \end{cases} \qquad (26)$$

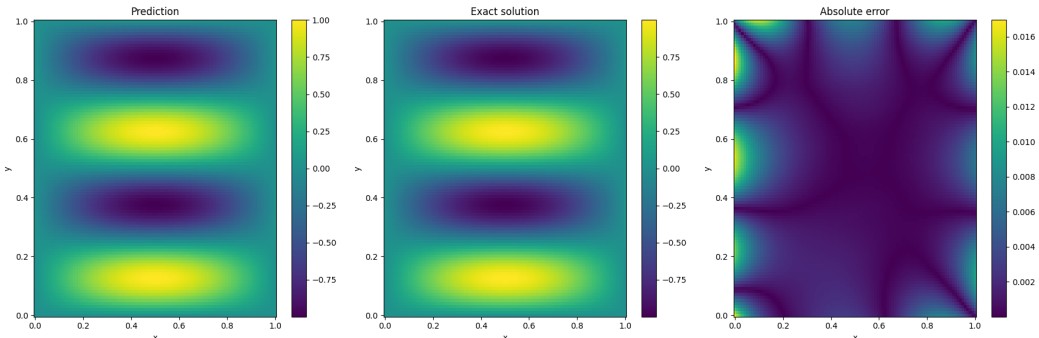

Figure 9: NGD Prediction and Analysis for the 2D Helmholtz Equation.

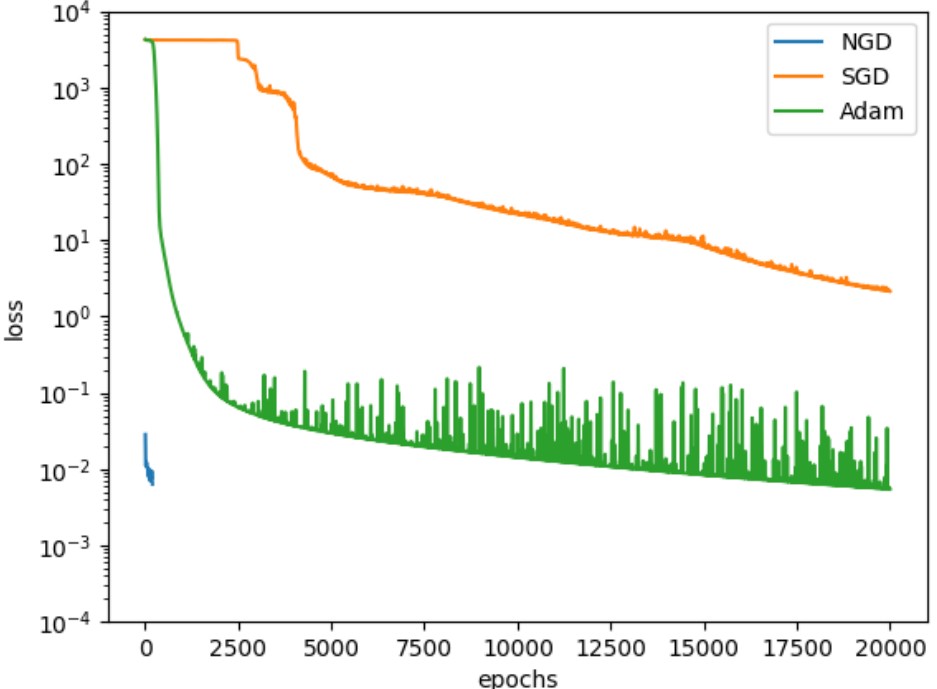

Figure 10: Loss Decay for the 2D Helmholtz Equation.

We use the manufactured solution

$$u^* : \mathbb{R}^{10} \to \mathbb{R}, \quad x \to \sum_{k=1}^{10} \sin(\pi x_k) \tag{27}$$

hence $f = \pi^2 u^*$. We sample $N_b = 1,000$ random points on the boundary $\partial\Omega$ and $N_f = 10,000$ random points within the domain $\Omega$. We employ a single hidden layer model with 128 units and $\tanh(\cdot)$ activation functions across all computations We run the NGD method for 200 epochs, while the L-BFGS method is trained for 1 epoch with a maximum of $5,000$ iterations per epoch. All other optimization methods are trained for $20,000$ epochs. The resulting relative $L^2$-error is $9.91e - 04$. It is clear that the predicted solution closely matches the reference solution, further demonstrating the superior performance of the NGD method.

## A.6 TRAINING EFFICIENCY COMPARISON

For training efficiency comparison among different optimizers, we present the computational time, memory usage, and error rates for both the 2D Poisson equation and the 10D Poisson equation. Table 5 and Table 6 demonstrate that training time and memory requirements increase for all four optimizers as the problem dimension grows. Despite this, the NGD method still achieves the lowest error while maintaining comparable computational overhead.

Table 5: Training efficiency comparison for 2D Poisson equation.

| Optimizers | $lr$ | Epochs | Training efficiency | Training time | Max memory | Rel. L2 error |
|---|---|---|---|---|---|---|
| SGD | 0.001 | 20,000 | 0.047 s/epoch | 15min49s | 14.62 MB | 1.45e-01 |
| Adam | 0.001 | 20,000 | 0.054 s/epoch | 18min2s | 14.75 MB | 5.32e-03 |
| L-BFGS | - | 200 | 0.51 s/epoch | **1min41s** | 41.53 MB | 3.17e-03 |
| NGD | 0.1 | 200 | 3.67 s/epoch | 12min13s | 14.75 MB | **1.12e-04** |

Table 6: Training efficiency comparison for 10D Poisson equation.

| Optimizers | $lr$ | Epochs | Training efficiency | Training time | Max memory | Rel. L2 error |
|---|---|---|---|---|---|---|
| SGD | 0.001 | 20,000 | 0.92 s/epoch | 2h33min | 328.11 MB | 1.05e-02 |
| Adam | 0.001 | 20,000 | 0.95 s/epoch | 2h39min | 328.11 MB | 2.31e-03 |
| L-BFGS | - | 200 | 26.1 s/epoch | **1h27min** | 349.17 MB | nan |
| NGD | 0.1 | 200 | 37.1 s/epoch | 2h4min | 328.11 MB | **9.91e-04** |

## A.7 NUMERICAL EXAMINATION FOR MULTI-LAYER PINNS

While our convergence proof is based on the two-layer PINNs for simplicity, the extension to practical multi-layer PINNs are missing. The restriction of two-layer is primarily technical: it enables precise control of the NTK evolution and allows us to rigorously establish Jacobian stability (Lemma 4.6) and global convergence (Theorem 4.7). Extending these results to deeper networks is indeed possible but significantly more involved, as it requires layer-wise coupling analysis of the NTK (as in Allen-Zhu et al., 2019). Nevertheless, we report the NGD for different layer PINNs on 2D Poisson equation, to show the NGD can converge as depth increases. We train NGD with learning rate 0.1, and each are trained with 200 epochs. Table 7 shows the convergence trends remain consistent with our theoretical predictions: NGD maintains a small relatively $L^2$ error for different layers, while the overall convergence slows moderately as depth increases. Especially the NGD's memory requirements keeps almost the same as layers increased (note that the inverse of $JJ^T$ is independent of the parameters), and the computational burden only increase almost linearly with the total parameters.

## A.8 GENERALIZATION LOSS EXAMINATION

While our theoretical analysis focuses on the optimization of the training loss, the behavior of the generalization error after training is not covered by our current convergence guarantees and is beyond

Table 7: Training comparison of NGD for different layers on 2D Poisson equation.

| Hidden layers | Total parameters | Training efficiency | Training time | Max memory | Rel. L2 error |
|---|---|---|---|---|---|
| 1 | 512 | 3.67 s/epoch | 12min13s | 14.75 MB | **1.12e-04** |
| 3 | 33, 280 | 8.29 s/epoch | 27min37s | 14.75 MB | 3.41e-04 |
| 6 | 82, 432 | 17.53 s/epoch | 58min25s | 20.17 MB | 4.29e-04 |

the scope of this work. Nevertheless, we provide an empirical study to examine how the number of collocation points affects overfitting for different optimizers in the over-parameterized regime. We consider the 2D Poisson equation and vary the total number of collocation points $N = N_f + N_b$ used to train the physics-informed loss (5). NGD is trained with a learning rate of 0.1 for 200 epochs to ensure stable convergence of the training loss. SGD and Adam are trained for 20,000 epochs with a learning rate of 1e-3, and L-BFGS is run for one epoch with a maximum of 50,000 iterations. To approximate the generalization error, we evaluate the physics-informed loss (5) on a very fine grid with $N_f = 100,000$ interior points and $N_b = 1,000$ boundary points. As shown in Table 8, increasing the number of collocation points consistently reduces the generalization error, bringing it close to the training loss. For the 2D Poisson problem considered here, using approximately $N = 5,000$ samples appears sufficient to mitigate overfitting while maintaining small generalization error across all examined optimizers.

Table 8: Generalization error comparison using different collocation points.

| Optimizers | Training loss | $N = 100$ | $N = 500$ | $N = 1,000$ | $N = 5,000$ | $N = 20,000$ |
|---|---|---|---|---|---|---|
| SGD | 2.13e-03 | 5.80e-02 | 1.37e-02 | 1.03e-02 | 2.59e-03 | 1.68e-03 |
| Adam | 9.71e-06 | 1.03e-03 | 4.59e-04 | 1.24e-04 | 3.41e-05 | 1.39e-05 |
| L-BFGS | 7.74e-06 | 9.16e-04 | 4.31e-05 | 3.93e-05 | 1.02e-05 | 8.65e-06 |
| NGD | 2.86e-06 | 2.51e-04 | 2.04e-05 | 1.22e-05 | 2.78e-06 | 2.91e-06 |

## B PROOF OF SECTION 3

Before the proofs, we first define the event

$$A_{ir} := \{\exists \boldsymbol{w} : \|\boldsymbol{w} - \boldsymbol{w}_r(0)\|_2 \leq R, I\{\boldsymbol{w}^T \boldsymbol{x}_i \geq 0\} \neq I\{\boldsymbol{w}_r(0)^T \boldsymbol{x}_i \geq 0\}\} \tag{28}$$

for all $i \in [n]$.

Note that the event happens if and only if $|\boldsymbol{w}_r(0)^T \boldsymbol{x}_i| < \|\boldsymbol{x}_i\|_2 R$, thus by the anti-concentration inequality of Gaussian distribution, we have

$$P(A_{ir}) = P_{z \sim \mathcal{N}(0, \|\boldsymbol{x}_i\|_2^2)}(|z| < R) = P_{z \sim \mathcal{N}(0,1)}(|z| < R) \leq \frac{2R}{\sqrt{2\pi}}. \tag{29}$$

Let $S_i = \{r \in [m] : I\{A_{ir}\} = 0\}$ and $S_i^\perp = [m] \backslash S_i$.

Then, we need to recall that

$$\frac{\partial s_p(\boldsymbol{w})}{\partial \boldsymbol{w}_r} = \frac{a_r}{\sqrt{mn_1}} \left[ \sigma''(\boldsymbol{w}_r^T \boldsymbol{x}_p) w_{r0} \boldsymbol{x}_p + \sigma'(\boldsymbol{w}_r^T \boldsymbol{x}_p) \begin{pmatrix} 1 \\ \mathbf{0}_{d+1} \end{pmatrix} - \sigma'''(\boldsymbol{w}_r^T \boldsymbol{x}_p)\|\boldsymbol{w}_{r1}\|_2^2 \boldsymbol{x}_p - 2\sigma''(\boldsymbol{w}_r^T \boldsymbol{x}_p) \begin{pmatrix} 0 \\ \boldsymbol{w}_{r1} \end{pmatrix} \right] \tag{30}$$

and

$$\frac{\partial h_j(\boldsymbol{w})}{\partial \boldsymbol{w}_r} = \frac{a_r}{\sqrt{mn_2}} \sigma'(\boldsymbol{w}_r^T \boldsymbol{y}_j) \boldsymbol{y}_j. \tag{31}$$

### B.1 PROOF OF LEMMA 3.3

*Proof.* In the following, we aim to bound $\|\boldsymbol{H}(0) - \boldsymbol{H}^\infty\|_F$, as $\|\boldsymbol{H}(0) - \boldsymbol{H}^\infty\|_2 \leq \|\boldsymbol{H}(0) - \boldsymbol{H}^\infty\|_F$. Note that the entries of $\boldsymbol{H}(0) - \boldsymbol{H}^\infty$ have three forms as follows.

$$\sum_{r=1}^m \left\langle \frac{\partial s_i(\boldsymbol{w}(0))}{\partial \boldsymbol{w}_r}, \frac{\partial s_j(\boldsymbol{w}(0))}{\partial \boldsymbol{w}_r} \right\rangle - \mathbb{E}_{\boldsymbol{w}(0)} \left[ \sum_{r=1}^m \left\langle \frac{\partial s_i(\boldsymbol{w}(0))}{\partial \boldsymbol{w}_r}, \frac{\partial s_j(\boldsymbol{w}(0))}{\partial \boldsymbol{w}_r} \right\rangle \right], \tag{32}$$

$$\sum_{r=1}^{m} \left\langle \frac{\partial s_i(\boldsymbol{w}(0))}{\partial \boldsymbol{w}_r}, \frac{\partial h_j(\boldsymbol{w}(0))}{\partial \boldsymbol{w}_r} \right\rangle - \mathbb{E}_{\boldsymbol{w}(0)} \left[ \sum_{r=1}^{m} \left\langle \frac{\partial s_i(\boldsymbol{w}(0))}{\partial \boldsymbol{w}_r}, \frac{\partial h_j(\boldsymbol{w}(0))}{\partial \boldsymbol{w}_r} \right\rangle \right] \tag{33}$$

and

$$\sum_{r=1}^{m} \left\langle \frac{\partial h_i(\boldsymbol{w}(0))}{\partial \boldsymbol{w}_r}, \frac{\partial h_j(\boldsymbol{w}(0))}{\partial \boldsymbol{w}_r} \right\rangle - \mathbb{E}_{\boldsymbol{w}} \left[ \sum_{r=1}^{m} \left\langle \frac{\partial h_i(\boldsymbol{w}(0))}{\partial \boldsymbol{w}_r}, \frac{\partial h_j(\boldsymbol{w}(0))}{\partial \boldsymbol{w}_r} \right\rangle \right]. \tag{34}$$

For the first form (30), to simplify the analysis, we let

$$\boldsymbol{Z}_r(i) = \sigma^{''}(\boldsymbol{w}_r(0)^T \boldsymbol{x}_i) w_{r0}(0) \boldsymbol{x}_i + \sigma^{'}(\boldsymbol{w}_r(0)^T \boldsymbol{x}_i) \begin{pmatrix} 1 \\ \boldsymbol{0}_{d+1} \end{pmatrix}$$

$$- \sigma^{'''}(\boldsymbol{w}_r(0)^T \boldsymbol{x}_p) \|\boldsymbol{w}_{r1}(0)\|_2^2 \boldsymbol{x}_p - 2\sigma^{''}(\boldsymbol{w}_r(0)^T \boldsymbol{x}_i) \begin{pmatrix} 0 \\ \boldsymbol{w}_{r1}(0) \end{pmatrix}$$

and

$$X_r(ij) = \langle \boldsymbol{Z}_r(i), \boldsymbol{Z}_r(j) \rangle,$$

then

$$\sum_{r=1}^{m} \left\langle \frac{\partial s_p(\boldsymbol{w}(0))}{\partial \boldsymbol{w}_r}, \frac{\partial s_j(\boldsymbol{w}(0))}{\partial \boldsymbol{w}_r} \right\rangle - \mathbb{E}_{\boldsymbol{w}} \left[ \sum_{r=1}^{m} \left\langle \frac{\partial s_p(\boldsymbol{w}(0))}{\partial \boldsymbol{w}_r}, \frac{\partial s_j(\boldsymbol{w}(0))}{\partial \boldsymbol{w}_r} \right\rangle \right] = \frac{1}{n_1 m} \sum_{r=1}^{m} [X_r(ij) - \mathbb{E} X_r(ij)].$$

Note that $|X_r(ij)| \lesssim 1 + \|\boldsymbol{w}_r(0)\|_2^4$, thus

$$\|X_r(ij)\|_{\psi_{\frac{1}{2}}} \lesssim 1 + \left\| \|\boldsymbol{w}_r(0)\|_2^4 \right\|_{\psi_{\frac{1}{2}}} \lesssim 1 + \left\| \|\boldsymbol{w}_r(0)\|_2^2 \right\|_{\psi_1}^2 \lesssim d^2.$$

Here, for more details on the Orlicz norm, see the remarks after Lemma D.1.

For the centered random variable, the property of $\psi_{\frac{1}{2}}$ quasi-norm implies that

$$\|X_r(ij) - \mathbb{E}[X_r(ij)]\|_{\psi_{\frac{1}{2}}} \lesssim \|X_r(ij)\|_{\psi_{\frac{1}{2}}} + \|\mathbb{E}[X_r(ij)]\|_{\psi_{\frac{1}{2}}} \lesssim d^2.$$

Therefore, applying Lemma D.1 yields that with probability at least $1 - \delta$,

$$\left| \sum_{r=1}^{m} \frac{1}{m} [X_r(ij) - \mathbb{E} X_r(ij)] \right| \lesssim \frac{d^2}{\sqrt{m}} \sqrt{\log\left(\frac{1}{\delta}\right)} + \frac{d^2}{m} \left(\log\left(\frac{1}{\delta}\right)\right)^2,$$

which directly yields that

$$\left| \sum_{r=1}^{m} \left\langle \frac{\partial s_p(\boldsymbol{w}(0))}{\partial \boldsymbol{w}_r}, \frac{\partial s_j(\boldsymbol{w}(0))}{\partial \boldsymbol{w}_r} \right\rangle - \mathbb{E}_{\boldsymbol{w}(0)} \left[ \sum_{r=1}^{m} \left\langle \frac{\partial s_p(\boldsymbol{w}(0))}{\partial \boldsymbol{w}_r}, \frac{\partial s_j(\boldsymbol{w}(0))}{\partial \boldsymbol{w}_r} \right\rangle \right] \right| \lesssim \frac{d^2}{n_1 \sqrt{m}} \sqrt{\log\left(\frac{1}{\delta}\right)} + \frac{d^2}{n_1 m} \left(\log\left(\frac{1}{\delta}\right)\right)^2. \tag{35}$$

Similarly, for the second form (31) and third form (32), we can deduce that

$$\left\| \left\langle \frac{\partial s_i(\boldsymbol{w}(0)}{\partial \boldsymbol{w}_r}, \frac{\partial h_j(\boldsymbol{w}(0))}{\partial \boldsymbol{w}_r} \right\rangle - \mathbb{E}_{\boldsymbol{w}(0)} \left[ \left\langle \frac{\partial s_i(\boldsymbol{w}(0))}{\partial \boldsymbol{w}_r}, \frac{\partial h_j(\boldsymbol{w}(0))}{\partial \boldsymbol{w}_r} \right\rangle \right] \right\|_{\psi_{\frac{1}{2}}} \lesssim \frac{d^2}{\sqrt{n_1 n_2} m}$$

and

$$\left\| \left\langle \frac{\partial h_i(\boldsymbol{w}(0))}{\partial \boldsymbol{w}_r}, \frac{\partial h_j(\boldsymbol{w}(0))}{\partial \boldsymbol{w}_r} \right\rangle - \mathbb{E}_{\boldsymbol{w}(0)} \left[ \left\langle \frac{\partial h_i(\boldsymbol{w}(0))}{\partial \boldsymbol{w}_r}, \frac{\partial h_j(\boldsymbol{w}(0))}{\partial \boldsymbol{w}_r} \right\rangle \right] \right\|_{\psi_{\frac{1}{2}}} \lesssim \frac{d^2}{n_2 m}.$$

Thus applying Lemma D.1 yields that with probability at least $1 - \delta$,

$$\left| \sum_{r=1}^{m} \left\langle \frac{\partial s_i(\boldsymbol{w}(0))}{\partial \boldsymbol{w}_r}, \frac{\partial h_j(\boldsymbol{w}(0))}{\partial \boldsymbol{w}_r} \right\rangle - \mathbb{E}_{\boldsymbol{w}(0)} \left[ \sum_{r=1}^{m} \left\langle \frac{\partial s_i(\boldsymbol{w}(0))}{\partial \boldsymbol{w}_r}, \frac{\partial h_j(\boldsymbol{w}(0))}{\partial \boldsymbol{w}_r} \right\rangle \right] \right| \lesssim \frac{d^2}{\sqrt{n_1 n_2}\sqrt{m}} \sqrt{\log\left(\frac{1}{\delta}\right)} + \frac{d^2}{\sqrt{n_1 n_2} m} \log\left(\frac{1}{\delta}\right) \tag{36}$$

and with probability at least $1 - \delta$,

$$\left| \sum_{r=1}^{m} \left\langle \frac{\partial h_i(\boldsymbol{w}(0))}{\partial \boldsymbol{w}_r}, \frac{\partial h_j(\boldsymbol{w}(0))}{\partial \boldsymbol{w}_r} \right\rangle - \mathbb{E}_{\boldsymbol{w}(0)} \left[ \sum_{r=1}^{m} \left\langle \frac{\partial h_i(\boldsymbol{w}(0))}{\partial \boldsymbol{w}_r}, \frac{\partial h_j(\boldsymbol{w}(0))}{\partial \boldsymbol{w}_r} \right\rangle \right] \right| \lesssim \frac{d^2}{n_2 \sqrt{m}} \sqrt{\log\left(\frac{1}{\delta}\right)} + \frac{d^2}{n_2 m} \log\left(\frac{1}{\delta}\right).$$

$$(37)$$

Combining the above we can deduce that with probability at least $1 - \delta$,

$$\|\boldsymbol{H}(0) - \boldsymbol{H}^{\infty}\|_2^2$$
$$\leq \|\boldsymbol{H}(0) - \boldsymbol{H}^{\infty}\|_F^2$$
$$\lesssim \frac{d^4}{m} \log\left(\frac{n_1 + n_2}{\delta}\right) + \frac{d^4}{m^2} \left(\log\left(\frac{n_1 + n_2}{\delta}\right)\right)^4$$
$$\lesssim \frac{d^4}{m} \log\left(\frac{n_1 + n_2}{\delta}\right).$$

Thus when $\sqrt{\frac{d^4}{m} \log\left(\frac{n_1 + n_2}{\delta}\right)} \lesssim \frac{\lambda_0}{4}$, i.e.,

$$m = \Omega\left(\frac{d^4}{\lambda_0^2} \log\left(\frac{n_1 + n_2}{\delta}\right)\right),$$

we have $\lambda_{min}(\boldsymbol{H}(0)) \geq \frac{3}{4} \lambda_0$.

$$\square$$

## B.2 PROOF OF LEMMA 3.5

*Proof.* We first reformulate the term $\frac{\partial s_p(k)}{\partial \boldsymbol{w}_r}$ in (28) as follows.

$$\frac{\partial s_p(\boldsymbol{w})}{\partial \boldsymbol{w}_r} = \frac{a_r}{\sqrt{mn_1}} \left[ \sigma''(\boldsymbol{w}_r^T \boldsymbol{x}_p) \begin{pmatrix} w_{r0} x_{p0} \\ w_{r0} \boldsymbol{x}_{p1} - 2\boldsymbol{w}_{r1} \end{pmatrix} + \sigma'(\boldsymbol{w}_r^T \boldsymbol{x}_p) \begin{pmatrix} 1 \\ \boldsymbol{0}_{d+1} \end{pmatrix} - \sigma'''(\boldsymbol{w}_r^T \boldsymbol{x}_p) \|\boldsymbol{w}_{r1}\|_2^2 \boldsymbol{x}_p \right].$$

It suffices to bound $\|\boldsymbol{H}(\boldsymbol{w}) - \boldsymbol{H}(0)\|_F$, which can in turn allows us to bound each entry of $\boldsymbol{H}(\boldsymbol{w}) - \boldsymbol{H}(0)$.

For $i \in [n_1]$ and $j \in [n_1]$, we have that

$$H_{ij}(\boldsymbol{w}) = \sum_{r=1}^{m} \left\langle \frac{\partial s_i(\boldsymbol{w})}{\partial \boldsymbol{w}_r}, \frac{\partial s_j(\boldsymbol{w})}{\partial \boldsymbol{w}_r} \right\rangle$$

$$= \frac{1}{n_1 m} \sum_{r=1}^{m} \left\langle \sigma''(\boldsymbol{w}_r^T \boldsymbol{x}_i) \begin{pmatrix} w_{r0} x_{i0} \\ w_{r0} \boldsymbol{x}_{i1} - 2\boldsymbol{w}_{r1} \end{pmatrix} + \sigma'(\boldsymbol{w}_r^T \boldsymbol{x}_i) \begin{pmatrix} 1 \\ \boldsymbol{0}_{d+1} \end{pmatrix} - \sigma'''(\boldsymbol{w}_r^T \boldsymbol{x}_i) \|\boldsymbol{w}_{r1}\|_2^2 \boldsymbol{x}_i, \right.$$

$$\left. \sigma''(\boldsymbol{w}_r^T \boldsymbol{x}_j) \begin{pmatrix} w_{r0} x_{j0} \\ w_{r0} \boldsymbol{x}_{j1} - 2\boldsymbol{w}_{r1} \end{pmatrix} + \sigma'(\boldsymbol{w}_r^T \boldsymbol{x}_j) \begin{pmatrix} 1 \\ \boldsymbol{0}_{d+1} \end{pmatrix} - \sigma'''(\boldsymbol{w}_r^T \boldsymbol{x}_j) \|\boldsymbol{w}_{r1}\|_2^2 \boldsymbol{x}_j \right\rangle$$

After expanding the inner product term, we can find that although it has nine terms, it only consists of six classes. For simplicity, we use the following six symbols to represent the corresponding classes.

$$\sigma'' \sigma'', \sigma'' \sigma', \sigma' \sigma', \sigma''' \sigma'', \sigma''' \sigma', \sigma'' \sigma'''.$$

For instance, $\sigma'' \sigma'$ represents

$$\left\langle \sigma''(\boldsymbol{w}_r^T \boldsymbol{x}_i) \begin{pmatrix} w_{r0} x_{i0} \\ w_{r0} \boldsymbol{x}_{i1} - 2\boldsymbol{w}_{r1} \end{pmatrix}, \sigma'(\boldsymbol{w}_r^T \boldsymbol{x}_j) \begin{pmatrix} 1 \\ \boldsymbol{0}_{d+1} \end{pmatrix} \right\rangle, \left\langle \sigma'(\boldsymbol{w}_r^T \boldsymbol{x}_i) \begin{pmatrix} 1 \\ \boldsymbol{0}_{d+1} \end{pmatrix}, \sigma''(\boldsymbol{w}_r^T \boldsymbol{x}_j) \begin{pmatrix} w_{r0} x_{j0} \\ w_{r0} \boldsymbol{x}_{j1} - 2\boldsymbol{w}_{r1} \end{pmatrix} \right\rangle.$$

In fact, when bounding the corresponding terms for $H_{ij}(\boldsymbol{w}) - H_{ij}(0)$, the first four classes can be grouped into one category. They are of the form $f_1(\boldsymbol{w}) f_2(\boldsymbol{w}) f_3(\boldsymbol{w}) f_4(\boldsymbol{w})$, where for each $i$ $(1 \leq i \leq 4)$, $f_i(\boldsymbol{w})$ is Lipschitz continuous with respect to $\|\cdot\|_2$ and $|f_i(\boldsymbol{w})| \lesssim \|\boldsymbol{w}\|_2$ (Note that $\sigma'(\cdot) = (\sigma''(\cdot))^2$). On the other hand, when $\|\boldsymbol{w}_1 - \boldsymbol{w}_2\|_2 \leq R \leq 1$, we can deduce that

$$|f_1(\boldsymbol{w}_1) f_2(\boldsymbol{w}_1) f_3(\boldsymbol{w}_1) f_4(\boldsymbol{w}_1) - f_1(\boldsymbol{w}_2) f_2(\boldsymbol{w}_2) f_3(\boldsymbol{w}_2) f_4(\boldsymbol{w}_2)| \lesssim R(\|\boldsymbol{w}_1\|_2^3 + 1).$$

Thus, for the terms in $H_{ij}(\boldsymbol{w}) - H_{ij}(0)$ that belong to the first four classes, we can deduce that they are less than $CR(\|\boldsymbol{w}_r(0)\|_2^3 + 1)$, where $C$ is a universal constant.

For the classes $\sigma^{'''}\sigma^{''}$ and $\sigma^{'''}\sigma^{'}$, they are both involving $\sigma^{'''}$ that is not Lipschitz continuous. To make it precise, we write the class $\sigma^{'''}\sigma^{''}$ explicitly as follows.

$$\sigma^{''}(\boldsymbol{w}_r^T \boldsymbol{x}_i)\sigma^{'''}(\boldsymbol{w}_r^T \boldsymbol{x}_j)\|\boldsymbol{w}_{r1}\|_2^2 \begin{pmatrix} w_{r0}x_{i0} \\ w_{r0}x_{i1} - 2\boldsymbol{w}_{r1} \end{pmatrix}^T \boldsymbol{x}_j.$$

Note that when $\|\boldsymbol{w}_r - \boldsymbol{w}_r(0)\|_2 < R$, we have that

$$|\sigma^{'''}(\boldsymbol{w}_r^T \boldsymbol{x}_j) - \sigma^{'''}(\boldsymbol{w}_r(0)^T \boldsymbol{x}_j)| = |I\{\boldsymbol{w}_r^T \boldsymbol{x}_j \geq 0\} - I\{\boldsymbol{w}_r(0)^T \boldsymbol{x}_j \geq 0\}| \leq I\{A_{jr}\},$$

where the event $A_{jr}$ has been defined in (36).

Thus, we can deduce that for the terms in $H_{ij}(\boldsymbol{w}) - H_{ij}(0)$ that belong to the classes $\sigma^{'''}\sigma^{''}$ and $\sigma^{'''}\sigma^{'}$, they are less than

$$C\left[(I\{A_{ir}\} + I\{A_{jr}\})(\|\boldsymbol{w}_r(0)\|_2^3 + 1) + R(\|\boldsymbol{w}_r(0)\|_2^3 + 1)\right],$$

where $C$ is a universal constant.

Similarly, for the last class $\sigma^{'''}\sigma^{'''}$ that are of the form

$$\sigma^{'''}(\boldsymbol{w}_r^T \boldsymbol{x}_i)\sigma^{'''}(\boldsymbol{w}_r^T \boldsymbol{x}_j)\|\boldsymbol{w}_{r1}\|_2^4 \boldsymbol{x}_i^T \boldsymbol{x}_j,$$

we can deduce that

$$|\sigma^{'''}(\boldsymbol{w}_r^T \boldsymbol{x}_i)\sigma^{'''}(\boldsymbol{w}_r^T \boldsymbol{x}_j)\|\boldsymbol{w}_{r1}\|_2^4 \boldsymbol{x}_i^T \boldsymbol{x}_j - \sigma^{'''}(\boldsymbol{w}_r(0)^T \boldsymbol{x}_i)\sigma^{'''}(\boldsymbol{w}_r(0)^T \boldsymbol{x}_j)\|\boldsymbol{w}_{r1}(0)\|_2^4 \boldsymbol{x}_i^T \boldsymbol{x}_j|$$
$$\lesssim I\{A_{ir} \vee A_{jr}\}\|\boldsymbol{w}_r(0)\|_2^4 + R(\|\boldsymbol{w}_r(0)\|_2^3 + 1).$$

Combining the upper bounds for the terms in the six classes, we have that

$$|H_{ij}(\boldsymbol{w}) - H_{ij}(0)| \lesssim \frac{1}{n_1}\left[\frac{1}{m}\left(R\sum_{r=1}^m \|\boldsymbol{w}_r(0)\|_2^3\right) + \frac{1}{m}\sum_{r=1}^m (I\{A_{ir}\} + I\{A_{jr}\})(\|\boldsymbol{w}_r(0)\|_2^4 + \|\boldsymbol{w}_r(0)\|_2^3 + 1) + R\right]$$
$$\lesssim \frac{1}{n_1}\left[\frac{1}{m}\left(R\sum_{r=1}^m \|\boldsymbol{w}_r(0)\|_2^4\right) + \frac{1}{m}\sum_{r=1}^m (I\{A_{ir}\} + I\{A_{jr}\})(\|\boldsymbol{w}_r(0)\|_2^4 + 1) + R\right],$$

(38)

where the last inequality follows from that $\|\boldsymbol{w}_r(0)\|_2^3 \lesssim \|\boldsymbol{w}_r(0)\|_2^4 + 1$ due to Young's inequality for products.

Now, we focus on the term $\frac{1}{m}\sum_{r=1}^m I\{A_{ir}\}\|\boldsymbol{w}_r(0)\|_2^4$.

Since

$$P\left(|w_{ri}(0)|^2 \geq 2\log\left(\frac{2}{\delta}\right)\right) \leq \delta$$

and then

$$P\left(\|\boldsymbol{w}_r(0)\|_2^2 \geq 2(d+2)\log\left(\frac{2(d+2)}{\delta}\right)\right) \leq \delta.$$

This implies that

$$P\left(\exists r \in [m], \|\boldsymbol{w}_r(0)\|_2^2 \geq 2(d+2)\log\left(\frac{2m(d+2)}{\delta}\right)\right) \leq \delta. \tag{39}$$

Let $M = 2(d+2)\log\left(\frac{2m(d+2)}{\delta}\right)$, then

$$\frac{1}{m}\sum_{r=1}^m I\{A_{ir}\}\|\boldsymbol{w}_r(0)\|_2^4$$
$$= \frac{1}{m}\sum_{r=1}^m I\{A_{ir}\}\|\boldsymbol{w}_r(0)\|_2^4 I\{\|\boldsymbol{w}_r(0)\|_2^2 \leq M\} + \frac{1}{m}\sum_{r=1}^m I\{A_{ir}\}\|\boldsymbol{w}_r(0)\|_2^4 I\{\|\boldsymbol{w}_r(0)\|_2^2 > M\}$$
$$\leq \frac{M^2}{m}\sum_{r=1}^m I\{A_{ir}\} + \frac{1}{m}\sum_{r=1}^m \|\boldsymbol{w}_r(0)\|_2^4 I\{\|\boldsymbol{w}_r(0)\|_2^2 > M\}.$$

Applying Bernstein's inequality for the first term yields that with probability at least $1 - e^{-mR}$,

$$\frac{1}{m} \sum_{r=1}^{m} I\{A_{ir}\} \le 4R.$$

Moreover, from (39), we have that with probability at least $1 - \delta$, the second term $I\{\|\boldsymbol{w}_r(0)\|_2^2 > M\} = 0$ holds for all $r \in [m]$.

Thus with probability at least $1 - \delta - n_1 e^{-mR}$, we have that for any $i \in [n_1]$ and $j \in [n_1]$,

$$|H_{ij}(\boldsymbol{w}) - H_{ij}(0)| \lesssim \frac{1}{n_1} \left[ RM^2 + RM^2 + R \right]$$

$$\lesssim \frac{1}{n_1} M^2 R.$$

For $i \in [n_1], j \in [n_1 + 2, n_2]$ and $i \in [n_1 + 1, n_2], j \in [n_2]$, from the form of $\frac{\partial h_j(\boldsymbol{w})}{\partial \boldsymbol{w}_r}$, i.e.,

$$\frac{\partial h_j(\boldsymbol{w})}{\partial \boldsymbol{w}_r} = \frac{a_r}{\sqrt{n_2 m}} \sigma'(\boldsymbol{w}_r^T \boldsymbol{y}_j) \boldsymbol{y}_j,$$

we can obtain similar results for the terms $\left\langle \frac{\partial s_i}{\partial \boldsymbol{w}}, \frac{\partial h_j}{\partial \boldsymbol{w}} \right\rangle$ and $\left\langle \frac{\partial h_i}{\partial \boldsymbol{w}}, \frac{\partial h_j}{\partial \boldsymbol{w}} \right\rangle$.

With all results above, we have that with probability at least $1 - \delta - n_1 e^{-mR}$,

$$\|\boldsymbol{H}(\boldsymbol{w}) - \boldsymbol{H}(0)\|_F \lesssim M^2 R.$$

$\square$

### B.3 PROOF OF LEMMA B.1

Indeed, the stringent requirement of the learning rate in Du et al. (2018) stems from an inadequate decomposition method for the residual. Specifically, in Gao et al. (2023), the decomposition for the residual in the $(k + 1)$-th iteration is same as the one in Du et al. (2018), i.e.,

$$\begin{pmatrix} \boldsymbol{s}(k+1) \\ \boldsymbol{h}(k+1) \end{pmatrix} = \begin{pmatrix} \boldsymbol{s}(k) \\ \boldsymbol{h}(k) \end{pmatrix} + \left[ \begin{pmatrix} \boldsymbol{s}(k+1) \\ \boldsymbol{h}(k+1) \end{pmatrix} - \begin{pmatrix} \boldsymbol{s}(k) \\ \boldsymbol{h}(k) \end{pmatrix} \right], \quad (40)$$

which leads to the requirements that $\eta = \mathcal{O}(\lambda_0)$ and $m = Poly(n_1, n_2, 1/\delta)$. Thus, it requires a new approach to achieve the improvements for $\eta$ and $m$. In fact, we can derive the following recursion formula.

**Lemma B.1.** *For all $k \in \mathbb{N}$, we have*

$$\begin{pmatrix} \boldsymbol{s}(k+1) \\ \boldsymbol{h}(k+1) \end{pmatrix} = (\boldsymbol{I} - \eta \boldsymbol{H}(k)) \begin{pmatrix} \boldsymbol{s}(k) \\ \boldsymbol{h}(k) \end{pmatrix} + \boldsymbol{I}_1(k), \quad (41)$$

*where*

$$\boldsymbol{I}_1(k) = (I_1^1(k), \cdots, I_1^{n_1+n_2}(k))^T \in \mathbb{R}^{n_1+n_2}$$

*and for $p \in [n_1]$,*

$$I_1^p(k) = s_p(k+1) - s_p(k) - \left\langle \frac{\partial s_p(k)}{\partial \boldsymbol{w}}, \boldsymbol{w}(k+1) - \boldsymbol{w}(k) \right\rangle, \quad (42)$$

*for $j \in [n_2]$,*

$$I_1^{n_1+j}(k) = h_j(k+1) - h_j(k) - \left\langle \frac{\partial h_j(k)}{\partial \boldsymbol{w}}, \boldsymbol{w}(k+1) - \boldsymbol{w}(k) \right\rangle. \quad (43)$$

In the recursion formula (39), $\boldsymbol{I}_1(k)$ serves as a residual term. From the proof, we can see that $\|\boldsymbol{I}_1(k)\|_2 = \mathcal{O}(1/\sqrt{m})$ and thus, as $m$ becomes large enough, only the term $\boldsymbol{I} - \eta \boldsymbol{H}(k)$ is significant. This observation is the reason for the requirement of $\eta$.

*Proof.* First, we have

$$s_p(k+1) - s_p(k) = \left[ s_p(k+1) - s_p(k) - \left\langle \frac{\partial s_p(k)}{\partial \boldsymbol{w}}, \boldsymbol{w}(k+1) - \boldsymbol{w}(k) \right\rangle \right] + \left\langle \frac{\partial s_p(k)}{\partial \boldsymbol{w}}, \boldsymbol{w}(k+1) - \boldsymbol{w}(k) \right\rangle$$

$$:= I_1^p(k) + I_2^p(k). \tag{44}$$

For the second term $I_2^p(k)$, from the updating rule of gradient descent, we have that

$$
\begin{aligned}
I_2^p(k) &= \left\langle \frac{\partial s_p(k)}{\partial \boldsymbol{w}}, \boldsymbol{w}(k+1) - \boldsymbol{w}(k) \right\rangle \\
&= \left\langle \frac{\partial s_p(k)}{\partial \boldsymbol{w}}, -\eta \frac{\partial L(k)}{\partial \boldsymbol{w}} \right\rangle \\
&= -\sum_{r=1}^m \eta \left\langle \frac{\partial s_p(k)}{\partial \boldsymbol{w}_r}, \frac{\partial L(k)}{\partial \boldsymbol{w}_r} \right\rangle \\
&= -\sum_{r=1}^m \eta \left\langle \frac{\partial s_p(k)}{\partial \boldsymbol{w}_r}, \sum_{t=1}^{n_1} s_t(k) \frac{\partial s_t(k)}{\partial \boldsymbol{w}_r} + \sum_{j=1}^{n_2} h_j(k) \frac{\partial h_j(k)}{\partial \boldsymbol{w}_r} \right\rangle \\
&= -\eta \left[ \sum_{t=1}^{n_1} \left\langle \frac{\partial s_p(k)}{\partial \boldsymbol{w}_r}, \frac{\partial s_t(k)}{\partial \boldsymbol{w}_r} \right\rangle s_t(k) + \sum_{j=1}^{n_2} \left\langle \frac{\partial s_p(k)}{\partial \boldsymbol{w}_r}, \frac{\partial h_j(k)}{\partial \boldsymbol{w}_r} \right\rangle h_j(k) \right] \\
&= -\eta [\boldsymbol{H}(k)]_p \begin{pmatrix} \boldsymbol{s}(k) \\ \boldsymbol{h}(k) \end{pmatrix},
\end{aligned}
\tag{45}
$$

where $[\boldsymbol{H}(k)]_p$ denotes the $p$-row of $\boldsymbol{H}(k)$.

Similarly, for $h(k)$, we have

$$h_j(k+1) - h_j(k) = \left[ h_j(k+1) - h_j(k) - \left\langle \frac{\partial h_j(k)}{\partial \boldsymbol{w}}, \boldsymbol{w}(k+1) - \boldsymbol{w}(k) \right\rangle \right] + \left\langle \frac{\partial h_j(k)}{\partial \boldsymbol{w}}, \boldsymbol{w}(k+1) - \boldsymbol{w}(k) \right\rangle$$

$$:= I_1^{n_1+j}(k) + I_2^{n_1+j}(k) \tag{46}$$

and

$$I_2^{n_1+j}(k) = -\eta [\boldsymbol{H}(k)]_{n_1+j} \begin{pmatrix} \boldsymbol{s}(k) \\ \boldsymbol{h}(k) \end{pmatrix}. \tag{47}$$

Combining (42), (43), (44) and (45) yields that

$$
\begin{aligned}
\begin{pmatrix} \boldsymbol{s}(k+1) \\ \boldsymbol{h}(k+1) \end{pmatrix} - \begin{pmatrix} \boldsymbol{s}(k) \\ \boldsymbol{h}(k) \end{pmatrix} &= \boldsymbol{I}_1(k) + \boldsymbol{I}_2(k) \\
&= \boldsymbol{I}_1(k) - \eta \boldsymbol{H}(k) \begin{pmatrix} \boldsymbol{s}(k) \\ \boldsymbol{h}(k) \end{pmatrix}.
\end{aligned}
$$

A simple transformation directly leads to

$$\begin{pmatrix} \boldsymbol{s}(k+1) \\ \boldsymbol{h}(k+1) \end{pmatrix} = (\boldsymbol{I} - \eta \boldsymbol{H}(k)) \begin{pmatrix} \boldsymbol{s}(k) \\ \boldsymbol{h}(k) \end{pmatrix} + \boldsymbol{I}_1(k),$$

which is exactly (39), the new recursion formula we need to prove. □

### B.4 PROOF OF THEOREM 3.7

Similar to Du et al. (2018) and Gao et al. (2023), we prove Theorem 3.7 by induction. Our induction hypothesis is the following convergence rate of the empirical loss and upper bounds for the weights.

**Condition 2.** At the $t$-th iteration, we have that for each $r \in [m]$, $\|\boldsymbol{w}_r(t)\|_2 \leq B$ and

$$L(t) \leq \left( 1 - \frac{\eta \lambda_0}{2} \right)^t L(0), \tag{48}$$

where $B = \sqrt{2(d+2) \log \left( \frac{2m(d+2)}{\delta} \right)} + 1$ and $L(k)$ is an abbreviation of $L(\boldsymbol{w}(k))$.

From the update formula of gradient descent, we can directly derive the following corollary, which indicates that under over-parameterization, the weights are closed to their initializations.

**Corollary B.2.** *If Condition 2 holds for $t = 0, \cdots, k$, then we have for every $r \in [m]$,*

$$\|\boldsymbol{w}_r(k+1) - \boldsymbol{w}_r(0)\|_2 \leq \frac{CB^2\sqrt{L(0)}}{\sqrt{m}\lambda_0} := R', \tag{49}$$

*where $C$ is a universal constant.*

**Proof Sketch:** Assume that Condition 2 holds for $t = 0, \cdots, k$, it suffices to demonstrate that Condition 2 also holds for $t = k + 1$.

From the recursion formula (40), we have that

$$
\left\|\begin{pmatrix} \boldsymbol{s}(k+1) \\ \boldsymbol{h}(k+1) \end{pmatrix}\right\|_2^2
$$
$$
= \left\|(\boldsymbol{I} - \eta\boldsymbol{H}(k)) \begin{pmatrix} \boldsymbol{s}(k) \\ \boldsymbol{h}(k) \end{pmatrix} + \boldsymbol{I}_1(k)\right\|_2^2 \tag{50}
$$
$$
\leq \|\boldsymbol{I} - \eta\boldsymbol{H}(k)\|_2^2 \left\|\begin{pmatrix} \boldsymbol{s}(k) \\ \boldsymbol{h}(k) \end{pmatrix}\right\|_2^2 + \|\boldsymbol{I}_1(k)\|_2^2 + 2\|\boldsymbol{I} - \eta\boldsymbol{H}(k)\|_2 \left\|\begin{pmatrix} \boldsymbol{s}(k) \\ \boldsymbol{h}(k) \end{pmatrix}\right\|_2 \|\boldsymbol{I}_1(k)\|_2,
$$

where the inequality follows from the Cauchy's inequality.

Combining Corollary B.2 with Lemma 3.5, we can deduce that when $m$ is large enough, we have $\|\boldsymbol{H}(k) - \boldsymbol{H}(0)\|_2 \leq \lambda_0/4$. Thus, $\lambda_{min}(\boldsymbol{H}(k)) \geq \lambda_0/2$ and $\boldsymbol{I} - \eta\boldsymbol{H}(k)$ is positive definite when $\eta = \mathcal{O}(1/\|\boldsymbol{H}^\infty\|_2)$. On the other hand, with Corollary B.2, we can derive that $\|\boldsymbol{I}_1(k)\|_2 = \mathcal{O}(\eta\sqrt{L(k)}/\sqrt{m})$. Plugging these results into (48), we have

$$
\left\|\begin{pmatrix} \boldsymbol{s}(k+1) \\ \boldsymbol{h}(k+1) \end{pmatrix}\right\|_2^2
$$
$$
= \left(\left(1 - \frac{\eta\lambda_0}{2}\right)^2 + \mathcal{O}\left(\frac{\eta^2}{m}\right) + \mathcal{O}\left(\frac{\eta}{\sqrt{m}}\right)\right) \left\|\begin{pmatrix} \boldsymbol{s}(k) \\ \boldsymbol{h}(k) \end{pmatrix}\right\|_2^2 \tag{51}
$$
$$
\leq \left(1 - \frac{\eta\lambda_0}{2}\right) \left\|\begin{pmatrix} \boldsymbol{s}(k) \\ \boldsymbol{h}(k) \end{pmatrix}\right\|_2^2,
$$

where the last inequality holds when $m$ is large enough.

Now we come to prove Theorem 3.7.

*Proof.* Corollary B.2 implies that when $m$ is large enough, we have $\|\boldsymbol{w}_r(k+1) - \boldsymbol{w}_r(0)\|_2 \leq 1$ and then $\|\boldsymbol{w}_r(k+1)\|_2 \leq B$. Thus, in induction, we only need to prove that (46) also holds for $t = k+1$, which relies on the recursion formula (49).

Recall that the recursion formula is

$$
\begin{pmatrix} \boldsymbol{s}(k+1) \\ \boldsymbol{h}(k+1) \end{pmatrix} = (\boldsymbol{I} - \eta\boldsymbol{H}(k)) \begin{pmatrix} \boldsymbol{s}(k) \\ \boldsymbol{h}(k) \end{pmatrix} + \boldsymbol{I}_1(k).
$$

From Corollary B.2 and Lemma 3.5, taking $CM^2R < \frac{\lambda_0}{4}$ in (8) and $R' \leq R$ in (47) yields that $\lambda_{min}(\boldsymbol{H}(k)) \geq \lambda_{min}(\boldsymbol{H}(0)) - \frac{\lambda_0}{4} \geq \frac{\lambda_0}{2}$ and

$$\|\boldsymbol{H}(k)\|_2 \leq \|\boldsymbol{H}(0)\|_2 + \frac{\lambda_0}{4} \leq \|\boldsymbol{H}^\infty\|_2 + \frac{\lambda_0}{2} \leq \frac{3}{2}\|\boldsymbol{H}^\infty\|_2.$$

Therefore, if we take $\eta \leq \frac{2}{3}\frac{1}{\|\boldsymbol{H}^\infty\|_2}$, then $\boldsymbol{I} - \eta\boldsymbol{H}(k)$ is positive definite and $\|I - \eta\boldsymbol{H}(k)\|_2 \leq 1 - \frac{\eta\lambda_0}{2}$.

Combining these facts with the recursion formula, we have that

$$
\left\| \begin{pmatrix} s(k+1) \\ h(k+1) \end{pmatrix} \right\|_2^2
$$

$$
= \left\| (I - \eta H(k)) \begin{pmatrix} s(k) \\ h(k) \end{pmatrix} \right\|_2^2 + \|I_1(k)\|_2^2 + 2 \left\langle (I - \eta H(k)) \begin{pmatrix} s(k) \\ h(k) \end{pmatrix}, I_1(k) \right\rangle \qquad (52)
$$

$$
\leq \left( 1 - \frac{\eta \lambda_0}{2} \right)^2 \left\| \begin{pmatrix} s(k) \\ h(k) \end{pmatrix} \right\|_2^2 + \|I_1(k)\|_2^2 + 2 \left( 1 - \frac{\eta \lambda_0}{2} \right) \left\| \begin{pmatrix} s(k) \\ h(k) \end{pmatrix} \right\|_2 \|I_1(k)\|_2.
$$

Thus, it remains only to bound $\|I_1(k)\|_2$.

For $I_1(k)$, recall that $I_1(k) = (I_1^1(k), \cdots, I_1^{n_1}(k), I_1^{n_1+1}(k), \cdots, I_1^{n_1+n_2}(k))^T \in \mathbb{R}^{n_1+n_2}$ and for $p \in [n_1]$,

$$
I_1^p(k) = s_p(k+1) - s_p(k) - \left\langle \frac{\partial s_p(k)}{\partial w}, w(k+1) - w(k) \right\rangle,
$$

for $j \in [n_2]$,

$$
I_1^{n_1+j}(k) = h_j(k+1) - h_j(k) - \left\langle \frac{\partial h_j(k)}{\partial w}, w(k+1) - w(k) \right\rangle.
$$

Recall that

$$
s_p(k) = \frac{1}{\sqrt{n_1}} \left( \frac{1}{\sqrt{m}} \left( \sum_{r=1}^m a_r \sigma'(w_r(k)^T x_p) w_{r0}(k) - a_r \sigma''(w_r(k)^T x_p) \|w_{r1}(k)\|_2^2 \right) - f(x_p) \right)
$$

and

$$
\frac{\partial s_p(k)}{\partial w_r} = \frac{a_r}{\sqrt{n_1 m}} \left[ \sigma''(w_r(k)^T x_p) w_{r0}(k) x_p + \sigma'(w_r(k)^T x_p) \begin{pmatrix} 1 \\ 0_{d+2} \end{pmatrix} - \sigma'''(w_r(k)^T x_p) \|w_{r1}(k)\|_2^2 x_p \right.
$$
$$
\left. - 2\sigma''(w_r(k)^T x_p) \begin{pmatrix} 0 \\ w_{r1}(k) \end{pmatrix} \right].
$$

Define $\chi_{pr}^1(k) := \sigma'(w_r(k)^T x_p) w_{r0}(k)$ and $\chi_{pr}^2(k) := \sigma''(w_r(k)^T x_p) \|w_{r1}(k)\|_2^2$, i.e., $\chi_{pr}^1(k)$ and $\chi_{pr}^2(k)$ are related to the operators $\frac{\partial u}{\partial t}$ and $\Delta u$ respectively.

Then define

$$
\hat{\chi}_{pr}^1(k) = \chi_{pr}^1(k+1) - \chi_{pr}^1(k) - \left\langle \frac{\partial \chi_{pr}^1(k)}{\partial w_r}, w_r(k+1) - w_r(k) \right\rangle
$$

and

$$
\hat{\chi}_{pr}^2(k) = \chi_{pr}^2(k+1) - \chi_{pr}^2(k) - \left\langle \frac{\partial \chi_{pr}^2(k)}{\partial w_r}, w_r(k+1) - w_r(k) \right\rangle.
$$

At this time, we have

$$
I_1^p(k) = \frac{1}{\sqrt{n_1 m}} \sum_{r=1}^m a_r \left[ \hat{\chi}_{pr}^1(k) - \hat{\chi}_{pr}^2(k) \right].
$$

The purpose of defining $\hat{\chi}_{pr}^1(k)$ and $\hat{\chi}_{pr}^1(k)$ in this way is to enable us to handle the terms related to the operators $\frac{\partial u}{\partial t}$ and $\Delta u$ separately.

We first recall some definitions. For $p \in [n_1]$,

$$
A_{p,r} = \{\exists w : \|w - w_r(0)\|_2 \leq R, I\{w^T x_p \geq 0\} \neq I\{w_r(0)^T x_p \geq 0\}\}
$$

and $S_p = \{r \in [m] : I\{A_{p,r} = 0\}\}$, $S_p^\perp = [n_1] \backslash S_p$.

In the following, we are going to show that $|\hat{\chi}_{pr}^1(k)| = \mathcal{O}(\|w_r(k+1) - w_r(k)\|_2^2)$ for every $r \in [m]$ and $|\hat{\chi}_{pr}^2(k)| = \mathcal{O}(\|w_r(k+1) - w_r(k)\|_2^2)$ for $r \in S_p$, $|\hat{\chi}_{pr}^2(k)| = \mathcal{O}(\|w_r(k+1) - w_r(k)\|_2)$ for

$r \in S_p^{\perp}$. Thus, we can prove that $\|\boldsymbol{I}_1(k)\|_2 = \mathcal{O}\left(\frac{\sqrt{L(k)}}{\sqrt{m}}\right)$. Then combining with (69) leads to the conclusion.

For $\hat{\chi}_{pr}^1(k)$, from its definition, we have that

$$
\begin{aligned}
\hat{\chi}_{pr}^1(k) &= \sigma'(\boldsymbol{w}_r(k+1)^T \boldsymbol{x}_p) w_{r0}(k+1) - \sigma'(\boldsymbol{w}_r(k)^T \boldsymbol{x}_p) w_{r0}(k) \\
&\quad - \langle \boldsymbol{w}_r(k+1) - \boldsymbol{w}_r(k), \boldsymbol{x}_p \rangle \sigma''(\boldsymbol{w}_r(k)^T \boldsymbol{x}_p) w_{r0}(k) - (w_{r0}(k+1) - w_{r0}(k)) \sigma'(\boldsymbol{w}_r(k)^T \boldsymbol{x}_p) \\
&= (\sigma'(\boldsymbol{w}_r(k+1)^T \boldsymbol{x}_p) - \sigma'(\boldsymbol{w}_r(k)^T \boldsymbol{x}_p)) w_{r0}(k+1) - \langle \boldsymbol{w}_r(k+1) - \boldsymbol{w}_r(k), \boldsymbol{x}_p \rangle \sigma''(\boldsymbol{w}_r(k)^T \boldsymbol{x}_p) w_{r0}(k).
\end{aligned}
$$

From the mean value theorem, we can deduce that there exists $\zeta(k) \in \mathbb{R}$ such that

$$
\sigma'(\boldsymbol{w}_r(k+1)^T \boldsymbol{x}_p) - \sigma'(\boldsymbol{w}_r(k)^T \boldsymbol{x}_p) = \sigma''(\zeta(k)) \langle \boldsymbol{w}_r(k+1) - \boldsymbol{w}_r(k), \boldsymbol{x}_p \rangle
$$

and

$$
\begin{aligned}
|\sigma''(\zeta(k)) - \sigma''(\boldsymbol{w}_r(k)^T \boldsymbol{x}_p)| &\le |\zeta(k) - \boldsymbol{w}_r(k)^T \boldsymbol{x}_p| \\
&\le \sqrt{2} \|\boldsymbol{w}_r(k+1) - \boldsymbol{w}_r(k)\|_2.
\end{aligned}
$$

Then, for $\hat{\chi}_{pr}^1(k)$, we can rewrite it as follows.

$$
\begin{aligned}
\hat{\chi}_{pr}^1(k) &= \sigma''(\zeta(k)) \langle \boldsymbol{w}_r(k+1) - \boldsymbol{w}_r(k), \boldsymbol{x}_p \rangle w_{r0}(k+1) - \langle \boldsymbol{w}_r(k+1) - \boldsymbol{w}_r(k), \boldsymbol{x}_p \rangle \sigma''(\boldsymbol{w}_r(k)^T \boldsymbol{x}_p) w_{r0}(k) \\
&= \left[ \left( \sigma''(\zeta(k)) - \sigma''(\boldsymbol{w}_r(k)^T \boldsymbol{x}_p) \right) \langle \boldsymbol{w}_r(k+1) - \boldsymbol{w}_r(k), \boldsymbol{x}_p \rangle w_{r0}(k+1) \right] \\
&\quad + \left[ \langle \boldsymbol{w}_r(k+1) - \boldsymbol{w}_r(k), \boldsymbol{x}_p \rangle \sigma''(\boldsymbol{w}_r(k)^T \boldsymbol{x}_p)(w_{r0}(k+1) - w_{r0}(k)) \right].
\end{aligned}
$$

This implies that

$$
|\hat{\chi}_{pr}^1(k)| \lesssim B \|\boldsymbol{w}_r(k+1) - \boldsymbol{w}_r(k)\|_2^2.
$$

For $\hat{\chi}_{pr}^2(k)$, we write it as follows explicitly.

$$
\begin{aligned}
\hat{\chi}_{pr}^2(k) = {}& \sigma''(\boldsymbol{w}_r(k+1)^T \boldsymbol{x}_p) \|\boldsymbol{w}_{r1}(k+1)\|_2^2 - \sigma''(\boldsymbol{w}_r(k)^T \boldsymbol{x}_p) \|\boldsymbol{w}_{r1}(k)\|_2^2 \\
& - \langle \boldsymbol{w}_r(k+1) - \boldsymbol{w}_r(k), \boldsymbol{x}_p \rangle \sigma'''(\boldsymbol{w}_r(k)^T \boldsymbol{x}_p) \|\boldsymbol{w}_{r1}(k)\|_2^2 \\
& - 2 \langle \boldsymbol{w}_{r1}(k+1) - \boldsymbol{w}_{r1}(k), \boldsymbol{w}_{r1}(k) \rangle \sigma''(\boldsymbol{w}_r(k)^T \boldsymbol{x}_p).
\end{aligned}
\tag{53}
$$

Note that for the term $\sigma''(\boldsymbol{w}_r(k)^T \boldsymbol{w}_p) \|\boldsymbol{w}_{r1}(k)\|_2^2$, we can rewrite it as follows.

$$
\begin{aligned}
& \sigma''(\boldsymbol{w}_r(k)^T \boldsymbol{x}_p) \|\boldsymbol{w}_{r1}(k)\|_2^2 \\
={}& \sigma''(\boldsymbol{w}_r(k)^T \boldsymbol{x}_p) \|\boldsymbol{w}_{r1}(k) - \boldsymbol{w}_{r1}(k+1) + \boldsymbol{w}_{r1}(k+1)\|_2^2 \\
={}& \sigma''(\boldsymbol{w}_r(k)^T \boldsymbol{x}_p) [\|\boldsymbol{w}_{r1}(k) - \boldsymbol{w}_{r1}(k+1)\|_2^2 + \|\boldsymbol{w}_{r1}(k+1)\|_2^2 - 2\langle \boldsymbol{w}_{r1}(k+1) - \boldsymbol{w}_{r1}(k), \boldsymbol{w}_{r1}(k+1) \rangle],
\end{aligned}
\tag{54}
$$

where the first term $\sigma''(\boldsymbol{w}_r(k)^T \boldsymbol{x}_p) \|\boldsymbol{w}_{r1}(k) - \boldsymbol{w}_{r1}(k+1)\|_2^2 = \mathcal{O}(B \|\boldsymbol{w}_r(k+1) - \boldsymbol{w}_r(k)\|_2^2)$.

Plugging (52) into (51) yields that

$$
\begin{aligned}
\hat{\chi}_{pr}^2(k) = {}& [\sigma''(\boldsymbol{w}_r(k+1)^T \boldsymbol{x}_p) - \sigma''(\boldsymbol{w}_r(k)^T \boldsymbol{x}_p)] \|\boldsymbol{w}_{r1}(k+1)\|_2^2 \\
& - \langle \boldsymbol{w}_r(k+1) - \boldsymbol{w}_r(k), \boldsymbol{x}_p \rangle \sigma'''(\boldsymbol{w}_r(k)^T \boldsymbol{x}_p) \|\boldsymbol{w}_{r1}(k)\|_2^2 \\
& + 2 \langle \boldsymbol{w}_{r1}(k+1) - \boldsymbol{w}_{r1}(k), \boldsymbol{w}_{r1}(k+1) - \boldsymbol{w}_{r1}(k) \rangle \sigma''(\boldsymbol{w}_r(k)^T \boldsymbol{x}_p) + \mathcal{O}(B \|\boldsymbol{w}_r(k+1) - \boldsymbol{w}_r(k)\|_2^2) \\
= {}& [\sigma''(\boldsymbol{w}_r(k+1)^T \boldsymbol{x}_p) - \sigma''(\boldsymbol{w}_r(k)^T \boldsymbol{x}_p) - \langle \boldsymbol{w}_r(k+1) - \boldsymbol{w}_r(k), \boldsymbol{x}_p \rangle \sigma'''(\boldsymbol{w}_r(k)^T \boldsymbol{x}_p)] \|\boldsymbol{w}_{r1}(k+1)\|_2^2 \\
& + \langle \boldsymbol{w}_r(k+1) - \boldsymbol{w}_r(k), \boldsymbol{x}_p \rangle \sigma'''(\boldsymbol{w}_r(k)^T \boldsymbol{x}_p)(\|\boldsymbol{w}_{r1}(k+1)\|_2^2 - \|\boldsymbol{w}_{r1}(k)\|_2^2) \\
& + \mathcal{O}(B \|\boldsymbol{w}_r(k+1) - \boldsymbol{w}_r(k)\|_2^2) \\
= {}& \left[ \sigma''(\boldsymbol{w}_r(k+1)^T \boldsymbol{x}_p) - \sigma''(\boldsymbol{w}_r(k)^T \boldsymbol{x}_p) - \langle \boldsymbol{w}_r(k+1) - \boldsymbol{w}_r(k), \boldsymbol{x}_p \rangle \sigma'''(\boldsymbol{w}_r(k)^T \boldsymbol{x}_p) \right] \|\boldsymbol{w}_{r1}(k+1)\|_2^2 \\
& + \mathcal{O}(B \|\boldsymbol{w}_r(k+1) - \boldsymbol{w}_r(k)\|_2^2).
\end{aligned}
\tag{55}
$$

Thus, we only need to consider the term

$$\sigma''(\boldsymbol{w}_r(k+1)^T\boldsymbol{x}_p) - \sigma''(\boldsymbol{w}_r(k)^T\boldsymbol{x}_p) - \langle\boldsymbol{w}_r(k+1) - \boldsymbol{w}_r(k),\boldsymbol{x}_p\rangle\sigma'''(\boldsymbol{w}_r(k)^T\boldsymbol{x}_p).$$

For $r \in S_p$, since $\|\boldsymbol{w}_r(k+1) - \boldsymbol{w}_r(0)\|_2 \leq R, \|\boldsymbol{w}_r(k) - \boldsymbol{w}_r(0)\|_2 \leq R$, we have that $I\{\boldsymbol{w}_r(k+1)^T\boldsymbol{x}_p \geq 0\} = I\{\boldsymbol{w}_r(k)^T\boldsymbol{x}_p \geq 0\}$, which yields that

$$
\begin{aligned}
&\sigma''(\boldsymbol{w}_r(k+1)^T\boldsymbol{x}_p) - \sigma''(\boldsymbol{w}_r(k)^T\boldsymbol{x}_p) - \langle\boldsymbol{w}_r(k+1) - \boldsymbol{w}_r(k),\boldsymbol{x}_p\rangle\sigma'''(\boldsymbol{w}_r(k)^T\boldsymbol{x}_p) \\
&= [(\boldsymbol{w}_r(k+1)^T\boldsymbol{x}_p)I\{\boldsymbol{w}_r(k+1)^T\boldsymbol{x}_p \geq 0\} - (\boldsymbol{w}_r(k)^T\boldsymbol{x}_p)I\{\boldsymbol{w}_r(k)^T\boldsymbol{x}_p \geq 0\}] \\
&\quad - \langle\boldsymbol{w}_r(k+1) - \boldsymbol{w}_r(k),\boldsymbol{x}_p\rangle I\{\boldsymbol{w}_r(k)^T\boldsymbol{x}_p \geq 0\} \\
&= [(\boldsymbol{w}_r(k+1)^T\boldsymbol{x}_p)I\{\boldsymbol{w}_r(k)^T\boldsymbol{x}_p \geq 0\} - (\boldsymbol{w}_r(k)^T\boldsymbol{x}_p)I\{\boldsymbol{w}_r(k)^T\boldsymbol{x}_p \geq 0\}] \\
&\quad - \langle\boldsymbol{w}_r(k+1) - \boldsymbol{w}_r(k),\boldsymbol{x}_p\rangle I\{\boldsymbol{w}_r(k)^T\boldsymbol{x}_p \geq 0\} \\
&= 0.
\end{aligned}
\tag{56}
$$

For $r \in S_p^{\perp}$, the Lipschitz continuity of $\sigma''$ implies that

$$\sigma''(\boldsymbol{w}_r(k+1)^T\boldsymbol{x}_p) - \sigma''(\boldsymbol{w}_r(k)^T\boldsymbol{x}_p) - \langle\boldsymbol{w}_r(k+1) - \boldsymbol{w}_r(k),\boldsymbol{x}_p\rangle\sigma'''(\boldsymbol{w}_r(k)^T\boldsymbol{x}_p) = \mathcal{O}(\|\boldsymbol{w}_r(k+1) - \boldsymbol{w}_r(k)\|_2).$$
$$\tag{57}$$

Combining (53), (54) and (55), we can deduce that for $r \in S_p$,

$$|\hat{\chi}_{pr}^2(k)| \lesssim B\|\boldsymbol{w}_r(k+1) - \boldsymbol{w}_r(k)\|_2^2$$

and for $r \in S_p^{\perp}$,

$$|\hat{\chi}_{pr}^2(k)| \lesssim B\|\boldsymbol{w}_r(k+1) - \boldsymbol{w}_r(k)\|_2^2 + B^2\|\boldsymbol{w}_r(k+1) - \boldsymbol{w}_r(k)\|_2.$$

With the estimations for $\hat{\chi}_{pr}^1(k)$ and $\hat{\chi}_{pr}^2(k)$, we have

$$
\begin{aligned}
|I_1^p(k)| &\leq \frac{1}{\sqrt{n_1 m}}\sum_{r=1}^{m}(|\hat{\chi}_{pr}^1(k)| + |\hat{\chi}_{pr}^2(k)|) \\
&\lesssim \frac{1}{\sqrt{n_1 m}}\sum_{r=1}^{m}B\|\boldsymbol{w}_r(k+1) - \boldsymbol{w}_r(k)\|_2^2 + \frac{1}{\sqrt{n_1 m}}\sum_{r\in S_p^{\perp}}B^2\|\boldsymbol{w}_r(k+1) - \boldsymbol{w}_r(k)\|_2.
\end{aligned}
\tag{58}
$$

For $j \in [n_2]$, we consider $I_1^{n_1+j}(k)$, which can be written as follows.

$$
\begin{aligned}
I_1^{n_1+j}(k) &= h_j(k+1) - h_j(k) - \left\langle \boldsymbol{w}(k+1) - \boldsymbol{w}(k), \frac{\partial h_j(k)}{\partial\boldsymbol{w}}\right\rangle \\
&= \sum_{r=1}^{m}\frac{a_r}{\sqrt{n_2 m}}\left[\sigma(\boldsymbol{w}_r(k+1)^T\boldsymbol{y}_j) - \sigma(\boldsymbol{w}_r(k)^T\boldsymbol{y}_j) - \langle\boldsymbol{w}_r(k+1) - \boldsymbol{w}_r(k),\boldsymbol{y}_j\rangle\sigma'(\boldsymbol{w}_r(k)^T\boldsymbol{y}_j)\right].
\end{aligned}
$$

From the mean value theorem, we have that there exists $\zeta(k) \in \mathbb{R}$ such that

$$\sigma(\boldsymbol{w}_r(k+1)^T\boldsymbol{y}_j) - \sigma(\boldsymbol{w}_r(k)^T\boldsymbol{y}_j) = \sigma'(\zeta(k))\langle\boldsymbol{w}_r(k+1) - \boldsymbol{w}_r(k),\boldsymbol{y}_j\rangle$$

and

$$
\begin{aligned}
|\sigma'(\zeta(k)) - \sigma'(\boldsymbol{w}_r(k)^T\boldsymbol{y}_j)| &\leq 2B|\zeta(k) - \boldsymbol{w}_r(k)^T\boldsymbol{y}_j| \\
&\leq 2\sqrt{2}B\|\boldsymbol{w}_r(k+1) - \boldsymbol{w}_r(k)\|_2.
\end{aligned}
$$

Thus,

$$
\begin{aligned}
&|\sigma(\boldsymbol{w}_r(k+1)^T\boldsymbol{y}_j) - \sigma(\boldsymbol{w}_r(k)^T\boldsymbol{y}_j) - \langle\boldsymbol{w}_r(k+1) - \boldsymbol{w}_r(k),\boldsymbol{y}_j\rangle\sigma'(\boldsymbol{w}_r(k)^T\boldsymbol{y}_j)| \\
&= |\sigma'(\zeta(k))\langle\boldsymbol{w}_r(k+1) - \boldsymbol{w}_r(k),\boldsymbol{y}_j\rangle - \sigma(\boldsymbol{w}_r(k)^T\boldsymbol{y}_j) - \langle\boldsymbol{w}_r(k+1) - \boldsymbol{w}_r(k),\boldsymbol{y}_j\rangle\sigma'(\boldsymbol{w}_r(k)^T\boldsymbol{y}_j)| \\
&= |(\sigma'(\zeta(k)) - \sigma'(\boldsymbol{w}_r(k)^T\boldsymbol{y}_j))\langle\boldsymbol{w}_r(k+1) - \boldsymbol{w}_r(k),\boldsymbol{y}_j\rangle| \\
&\lesssim B\|\boldsymbol{w}_r(k+1) - \boldsymbol{w}_r(k)\|_2.
\end{aligned}
$$

Therefore, for $j \in [n_2]$,

$$|I_1^{n_1+j}(k)| \lesssim \frac{B}{\sqrt{n_2 m}} \sum_{r=1}^m \|\boldsymbol{w}_r(k+1) - \boldsymbol{w}_r(k)\|_2^2. \tag{59}$$

From the updating rule of gradient descent, we can deduce that for every $r \in [m]$,

$$\|\boldsymbol{w}_r(k+1) - \boldsymbol{w}_r(k)\|_2 = \left\| -\eta \frac{\partial L(k)}{\partial \boldsymbol{w}_r} \right\|_2 \lesssim \frac{\eta B^2}{\sqrt{m}} \sqrt{L(k)}. \tag{60}$$

Plugging (58) into (57) and (56), we can deduce that

$$
\begin{aligned}
|I_1^p(k)| &\lesssim \frac{B}{\sqrt{n_1 m}} \sum_{r=1}^m \|\boldsymbol{w}_r(k+1) - \boldsymbol{w}_r(k)\|_2^2 + \frac{B^2}{\sqrt{n_1 m}} \sum_{r \in S_p^\perp} \|\boldsymbol{w}_r(k+1) - \boldsymbol{w}_r(k)\|_2 \\
&\lesssim \frac{B}{\sqrt{n_1 m}} \sum_{r=1}^m \frac{\eta^2 B^4}{m} L(k) + \frac{B^2}{\sqrt{n_1 m}} \sum_{r \in S_p^\perp} \frac{\eta B^2}{\sqrt{m}} \sqrt{L(k)} \\
&= \frac{\eta^2 B^5 L(k)}{\sqrt{n_1 m}} + \frac{\eta B^4 \sqrt{L(k)}}{\sqrt{n_1}} \frac{1}{m} \sum_{r=1}^m I\{r \in S_p^\perp\} \\
&\le \frac{\eta^2 B^5 \sqrt{L(0)} \sqrt{L(k)}}{\sqrt{n_1 m}} + \frac{\eta B^4 \sqrt{L(k)}}{\sqrt{n_1}} \frac{1}{m} \sum_{r=1}^m I\{r \in S_p^\perp\}
\end{aligned}
\tag{61}
$$

and

$$
\begin{aligned}
|I_1^{n_1+j}(k)| &\lesssim \frac{B}{\sqrt{n_2 m}} \sum_{r=1}^m \|\boldsymbol{w}_r(k+1) - \boldsymbol{w}_r(k)\|_2^2 \\
&\lesssim \frac{B}{\sqrt{n_2 m}} \sum_{r=1}^m \frac{\eta^2 B^4}{m} L(k) \\
&\le \frac{\eta^2 B^5 \sqrt{L(0)} \sqrt{L(k)}}{\sqrt{n_2 m}}.
\end{aligned}
\tag{62}
$$

Note that

$$P(A_{p,r}) \le \frac{2R}{\sqrt{2\pi}}, \ S_p = \{r \in [m] : I\{A_{p,r}\} = 0\}.$$

Thus, from Bernstein's inequality, we have that with probability at least $1 - e^{-mR}$,

$$\frac{1}{m} \sum_{r=1}^m I\{r \in S_p^\perp\} = \frac{1}{m} \sum_{r=1}^m I\{A_{pr}\} \lesssim 4R.$$

Then the inequality holds for all $p \in [n_1]$ with probability at least $1 - n_1 e^{-mR}$. Plugging this into (59), we can conclude that for every $p \in [n_1]$

$$|I_1^p(k)| \lesssim \frac{\eta^2 B^5 \sqrt{L(0)} \sqrt{L(k)}}{\sqrt{n_1 m}} + \frac{\eta B^4 \sqrt{L(k)}}{\sqrt{n_1}} R. \tag{63}$$

Combining (60) and (61), we have that

$$
\begin{aligned}
\|\boldsymbol{I}_1(k)\|_2 &= \sqrt{\sum_{p=1}^{n_1} |I_1^p(k)|^2 + \sum_{j=1}^{n_2} |I_1^{n_1+j}(k)|^2} \\
&\lesssim \frac{\eta^2 B^5 \sqrt{L(0)} \sqrt{L(k)}}{\sqrt{m}} + \eta B^4 \sqrt{L(k)} R.
\end{aligned}
$$

Plugging this into (50) yields that

$$
\left\| \begin{pmatrix} \boldsymbol{s}(k+1) \\ \boldsymbol{h}(k+1) \end{pmatrix} \right\|_2^2
$$

$$
\leq \left(1 - \frac{\eta\lambda_0}{2}\right)^2 \left\| \begin{pmatrix} \boldsymbol{s}(k) \\ \boldsymbol{h}(k) \end{pmatrix} \right\|_2^2 + \|\boldsymbol{I}_1(k)\|_2^2 + 2\left(1 - \frac{\eta\lambda_0}{2}\right)\left\| \begin{pmatrix} \boldsymbol{s}(k) \\ \boldsymbol{h}(k) \end{pmatrix} \right\|_2 \|\boldsymbol{I}_1(k)\|_2
$$

$$
\leq \left[\left(1 - \frac{\eta\lambda_0}{2}\right)^2 + C^2\left(\frac{\eta^2 B^5 \sqrt{L(0)}}{\sqrt{m}} + \eta B^4 R\right)^2 + 2C\left(\frac{\eta^2 B^5 \sqrt{L(0)}}{\sqrt{m}} + \eta B^4 R\right)\right] \left\| \begin{pmatrix} \boldsymbol{s}(k) \\ \boldsymbol{h}(k) \end{pmatrix} \right\|_2^2
$$

$$
\leq \left(1 - \frac{\eta\lambda_0}{2}\right)\left\| \begin{pmatrix} \boldsymbol{s}(k) \\ \boldsymbol{h}(k) \end{pmatrix} \right\|_2^2,
$$

where $C$ is a universal constant and the last inequality requires that

$$
\frac{\eta^2 B^5 \sqrt{L(0)}}{\sqrt{m}} \lesssim \eta\lambda_0, \ \eta B^4 R \lesssim \eta\lambda_0.
$$

Recall that we also require $CM^2 R < \frac{\lambda_0}{4}$ for $R$ in (8) and

$$
R^{'} = \frac{CB^2 \sqrt{L(0)}}{\sqrt{m}\lambda_0} < R
$$

for $R^{'}$ in (47) to make sure $\|\boldsymbol{H}(k) - \boldsymbol{H}(0)\|_2 \leq \frac{\lambda_0}{4}$.

Finally, with $R = \mathcal{O}(\frac{\lambda_0}{M^2})$ and the upper bound of $L(0)$, $m$ needs to satisfies that

$$
m = \Omega\left(\frac{M^4 B^4 L(0)}{\lambda_0^4}\right) = \Omega\left(\frac{d^8}{\lambda_0^4} \log^6\left(\frac{md}{\delta}\right) \log\left(\frac{n_1 + n_2}{\delta}\right)\right).
$$

$\square$

## C  PROOF OF SECTION 4

### C.1  PROOF OF LEMMA 4.4

*Proof.* Recall that

$$
\boldsymbol{H}(\boldsymbol{w}) = \boldsymbol{D}^T \boldsymbol{D}, \quad \boldsymbol{D} = \left[\frac{\partial s_1(\boldsymbol{w})}{\partial \boldsymbol{w}}, \cdots, \frac{\partial s_{n_1}(\boldsymbol{w})}{\partial \boldsymbol{w}}, \frac{\partial h_1(\boldsymbol{w})}{\partial \boldsymbol{w}}, \cdots, \frac{\partial h_{n_2}(\boldsymbol{w})}{\partial \boldsymbol{w}}\right],
$$

and $\boldsymbol{H}^\infty = \mathbb{E}_{\boldsymbol{w}\sim\mathcal{N}(\boldsymbol{0},\boldsymbol{I})} \boldsymbol{H}(\boldsymbol{w})$.

We denote $\varphi(\boldsymbol{x};\boldsymbol{w}) = \sigma^{'}(\boldsymbol{w}^T\boldsymbol{x})w_0 - \sigma^{''}(\boldsymbol{w}^T\boldsymbol{x})\|\boldsymbol{w}_1\|_2^2$, where $\boldsymbol{w} = (w_0, \boldsymbol{w}_1^T)^T$, $w_0 \in \mathbb{R}$, $\boldsymbol{w}_1 \in \mathbb{R}^d$, then

$$
\frac{\partial s_p(\boldsymbol{w})}{\partial \boldsymbol{w}_r} = \frac{1}{\sqrt{n_1}} \frac{a_r}{\sqrt{m}} \frac{\partial \varphi(\boldsymbol{x}_p;\boldsymbol{w}_r)}{\partial \boldsymbol{w}_r}.
$$

Similarly, we denote $\psi(\boldsymbol{y};\boldsymbol{w}) = \sigma(\boldsymbol{w}^T\boldsymbol{y})$, then

$$
\frac{\partial h_j(\boldsymbol{w})}{\partial \boldsymbol{w}_r} = \frac{1}{\sqrt{n_2}} \frac{a_r}{\sqrt{m}} \frac{\partial \psi(\boldsymbol{y}_j,\boldsymbol{w}_r)}{\partial \boldsymbol{w}_r}.
$$

With the notations, we can deduce that

$$
H_{p,j}^\infty = \begin{cases} \dfrac{1}{n_1}\mathbb{E}_{\boldsymbol{w}\sim\mathcal{N}(\boldsymbol{0},\boldsymbol{I})}\left\langle \dfrac{\partial\varphi(\boldsymbol{x}_p;\boldsymbol{w})}{\partial\boldsymbol{w}}, \dfrac{\partial\varphi(\boldsymbol{x}_j;\boldsymbol{w})}{\partial\boldsymbol{w}}\right\rangle, & 1\leq p\leq n_1, 1\leq j\leq n_1, \\[3mm] \dfrac{1}{\sqrt{n_1 n_2}}\mathbb{E}_{\boldsymbol{w}\sim\mathcal{N}(\boldsymbol{0},\boldsymbol{I})}\left\langle \dfrac{\partial\varphi(\boldsymbol{x}_p;\boldsymbol{w})}{\partial\boldsymbol{w}}, \dfrac{\partial\psi(\boldsymbol{y}_j;\boldsymbol{w})}{\partial\boldsymbol{w}}\right\rangle, & 1\leq p\leq n_1, n_1+1\leq j\leq n_1+n_2, \\[3mm] \dfrac{1}{n_2}\mathbb{E}_{\boldsymbol{w}\sim\mathcal{N}(\boldsymbol{0},\boldsymbol{I})}\left\langle \dfrac{\partial\psi(\boldsymbol{y}_p;\boldsymbol{w})}{\partial\boldsymbol{w}}, \dfrac{\partial\psi(\boldsymbol{y}_j;\boldsymbol{w})}{\partial\boldsymbol{w}}\right\rangle, & n_1+1\leq p\leq n_1+n_2, n_1+1\leq j\leq n_1+n_2, \end{cases}
$$

where $H_{p,j}^\infty$ is the $(p, j)$-th entry of $\boldsymbol{H}^\infty$.

The proof of this lemma requires tools from functional analysis. Let $\mathcal{H}$ be a Hilbert space of integrable $(d+2)$-dimensional vector fields on $\mathbb{R}^{d+2}$, i.e., $f \in \mathcal{H}$ if $\mathbb{E}_{\boldsymbol{w} \sim \mathcal{N}(\boldsymbol{0}, \boldsymbol{I})}[\|f(\boldsymbol{w})\|_2^2] < \infty$. The inner product for any two elements $f, g$ in $\mathcal{H}$ is $\mathbb{E}_{\boldsymbol{w} \sim \mathcal{N}(\boldsymbol{0}, \boldsymbol{I})}[\langle f(\boldsymbol{w}), g(\boldsymbol{w}) \rangle]$. Thus, proving $\boldsymbol{H}^\infty$ is strictly positive definite is equivalent to show that

$$\frac{\partial \varphi(\boldsymbol{x}_1; \boldsymbol{w})}{\partial \boldsymbol{w}}, \cdots, \frac{\partial \varphi(\boldsymbol{x}_{n_1}; \boldsymbol{w})}{\partial \boldsymbol{w}}, \frac{\partial \psi(\boldsymbol{y}_1; \boldsymbol{w})}{\partial \boldsymbol{w}}, \cdots, \frac{\partial \psi(\boldsymbol{y}_{n_2}; \boldsymbol{w})}{\partial \boldsymbol{w}} \in \mathcal{H}$$

are linearly independent. Suppose that there are $\alpha_1, \cdots, \alpha_{n_1}, \beta_1, \cdots, \beta_{n_2} \in \mathbb{R}$ such that

$$\alpha_1 \frac{\partial \varphi(\boldsymbol{x}_1; \boldsymbol{w})}{\partial \boldsymbol{w}} + \cdots + \alpha_{n_1} \frac{\partial \varphi(\boldsymbol{x}_{n_1}; \boldsymbol{w})}{\partial \boldsymbol{w}} + \beta_1 \frac{\partial \psi(\boldsymbol{y}_1; \boldsymbol{w})}{\partial \boldsymbol{w}} + \cdots + \beta_{n_2} \frac{\partial \psi(\boldsymbol{y}_{n_2}; \boldsymbol{w})}{\partial \boldsymbol{w}} = 0 \ in \ \mathcal{H}.$$

This implies that

$$\alpha_1 \frac{\partial \varphi(\boldsymbol{x}_1; \boldsymbol{w})}{\partial \boldsymbol{w}} + \cdots + \alpha_{n_1} \frac{\partial \varphi(\boldsymbol{x}_{n_1}; \boldsymbol{w})}{\partial \boldsymbol{w}} + \beta_1 \frac{\partial \psi(\boldsymbol{y}_1; \boldsymbol{w})}{\partial \boldsymbol{w}} + \cdots + \beta_{n_2} \frac{\partial \psi(\boldsymbol{y}_{n_2}; \boldsymbol{w})}{\partial \boldsymbol{w}} = 0 \quad (64)$$

holds for all $\boldsymbol{w} \in \mathbb{R}^{d+1}$, as $\sigma(\cdot)$ is smooth.

First, we compute the derivatives of $\varphi$ and $\psi$ with respect to $\boldsymbol{w}$. For the $k$-th derivative of $\psi(\boldsymbol{y}; \boldsymbol{w})$ with respect to $\boldsymbol{w}$, we have

$$\frac{\partial^k \psi(\boldsymbol{y}; \boldsymbol{w})}{\partial \boldsymbol{w}^k} = \sigma^{(k)}(\boldsymbol{w}^T \boldsymbol{y}) \boldsymbol{y}^{\otimes(k)},$$

where $\otimes$ denotes the tensor product.

For $\varphi(\boldsymbol{x}; \boldsymbol{w})$, let $\varphi_0(\boldsymbol{x}; \boldsymbol{w}) = \sigma'(\boldsymbol{w}^T \boldsymbol{x}) w_0$ and $\varphi_i(\boldsymbol{x}; \boldsymbol{w}) = \sigma''(\boldsymbol{w}^T \boldsymbol{x}) w_i^2$ for $1 \le i \le d$. Then

$$\varphi(\boldsymbol{x}; \boldsymbol{w}) = \varphi_0(\boldsymbol{x}; \boldsymbol{w}) - \sum_{i=1}^d \varphi_i(\boldsymbol{x}; \boldsymbol{w}).$$

For the $k$-th derivative of $\varphi_0(\boldsymbol{x}; \boldsymbol{w})$ with respect to $\boldsymbol{w}$, analogous to the Leibniz rule for the $k$-th derivative of the product of two scalar functions, we have

$$\begin{aligned}
\frac{\partial^k \varphi_0(\boldsymbol{x}; \boldsymbol{w})}{\partial \boldsymbol{w}^k} &= \sigma^{(k+1)}(\boldsymbol{w}^T \boldsymbol{x}) w_0 \boldsymbol{x}^{\otimes(k)} + \sum_{i=1}^k \boldsymbol{x}^{\otimes(i-1)} \otimes \boldsymbol{e}_0 \otimes \boldsymbol{x}^{\otimes(k-i)} \sigma^{(k)}(\boldsymbol{w}^T \boldsymbol{x}) \\
&= \sigma^{(k+1)}(\boldsymbol{w}^T \boldsymbol{x}) w_0 \boldsymbol{x}^{\otimes(k)} + \sigma^{(k)}(\boldsymbol{w}^T \boldsymbol{x}) \sum_{i=1}^k \boldsymbol{e}_0^{(i)} \otimes \boldsymbol{x}^{\otimes(k)},
\end{aligned} \quad (65)$$

where $\boldsymbol{e}_0 = (1, 0, \cdots, 0)^T \in \mathbb{R}^{d+1}$ and in the second equality, $\boldsymbol{e}_0^{(i)}$ denotes that $\boldsymbol{e}_0$ is placed at the $i$-th position.

Similarly, for $\varphi_i(\boldsymbol{x}; \boldsymbol{w})$ where $1 \le i \le d$, taking $i = 1$ as example, we have

$$\begin{aligned}
\frac{\partial^k \varphi_1(\boldsymbol{x}; \boldsymbol{w})}{\partial \boldsymbol{w}^k} &= \sigma^{(k+2)}(\boldsymbol{w}^T \boldsymbol{x}) w_1^2 \boldsymbol{x}^{\otimes(k)} + 2k w_1 \sigma^{(k+1)}(\boldsymbol{w}^T \boldsymbol{x}) \sum_{i=1}^k \boldsymbol{e}_1^{(i)} \otimes \boldsymbol{x}^{\otimes(k-1)} \\
&\quad + k(k-1) \sigma^{(k)}(\boldsymbol{w}^T \boldsymbol{x}) \sum_{1 \le i < j \le k} \boldsymbol{e}_1^{(i)} \otimes \boldsymbol{e}_1^{(j)} \otimes \boldsymbol{x}^{\otimes(k-2)},
\end{aligned} \quad (66)$$

where $\boldsymbol{e}_i \in \mathbb{R}^{d+1}$ is a vector with the $(i+1)$-th component equal to 1 and all other components equal to 0, and $\boldsymbol{e}_1^{(i)}$ indicates that $\boldsymbol{e}_1$ is placed at the $i$-th position.

By combining the derivatives of $\varphi_0(\boldsymbol{x}; \boldsymbol{w}), \cdots, \varphi_d(\boldsymbol{x}; \boldsymbol{w})$ from equations (65) and (66), we can compute the $k$-th derivative of $w_0 \sigma'(\boldsymbol{w}^T \boldsymbol{x}) - \sum_{i=1}^{d} w_i^2 \sigma''(\boldsymbol{w}^T \boldsymbol{x})$ as follows:

$$
\begin{aligned}
\frac{\partial^k \varphi(\boldsymbol{x}; \boldsymbol{w})}{\partial \boldsymbol{w}^k} &= \frac{\partial^k \varphi_0(x; \boldsymbol{w})}{\partial \boldsymbol{w}^k} - \sum_{i=1}^{d} \frac{\partial^k \varphi_i(\boldsymbol{x}; \boldsymbol{w})}{\partial \boldsymbol{w}^k} \\
&= w_0 \sigma^{(k+1)}(\boldsymbol{w}^T \boldsymbol{x}) \boldsymbol{x}^{\otimes(k)} + \sigma^{(k)}(\boldsymbol{w}^T \boldsymbol{x}) \sum_{i=1}^{k} \boldsymbol{e}_0^{(i)} \otimes \boldsymbol{x}^{\otimes(k)} \\
&\quad - \sum_{t=1}^{d} \left[ w_t^2 \sigma^{(k+2)}(\boldsymbol{w}^T \boldsymbol{x}) \boldsymbol{x}^{\otimes(k)} + 2k w_t \sigma^{(k+1)}(\boldsymbol{w}^T \boldsymbol{x}) \sum_{i=1}^{k} \boldsymbol{e}_t^{(i)} \otimes \boldsymbol{x}^{\otimes(k-1)} \right. \\
&\quad \left. + k(k-1) \sigma^{(k)}(\boldsymbol{w}^T \boldsymbol{x}) \sum_{1 \le i < j \le k} \boldsymbol{e}_t^{(i)} \otimes \boldsymbol{e}_t^{(j)} \otimes \boldsymbol{x}^{\otimes(k-2)} \right].
\end{aligned}
\tag{67}
$$

Note that when any two points in $\{\boldsymbol{x}_1, \cdots, \boldsymbol{x}_{n_1}, \boldsymbol{y}_1, \cdots, \boldsymbol{y}_{n_2}\}$ are non-parallel, the tensors

$$
\boldsymbol{x}_1^{\otimes(n_1+n_2)}, \cdots, \boldsymbol{x}_{n_1}^{\otimes(n_1+n_2)}, \boldsymbol{y}_1^{\otimes(n_1+n_2)}, \cdots, \boldsymbol{y}_{n_2}^{\otimes(n_1+n_2)}
$$

are linearly independent (see Lemma G.6 in Du et al. (2019)). This observation motivates us to take the $(k-1)$-th derivative of both sides of equation (64) with respect to $\boldsymbol{w}$, yielding

$$
\alpha_1 \frac{\partial^k \varphi(\boldsymbol{x}_1; \boldsymbol{w})}{\partial \boldsymbol{w}^k} + \cdots + \alpha_{n_1} \frac{\partial^k \varphi(\boldsymbol{x}_{n_1}; \boldsymbol{w})}{\partial \boldsymbol{w}^k} + \beta_1 \frac{\partial^k \psi(\boldsymbol{y}_1; \boldsymbol{w})}{\partial \boldsymbol{w}^k} + \cdots + \beta_{n_2} \frac{\partial^k \psi(\boldsymbol{y}_{n_2}; \boldsymbol{w})}{\partial \boldsymbol{w}^k} = 0. \tag{68}
$$

Since this equation holds for all $\boldsymbol{w} \in \mathbb{R}^{d+1}$, we specifically consider $\boldsymbol{w} = (w_0, \boldsymbol{0}_d)$, where $w_0$ is to be determined. Under this condition, equation (68) becomes

$$
w_0 \sum_{p=1}^{n_1} \alpha_p \left[ \sigma^{(k+1)}(w_0 x_p^0) \boldsymbol{x}_p^{\otimes(k)} \right] + \sum_{p=1}^{n_1} \alpha_p \left[ \sigma^{(k)}(w_0 x_p^0) \boldsymbol{z}_p \right] + \sum_{j=1}^{n_2} \beta_j \left[ \sigma^{(k)}(w_0 y_j^0) \boldsymbol{y}_j^{\otimes(k)} \right] = 0,
\tag{69}
$$

where the tensor $\boldsymbol{z}_p$ is defined as

$$
\boldsymbol{z}_p = \sum_{i=1}^{k} \boldsymbol{e}_0^{(i)} \otimes \boldsymbol{x}_p^{\otimes(k)} - k(k-1) \sum_{t=1}^{d} \sum_{1 \le i < j \le k} \boldsymbol{e}_t^{(i)} \otimes \boldsymbol{e}_t^{(j)} \otimes \boldsymbol{x}_p^{\otimes(k-2)}. \tag{70}
$$

By assumption, for any positive integer $n \ge 0$ we have $\lim_{x \to +\infty} \frac{\sigma^{(n)}(x)}{\phi(x)} = c_n \ne 0$. We first consider the case where all input components satisfy $x_1^0 = \cdots = x_{n_1}^0 = y_1^0 = \cdots = y_{n_2}^0 = a > 0$. Under this condition, equation (69) simplifies to:

$$
w_0 \sigma^{(k+1)}(w_0 a) \left[ \sum_{p=1}^{n_1} \alpha_p \boldsymbol{x}_p^{\otimes(k)} \right] + \sigma^{(k)}(w_0 a) \left[ \sum_{p=1}^{n_1} \alpha_p \boldsymbol{z}_p \right] + \sigma^{(k)}(w_0 a) \left[ \sum_{j=1}^{n_2} \beta_j \boldsymbol{y}_j^{\otimes(k)} \right] = 0. \tag{71}
$$

Dividing both sides of equation (71) by $\phi(w_0 a)$ yields

$$
w_0 \frac{\sigma^{(k+1)}(w_0 a)}{\phi(w_0 a)} \left[ \sum_{p=1}^{n_1} \alpha_p \boldsymbol{x}_p^{\otimes(k)} \right] + \frac{\sigma^{(k)}(w_0 a)}{\phi(w_0 a)} \left[ \sum_{p=1}^{n_1} \alpha_p \boldsymbol{z}_p \right] + \frac{\sigma^{(k)}(w_0 a)}{\phi(w_0 a)} \left[ \sum_{j=1}^{n_2} \beta_j \boldsymbol{y}_j^{\otimes(k)} \right] = 0. \tag{72}
$$

Now taking the limit as $w_0$ tends to positive infinity, we observe that $\frac{\sigma^{(k)}(w_0 a)}{\phi(w_0 a)}$ converges to a non-zero constant, while $w_0 \frac{\sigma^{(k+1)}(w_0 a)}{\phi(w_0 a)}$ diverges to infinity. This asymptotic behavior leads to the following conclusions:

$$
\sum_{p=1}^{n_1} \alpha_p \boldsymbol{x}_p^{\otimes(k)} = 0, \quad \sum_{p=1}^{n_1} \alpha_p \boldsymbol{z}_p + \sum_{j=1}^{n_2} \beta_j \boldsymbol{y}_j^{\otimes(k)} = 0. \tag{73}
$$

By the linear independence of the tensor products (established earlier), we can deduce that $\alpha_p = 0$ for all $p = 1, \cdots, n_1$, which subsequently implies $\sum_{j=1}^{n_2} \beta_j \boldsymbol{y}_j^{\otimes(k)} = 0$ and thus $\beta_j = 0$ for all $j = 1, \cdots, n_2$.

When our previous assumption does not hold—that is, when $x_1^0, \cdots, x_{n_1}^0, y_1^0, \cdots, y_{n_2}^0$ are not necessarily all equal—we proceed as follows:

Case 1: All elements are strictly positive.

Let $b = \min\{x_1^0, \cdots, x_{n_1}^0, y_1^0, \cdots, y_{n_2}^0\}$. Dividing both sides of (68) by $\phi(w_0 b)$, we observe that for any $x > b$,

$$\lim_{w_0 \to +\infty} \frac{\sigma^{(n)}(w_0 x)}{\phi(w_0 b)} = 0.$$

Thus, the problem reduces to the previously considered case where all inputs are equal (to $b$). Due to linear independence, the coefficients $\alpha_p$ and $\beta_j$ corresponding to the minimal $b$ must vanish. Repeating this process iteratively, we conclude that all $\alpha_p$ and $\beta_j$ must be zero.

Case 2: Some elements are zero.

Since the $\boldsymbol{x}_p$ are interior points (and thus non-zero), any zero-valued inputs must correspond to boundary or initial conditions. Let us assume without loss of generality that $y_1^0, \cdots, y_{n_2}^0$ are all zero.

1. If $\sigma^{(k)}(0) = 0$, , then following our previous method, we can conclude that the coefficients corresponding to non-zero inputs vanish. Returning to equation (68), we have

$$\sum_{j=1}^{n_2} \beta_j \sigma^{(k)}(\boldsymbol{w}^T \boldsymbol{y}_j) \boldsymbol{y}_j^{\otimes(k)} = 0. \tag{74}$$

By the independence of $\boldsymbol{y}_1, \cdots, \boldsymbol{y}_{n_2}$, we can deduce that $\beta_j \sigma^{(k)}(\boldsymbol{w}^T \boldsymbol{y}_j) = 0$ holds for all $j \in [n_2]$. From the assumption, we can select $\boldsymbol{w}$ such that $\sigma^{(k)}(\boldsymbol{w}^T \boldsymbol{y}_j) \neq 0$, and consequently, $\beta_j = 0$ holds for all $j \in [n_2]$.

2. If $\sigma^{(k)}(0) \neq 0$, let $b$ be the smallest strictly positive value among $x_1^0, \cdots, x_{n_1}^0, y_1^0, \cdots, y_{n_2}^0$. Divide (71) by $\phi(w_0 b/2)$. Since all other positive terms decay to zero as $w_0 \to +\infty$, we obtain:

$$\lim_{w_0 \to \infty} \sum_{j=1}^{n_2} \beta_j \sigma^{(k)}(0) \boldsymbol{y}_j^{\otimes(k)} / \phi(w_0 b/2) = 0. \tag{75}$$

This implies that $\sum_{j=1}^{n_2} \beta_j \boldsymbol{y}_j^{\otimes(k)} = 0$. By linear independence, all $\beta_j = 0$. $\qquad \square$

**Remark C.1.** The key point in the proof lies in the fact that the order of the PDE in the interior is higher than that of the initial and boundary conditions, allowing for a natural extension to broader classes of PDEs. For general PDEs, we may focus solely on the interior and boundary, assuming the interior is of second order and the boundary is of first order. Suppose the second-order interior term is taken at $x_0$, i.e., it has the form $\frac{\partial^2 u}{\partial x_0^2}$, and the first-order boundary term is also taken at $x_0$. Since we can translate the coordinates, without loss of generality, we can assume that all $x_0$-components are positive.

For the interior, taking the $k$-th derivative of $w_0^2 \sigma^{(2)}(\boldsymbol{w}^T \boldsymbol{x})$ yields that

$$w_0^2 \sigma^{(k+2)}(\boldsymbol{w}^T \boldsymbol{x}) \boldsymbol{x}^{\otimes(k)} + 2k w_0 \sigma^{(k+1)}(\boldsymbol{w}^T \boldsymbol{x}) \sum_{i=1}^{k} \boldsymbol{e}_1^{(i)} \otimes \boldsymbol{x}^{\otimes(k-1)}$$

$$+ k(k-1) \sigma^{(k)}(\boldsymbol{w}^T \boldsymbol{x}) \sum_{1 \leq i < j \leq k} \boldsymbol{e}_0^{(i)} \otimes \boldsymbol{e}_0^{(j)} \otimes \boldsymbol{x}^{\otimes(k-2)}.$$

For the boundary, taking the $k$-th derivative of $w_0 \sigma^{(1)}(\boldsymbol{w}^T \boldsymbol{x})$ yields that

$$\sigma^{(k+1)}(\boldsymbol{w}^T \boldsymbol{x}) w_0 \boldsymbol{x}^{\otimes(k)} + \sigma^{(k)}(\boldsymbol{w}^T \boldsymbol{x}) \sum_{i=1}^{k} \boldsymbol{e}_0^{(i)} \otimes \boldsymbol{x}^{\otimes(k)}.$$

As before, we set $\boldsymbol{w} = (w_0, \boldsymbol{0})$. Then, the equation (69) becomes

$$w_0^2 \sum_{p=1}^{n_1} \alpha_p \left[ \sigma^{(k+1)}(w_0 x_p^0) \boldsymbol{x}_p^{\otimes(k)} \right] + w_0 \sum_{p=1}^{n_1} \alpha_p \left[ \sigma^{(k)}(w_0 x_p^0) \boldsymbol{z}_p^0 \right] + \sum_{p=1}^{n_1} \alpha_p \left[ \sigma^{(k)}(w_0 x_p^0) \boldsymbol{z}_p^1 \right]$$
$$+ w_0 \sum_{j=1}^{n_2} \beta_j \left[ \sigma^{(k)}(w_0 y_j^0) \boldsymbol{y}_j^{\otimes(k)} \right] + \sum_{j=1}^{n_2} \beta_j \left[ \sigma^{(k)}(w_0 y_j^0) \boldsymbol{z}_j^2 \right] = 0,$$

where $\boldsymbol{z}_p^0, \boldsymbol{z}_p^1, \boldsymbol{z}_j^2$ are tensors of similar form $\boldsymbol{z}_p$ in (70), whose explicit definitions are omitted for simplicity. Dividing both sides by $w_0 \phi(w_0 x_p^0)$ reduces it to the form considered earlier. We can therefore conclude that the Gram matrix is strictly positive definite. Indeed, since the orders of the interior and boundary terms in the partial differential equation differ, we can relax the conditions in Lemma 4.4 to simply requiring that no two samples in $\{\boldsymbol{x}_p\}_{p=1}^{n_1}$ are parallel and no two samples in $\{\boldsymbol{y}_j\}_{j=1}^{n_2}$ are parallel. In brief, we can set $k = n_1 - 1$ and $k = n_2 - 1$ in equation (68), and then use the method described above.

**Remark C.2.** For the activation functions $\sin(x)$ and $\cos(x)$, in equation (69), we may assume that $\sigma^{(k+1)}(x) = \sin(x), \sigma^{(k)}(x) = -\cos(x)$. Dividing both sides of equation (69) by $w_0$ and letting $w_0 \to +\infty$, we can obtain

$$\lim_{w_0 \to +\infty} \sum_{p=1}^{n_1} \alpha_p \left[ \sin(w_0 x_p^0) \boldsymbol{x}_p^{\otimes(k)} \right] = 0.$$

We express the general form of the components of the tensor above as

$$\sum_{p=1}^{n_1} \alpha_p c_p \sin(w_0 x_p^0) = \sum_{i=1}^{n} a_i \sin(w_0 b_i),$$

where the $b_i > 0$ are distinct and $c_p$ denotes the components of the tensor $\boldsymbol{x}_p^{\otimes(k)}$. For simplicity, we denote $w_0$ as $x$. To prove that $\sum_{p=1}^{n_1} \alpha_p \boldsymbol{x}_p^{\otimes(k)} = 0$, we need to show that any component of this tensor is zero, i.e., its general form satisfies $\sum_{p=1}^{n_1} \alpha_p c_p = 0$. This is equivalent to proving $\sum_{i=1}^{n} a_i = 0$.

Let $f(x) = \sum_{i=1}^{n} a_i \sin(b_i x)$, note that dividing both sides of equation (69) by $w_0$ yields that $f(x) = \mathcal{O}(1/x)$. Thus, we can consider the average energy of $f^2(x)$ over the interval $[T, T+L]$, i.e.,

$$\frac{1}{L} \int_T^{T+L} f^2(x) dx.$$

Expanding this, we obtain

$$\frac{1}{L} \int_T^{T+L} \left( \sum_{i=1}^{n} a_i \sin(b_i x) \right)^2 dx$$
$$= \frac{1}{L} \int_T^{T+L} \left[ \sum_{i=1}^{n} a_i^2 \sin^2(b_i x) + \sum_{i \neq j} a_i a_j \sin(b_i x) \sin(b_j x) \right] dx$$
$$= \frac{1}{2} \sum_{i=1}^{n} a_i^2 - \frac{1}{L} \sum_{i=1}^{n} \frac{\sin(2b_i(T+L)) - \sin(2b_i T)}{4 b_i}$$
$$+ \frac{1}{L} \sum_{i \neq j} a_i a_j \frac{\sin((b_i - b_j)(T+L)) - \sin((b_i - b_j)T)}{2(b_i - b_j)}$$
$$- \frac{1}{L} \sum_{i \neq j} a_i a_j \frac{\sin((b_i + b_j)(T+L)) - \sin((b_i + b_j)T)}{2(b_i + b_j)}.$$

Taking the limits $L \to +\infty$ and $T \to +\infty$ in the above equation, the right-hand side tends to $\frac{1}{2} \sum_{i=1}^{n} a_i^2$.

Regarding the left-hand side, recall that $f(x) = \mathcal{O}(1/x)$, thus for any $\epsilon > 0$, there exists $T_0$ such that for all $x > T_0$, $|f(x)| < \epsilon$. Therefore, for $T > T_0$ and any $L$, we have

$$\frac{1}{L} \int_T^{T+L} f^2(x)dx \le \epsilon^2.$$

By the arbitrariness of $\epsilon$, we can deduce that

$$\lim_{T,L \to +\infty} \frac{1}{L} \int_T^{T+L} f^2(x)dx = 0.$$

Hence, we can deduce that $\frac{1}{2} \sum_{i=1}^{n} a_i^2 = 0$, which implies that $\sum_{i=1}^{n} a_i = 0$, i.e., $\sum_{p=1}^{n_1} \alpha_p c_p = 0$. Finally, we obtain

$$\sum_{p=1}^{n_1} \alpha_p \boldsymbol{x}_p^{\otimes(k)} = 0.$$

Applying the same approach as before, we conclude that $\alpha_p = 0$ for all $p \in [n_1]$ and $\beta_j = 0$ for all $j \in [n_2]$.

## C.2 PROOF OF LEMMA 4.6

*Proof.* Recall that

$$\frac{\partial s_p(\boldsymbol{w})}{\partial \boldsymbol{w}_r} = \frac{a_r}{\sqrt{n_1 m}} \left[ \sigma''(\boldsymbol{w}_r^T \boldsymbol{x}_p) w_{r0} \boldsymbol{x}_p + \sigma'(\boldsymbol{w}_r^T \boldsymbol{x}_p) \begin{pmatrix} 1 \\ \boldsymbol{0}_{d+1} \end{pmatrix} - \sigma'''(\boldsymbol{w}_r^T \boldsymbol{x}_p) \|\boldsymbol{w}_{r1}\|_2^2 \boldsymbol{x}_p \right. $$
$$\left. -2\sigma''(\boldsymbol{w}_r^T \boldsymbol{x}_p) \begin{pmatrix} 0 \\ \boldsymbol{w}_{r1} \end{pmatrix} \right]$$

and

$$\frac{\partial h_j(\boldsymbol{w})}{\partial \boldsymbol{w}_r} = \frac{a_r}{\sqrt{n_2 m}} \sigma'(\boldsymbol{w}_r^T \boldsymbol{y}_j) \boldsymbol{y}_j.$$

(1) When $\sigma(\cdot)$ is the ReLU$^3$ activation function.

From the form of $\frac{\partial s_p(\boldsymbol{w})}{\partial \boldsymbol{w}_r}$, we can deduce that

$$\left\| \frac{\partial s_p(\boldsymbol{w})}{\partial \boldsymbol{w}_r} - \frac{\partial s_p(0)}{\partial \boldsymbol{w}_r} \right\|_2$$
$$\lesssim \frac{1}{\sqrt{n_1 m}} \left[ R(\|\boldsymbol{w}_r(0)\|_2 + 1) + |I\{\boldsymbol{w}_r^T \boldsymbol{x}_p \ge 0\} - I\{\boldsymbol{w}_r(0)^T \boldsymbol{x}_p \ge 0\}|(\|\boldsymbol{w}_r(0)\|_2^2 + 1)] \quad (76)$$
$$\le \frac{1}{\sqrt{n_1 m}} \left[ R(\|\boldsymbol{w}_r(0)\|_2 + 1) + I\{A_{pr}\}(\|\boldsymbol{w}_r(0)\|_2^2 + 1) \right],$$

where the second inequality follows from the fact $\|\boldsymbol{w} - \boldsymbol{w}_r(0)\|_2 < R \le 1$ and the definition of $A_{pr}$ in (28).

Similarly, we have that

$$\left\| \frac{\partial h_j(\boldsymbol{w})}{\partial \boldsymbol{w}_r} - \frac{\partial h_j(0)}{\partial \boldsymbol{w}_r} \right\|_2 \lesssim \frac{1}{\sqrt{n_2 m}} R(\|\boldsymbol{w}_r(0)\|_2 + 1). \quad (77)$$

Combining the above equations, we can deduce that

$$\|\boldsymbol{J}(\boldsymbol{w}) - \boldsymbol{J}(0)\|_2^2$$
$$\leq \|\boldsymbol{J}(\boldsymbol{w}) - \boldsymbol{J}(0)\|_F^2$$
$$= \sum_{i=1}^{n_1+n_2} \|\boldsymbol{J}_i(\boldsymbol{w}) - \boldsymbol{J}_i(0)\|_2^2$$
$$= \sum_{r=1}^{m} \left( \sum_{p=1}^{n_1} \left\| \frac{\partial s_p(\boldsymbol{w})}{\partial \boldsymbol{w}_r} - \frac{\partial s_p(0)}{\partial \boldsymbol{w}_r} \right\|_2^2 + \sum_{j=1}^{n_2} \left\| \frac{\partial h_j(\boldsymbol{w})}{\partial \boldsymbol{w}_r} - \frac{\partial h_j(0)}{\partial \boldsymbol{w}_r} \right\|_2^2 \right)$$
$$\lesssim \sum_{r=1}^{m} \left( \sum_{p=1}^{n_1} \frac{1}{n_1 m} \left( R(\|\boldsymbol{w}_r(0)\|_2 + 1) + I\{A_{pr}\}(\|\boldsymbol{w}_r(0)\|_2^2 + 1) \right)^2 + \sum_{j=1}^{n_2} \frac{1}{n_2 m} (R\|\boldsymbol{w}_r(0)\|_2 + R)^2 \right)$$
$$\lesssim \frac{R^2}{m} \sum_{r=1}^{m} (\|\boldsymbol{w}_r(0)\|_2^2 + 1) + \frac{1}{n_1 m} \sum_{p=1}^{n_1} \sum_{r=1}^{m} I\{A_{pr}\}(\|\boldsymbol{w}_r(0)\|_2^4 + 1)$$
$$= \frac{R^2}{m} \sum_{r=1}^{m} (\|\boldsymbol{w}_r(0)\|_2^2 + 1)$$
$$+ \frac{1}{n_1 m} \sum_{p=1}^{n_1} \sum_{r=1}^{m} I\{A_{pr}\} \left( \|\boldsymbol{w}_r(0)\|_2^4 I\{\|\boldsymbol{w}_r(0)\|_2^2 \leq M\} + \|\boldsymbol{w}_r(0)\|_2^4 I\{\|\boldsymbol{w}_r(0)\|_2^2 > M\} + 1 \right)$$
$$\lesssim \frac{R^2}{m} \sum_{r=1}^{m} (\|\boldsymbol{w}_r(0)\|_2^2 + 1) + \frac{M^2}{n_1 m} \sum_{p=1}^{n_1} \sum_{r=1}^{m} I\{A_{pr}\} + \frac{1}{m} \sum_{r=1}^{m} \|\boldsymbol{w}_r(0)\|_2^4 I\{\|\boldsymbol{w}_r(0)\|_2^2 > M\},$$

where $M = 2(d+2)\log(2m(d+2)/\delta)$. Note that from (39), we have

$$P\left( \exists r \in [m], \|\boldsymbol{w}_r(0)\|_2^2 \geq 2(d+2)\log\left( \frac{2m(d+2)}{\delta} \right) \right) \leq \delta.$$

On the other hand, applying Bernstein's inequality yields that with probability at least $1 - n_1 e^{-mR}$,

$$\frac{1}{m} \sum_{r=1}^{m} I\{A_{pr}\} < 4R$$

holds for all $p \in [n_1]$.

Therefore, we have that

$$\|\boldsymbol{J}(\boldsymbol{w}) - \boldsymbol{J}(0)\|_2^2 \lesssim MR^2 + R^2 + M^2 R \lesssim M^2 R$$

holds with probability at least $1 - \delta - n_1 e^{-mR}$.

(2) Note that when $\sigma$ satisfies Assumption 4.3, $\sigma', \sigma''$ and $\sigma'''$ are all Lipschitz continuous and bounded. Thus, we can obtain that

$$\left\| \frac{\partial s_p(\boldsymbol{w})}{\partial \boldsymbol{w}_r} - \frac{\partial s_p(0)}{\partial \boldsymbol{w}_r} \right\|_2 \lesssim \frac{1}{\sqrt{n_1 m}} R(\|\boldsymbol{w}_r(0)\|_2^2 + \|\boldsymbol{w}_r(0)\|_2 + 1) \lesssim \frac{1}{\sqrt{n_1 m}} R(\|\boldsymbol{w}_r(0)\|_2^2 + 1), \quad (78)$$

where the second inequality is from Young's inequality.

Similarly, we have

$$\left\| \frac{\partial h_j(\boldsymbol{w})}{\partial \boldsymbol{w}_r} - \frac{\partial h_j(0)}{\partial \boldsymbol{w}_r} \right\|_2 \lesssim \frac{1}{\sqrt{n_2 m}} R(\|\boldsymbol{w}_r(0)\|_2 + 1). \quad (79)$$

Combining the above equations yields that

$$\|\boldsymbol{J}(\boldsymbol{w}) - \boldsymbol{J}(0)\|_2^2$$

$$\leq \sum_{r=1}^{m} \left( \sum_{p=1}^{n_1} \left\| \frac{\partial s_p(\boldsymbol{w})}{\partial \boldsymbol{w}_r} - \frac{\partial s_p(0)}{\partial \boldsymbol{w}_r} \right\|_2^2 + \sum_{j=1}^{n_2} \left\| \frac{\partial h_j(\boldsymbol{w})}{\partial \boldsymbol{w}_r} - \frac{\partial h_j(0)}{\partial \boldsymbol{w}_r} \right\|_2^2 \right)$$

$$\lesssim \sum_{r=1}^{m} \left( \sum_{p=1}^{n_1} \frac{1}{n_1 m}(R\|\boldsymbol{w}_r(0)\|_2^2 + R)^2 + \sum_{j=1}^{n_2} \frac{1}{n_2 m}(R\|\boldsymbol{w}_r(0)\|_2 + R)^2 \right)$$

$$\lesssim \frac{R^2}{m} \sum_{r=1}^{m} (\|\boldsymbol{w}_r(0)\|_2^4 + 1)$$

$$\lesssim R^2 \left[ d^2 + \frac{d^2}{\sqrt{m}} \sqrt{\log\left(\frac{1}{\delta}\right)} + \frac{d^2}{m} \left( \log\left(\frac{1}{\delta}\right) \right)^2 \right],$$

where the last inequality follows from the fact that $\left\| \|\boldsymbol{w}_r(0)\|_2^4 \right\|_{\psi_{\frac{1}{2}}} \lesssim d^2$ and Lemma D.1. $\qquad\square$

## C.3 PROOF OF THEOREM 4.7

For the sake of completeness in the proof, we restate Condition 1 and Corollary 4.11 from the main text, and label them as Condition 3 and Corollary C.3, respectively.

**Condition 3.** At the $t$-th iteration, we have $\|\boldsymbol{w}_r(t)\|_2 \leq B$ and

$$\|\boldsymbol{w}_r(t) - \boldsymbol{w}_r(0)\|_2 \leq \frac{CB^2\sqrt{L(0)}}{\sqrt{m}\lambda_0} := R'$$

for all $r \in [m]$, where $C$ is a universal constant and $B = \sqrt{2(d+2)\log\left(\frac{2m(d+2)}{\delta}\right)} + 1$.

**Corollary C.3.** *If Condition 3 holds for $t = 0, \cdots, k$ and $R' \leq R$ and $R'' \lesssim \sqrt{1-\eta}\sqrt{\lambda_0}$, then*

$$L(t) \leq (1-\eta)^t L(0),$$

*holds for $t = 0, \cdots, k$, where $R$ is the constant in Lemma 4.5 and $R'' = CM\sqrt{R}$ in (16) when $\sigma$ is the ReLU$^3$ activation function, $R'' = CdR$ in (18) when $\sigma$ satisfies Assumption 4.3.*

Thanks to Corollary C.3, it is sufficient to prove that Condition 3 also holds for $t = k + 1$. For readability, we defer the proof of Corollary C.3 to the end of this section. In the following, we are going to show that the Condition 3 also holds for $t = k + 1$, thus combining Condition 3 and Corollary C.3 leads to Theorem 4.7.

*Sketch Proof of Theorem 4.7.* First, let $\boldsymbol{u}(t) = \begin{pmatrix} \boldsymbol{s}(t) \\ \boldsymbol{h}(t) \end{pmatrix}$, then from the updating formula of NGD (11), we have

$$\boldsymbol{u}(t+1) - \boldsymbol{u}(t)$$
$$= \boldsymbol{u}\left(\boldsymbol{w}(t) - \eta\boldsymbol{J}(t)^T\boldsymbol{H}(t)^{-1}\boldsymbol{u}(\boldsymbol{w}(t))\right) - \boldsymbol{u}(\boldsymbol{w}(t))$$
$$= -\int_0^1 \left\langle \frac{\partial \boldsymbol{u}(\boldsymbol{w}(s))}{\partial \boldsymbol{w}}, \eta\boldsymbol{J}(t)^T\boldsymbol{H}(t)^{-1}\boldsymbol{u}(\boldsymbol{w}(t)) \right\rangle ds$$
$$= -\int_0^1 \left\langle \frac{\partial \boldsymbol{u}(\boldsymbol{w}(t))}{\partial \boldsymbol{w}}, \eta\boldsymbol{J}(t)^T\boldsymbol{H}(t)^{-1}\boldsymbol{u}(\boldsymbol{w}(t)) \right\rangle ds \qquad (80)$$
$$+ \int_0^1 \left\langle \frac{\partial \boldsymbol{u}(\boldsymbol{w}(t))}{\partial \boldsymbol{w}} - \frac{\partial \boldsymbol{u}(\boldsymbol{w}(s))}{\partial \boldsymbol{w}}, \eta\boldsymbol{J}(t)^T\boldsymbol{H}(t)^{-1}\boldsymbol{u}(t) \right\rangle ds$$
$$:= \boldsymbol{I}_1(t) + \boldsymbol{I}_2(t),$$

where the second equality is from the fundamental theorem of calculus and $\boldsymbol{w}(s) = s\boldsymbol{w}(t+1) + (1-s)\boldsymbol{w}(t) = \boldsymbol{w}(t) - s\eta\boldsymbol{J}(t)^T\boldsymbol{H}(t)^{-1}\boldsymbol{u}(t)$.

In the proof, we assume that Condition 3 holds for $t = 0, \cdots, k$. Then from Corollary C.3, to prove Theorem 4.7, it suffices to demonstrate that this condition also holds for $t = k+1$. Here, we primarily explain the process from Condition 3 to Corollary C.3, while other content is placed in the following full proof of Theorem 4.7.

Note that $\frac{\partial \boldsymbol{u}(\boldsymbol{w}(t))}{\partial \boldsymbol{w}} = \boldsymbol{J}(t)$, thus $\boldsymbol{I}_1(t) = \eta \boldsymbol{u}(t)$. Plugging this into (80) yields that

$$\boldsymbol{u}(t+1) = (1-\eta)\boldsymbol{u}(t) + \boldsymbol{I}_2(t). \tag{81}$$

From the above equation, we can see the difference between NGD and GD. Recall that the iteration formula for GD is

$$\boldsymbol{u}(t+1) = (1 - \eta \boldsymbol{H}(t))\boldsymbol{u}(t) + \boldsymbol{I}_1(t).$$

Precisely because of this, the convergence rate of GD is inevitably influenced by $\lambda_0$, whereas that of NGD is not.

From the stability of the Jacobian matrix, we can deduce that $\|\boldsymbol{I}_2(t)\|_2 = \mathcal{O}(\eta \|\boldsymbol{u}(t)\|_2/\sqrt{m})$. Plugging this into (81) yields that

$$
\begin{aligned}
&\|\boldsymbol{u}(t+1)\|_2^2 \\
&\leq \|(1-\eta)\boldsymbol{u}(t)\|_2^2 + \|\boldsymbol{I}_2(t)\|_2^2 + 2(1-\eta)\|\boldsymbol{u}(t)\|_2\|\boldsymbol{I}_2(t)\|_2 \\
&= \left((1-\eta)^2 + \mathcal{O}\left(\frac{\eta^2}{m}\right) + 2(1-\eta)\mathcal{O}\left(\frac{\eta}{\sqrt{m}}\right)\right)\|\boldsymbol{u}(t)\|_2^2 \\
&\leq (1-\eta)\|\boldsymbol{u}(t)\|_2^2,
\end{aligned}
\tag{82}
$$

where the last inequality holds if $m$ is large enough. $\qquad\square$

*Full Proof of Theorem 4.7.* Recall that we let $R'' = CM\sqrt{R}$ in (16) when $\sigma$ is the ReLU$^3$ activation function and let $R'' = CdR$ in (18) when $\sigma$ satisfies Assumption 4.3.

First, we can set $R' \leq R$ and $R'' \leq \frac{\sqrt{3\lambda_0}}{6}$, since $R'' \lesssim \sqrt{1-\eta}\sqrt{\lambda_0}$. Then from Lemma 4.5 we have $\|\boldsymbol{J}(t) - \boldsymbol{J}(0)\|_2 \leq \frac{\sqrt{3\lambda_0}}{6}$, thus

$$\sigma_{min}(\boldsymbol{J}(t)) \geq \sigma_{min}(\boldsymbol{J}(0)) - \|\boldsymbol{J}(t) - \boldsymbol{J}(0)\|_2 \geq \frac{\sqrt{3\lambda_0}}{2} - \frac{\sqrt{3\lambda_0}}{6} = \frac{\sqrt{3\lambda_0}}{3}$$

and then $\lambda_{min}(\boldsymbol{H}(t)) \geq \frac{\lambda_0}{3}$ for $t = 0, \cdots, k$, where $\sigma_{min}(\cdot)$ denotes the least singular value.

From the updating rule of NGD, we have

$$\boldsymbol{w}_r(t+1) = \boldsymbol{w}_r(t) - \eta \left[\boldsymbol{J}(t)^T\right]_r (\boldsymbol{H}(t))^{-1} \begin{pmatrix} \boldsymbol{s}(t) \\ \boldsymbol{h}(t) \end{pmatrix},$$

where

$$\left[\boldsymbol{J}(t)^T\right]_r = \left[\frac{\partial s_1(t)}{\partial \boldsymbol{w}_r}, \cdots, \frac{\partial s_{n_1}(t)}{\partial \boldsymbol{w}_r}, \frac{\partial h_1(t)}{\partial \boldsymbol{w}_r}, \cdots, \frac{\partial h_{n_2}(t)}{\partial \boldsymbol{w}_r}\right].$$

Therefore, for $t = 0, \cdots, k$ and any $r \in [m]$, we have

$$
\begin{aligned}
&\|\boldsymbol{w}_r(t+1) - \boldsymbol{w}_r(t)\|_2 \\
&\leq \eta\| \left[\boldsymbol{J}(t)^T\right]_r \|_2 \|\boldsymbol{H}(t)^{-1}\|_2 \sqrt{L(t)} \\
&\leq \frac{3\eta}{\lambda_0}\| \left[\boldsymbol{J}(t)^T\right]_r \|_2 \sqrt{L(t)} \\
&\leq \frac{3\eta}{\lambda_0}\| \left[\boldsymbol{J}(t)^T\right]_r \|_F \sqrt{L(t)} \\
&= \frac{3\eta}{\lambda_0}\sqrt{\sum_{p=1}^{n_1}\left\|\frac{\partial s_p(t)}{\partial \boldsymbol{w}_r}\right\|_2^2 + \sum_{j=1}^{n_2}\left\|\frac{\partial h_j(t)}{\partial \boldsymbol{w}_r}\right\|_2^2}\sqrt{L(t)} \\
&\lesssim \frac{\eta}{\lambda_0}\sqrt{\frac{B^4+1}{m}}\sqrt{L(t)} \\
&\lesssim \frac{\eta B^2}{\sqrt{m}\lambda_0}\sqrt{L(t)} \\
&\leq \frac{\eta B^2}{\sqrt{m}\lambda_0}(1-\eta)^{t/2}\sqrt{L(0)},
\end{aligned}
\tag{83}
$$

where the last inequality is due to Corollary C.3.

Summing $t$ from $0$ to $k$ yields that

$$
\begin{aligned}
&\|\boldsymbol{w}_r(k+1) - \boldsymbol{w}_r(0)\|_2 \\
&\leq \sum_{t=0}^{k}\|\boldsymbol{w}_r(t+1) - \boldsymbol{w}_r(t)\|_2 \\
&\leq C\frac{\eta B^2}{\sqrt{m}\lambda_0}\sum_{t=0}^{k}(1-\eta)^{t/2}\sqrt{L(0)} \\
&\leq \frac{CB^2\sqrt{L(0)}}{\sqrt{m}\lambda_0},
\end{aligned}
$$

where $C$ is a universal constant.

Now, when $R' \leq 1$, we can deduce that $\|\boldsymbol{w}_r(k+1)\|_2 \leq B$, implying that Condition 3 also holds for $t = k+1$. Thus, it remains only to derive the requirement for $m$.

Recall that we need $m$ to satisfy that $R' = \frac{CB^2\sqrt{L(0)}}{\sqrt{m}\lambda_0} \leq R$ and $R'' \lesssim \sqrt{1-\eta}\sqrt{\lambda_0}$.

(1) When $\sigma$ is the ReLU$^3$ activation function, in Corollary C.3, $R'' = CM\sqrt{R} \lesssim \sqrt{1-\eta}\sqrt{\lambda_0}$, implying that $R \lesssim \frac{(1-\eta)\lambda_0}{M^2}$. Then $R' = \frac{CB^2\sqrt{L(0)}}{\sqrt{m}\lambda_0} \leq R$ implies that

$$
m = \Omega\left(\frac{1}{(1-\eta)^2}\frac{M^4 B^4 L(0)}{\lambda_0^4}\right).
$$

From Lemma D.4 for the estimation of $L(0)$, i.e.,

$$
L(0) \lesssim d^2 \log\left(\frac{n_1 + n_2}{\delta}\right),
$$

we can deduce that

$$
m = \Omega\left(\frac{1}{(1-\eta)^2}\frac{d^8}{\lambda_0^4}\log^6\left(\frac{md}{\delta}\right)\log\left(\frac{n_1+n_2}{\delta}\right)\right).
$$

(2) When $\sigma$ satisfies Assumption 4.3, we have that

$$
R \lesssim \frac{\sqrt{(1-\eta)\lambda_0}}{d}, R' = \frac{CB^2\sqrt{L(0)}}{\sqrt{m}\lambda_0} \leq R.
$$

From Lemma D.4, we can deduce that

$$m = \Omega \left( \frac{1}{1-\eta} \frac{d^6}{\lambda_0^3} \log^2 \left( \frac{md}{\delta} \right) \log \left( \frac{n_1 + n_2}{\delta} \right) \right).$$

$\square$

*Proof of Corollary C.3.* Similar as before, when $R' \leq R$ and $R'' \leq \frac{\sqrt{3\lambda_0}}{6}$, we have $\sigma_{min}(\boldsymbol{J}(t)) \geq \frac{\sqrt{3\lambda_0}}{3}$ and then $\lambda_{min}(\boldsymbol{H}(t)) \geq \frac{\lambda_0}{3}$ for $t = 0, \cdots, k$.

Let $\boldsymbol{u}(t) = \begin{pmatrix} \boldsymbol{s}(t) \\ \boldsymbol{h}(t) \end{pmatrix}$, then

$$
\begin{aligned}
&\boldsymbol{u}(t+1) - \boldsymbol{u}(t) \\
&= \boldsymbol{u} \left( \boldsymbol{w}(t) - \eta \boldsymbol{J}(t)^T \boldsymbol{H}(t)^{-1} \boldsymbol{u}(\boldsymbol{w}(t)) \right) - \boldsymbol{u}(\boldsymbol{w}(t)) \\
&= - \int_0^1 \left\langle \frac{\partial \boldsymbol{u}(\boldsymbol{w}(s))}{\partial \boldsymbol{w}}, \eta \boldsymbol{J}(t)^T \boldsymbol{H}(t)^{-1} \boldsymbol{u}(\boldsymbol{w}(t)) \right\rangle ds \\
&= - \int_0^1 \left\langle \frac{\partial \boldsymbol{u}(\boldsymbol{w}(t))}{\partial \boldsymbol{w}}, \eta \boldsymbol{J}(t)^T \boldsymbol{H}(t)^{-1} \boldsymbol{u}(\boldsymbol{w}(t)) \right\rangle ds \\
&\quad + \int_0^1 \left\langle \frac{\partial \boldsymbol{u}(\boldsymbol{w}(t))}{\partial \boldsymbol{w}} - \frac{\partial \boldsymbol{u}(\boldsymbol{w}(s))}{\partial \boldsymbol{w}}, \eta \boldsymbol{J}(t)^T \boldsymbol{H}(t)^{-1} \boldsymbol{u}(\boldsymbol{w}(t)) \right\rangle ds \\
&:= \boldsymbol{I}_1(t) + \boldsymbol{I}_2(t),
\end{aligned}
\tag{84}
$$

where the second equality is from the fundamental theorem of calculus and $\boldsymbol{w}(s) = s\boldsymbol{w}(t+1) + (1-s)\boldsymbol{w}(t) = \boldsymbol{w}(t) - s\eta \boldsymbol{J}(t)^T \boldsymbol{H}(t)^{-1} \boldsymbol{u}(t)$.

Note that $\frac{\partial \boldsymbol{u}(\boldsymbol{w}(t))}{\partial \boldsymbol{w}} = \boldsymbol{J}(t)$, thus $\boldsymbol{I}_1(t) = \eta \boldsymbol{u}(t)$. Plugging this into (84) yields that

$$\boldsymbol{u}(t+1) = (1-\eta)\boldsymbol{u}(t) + \boldsymbol{I}_2(t). \tag{85}$$

Therefore, it remains only to bound $\|\boldsymbol{I}_2(t)\|_2$.

$$
\begin{aligned}
\|\boldsymbol{I}_2(t)\|_2 &= \left\| \int_0^1 \left\langle \frac{\partial \boldsymbol{u}(\boldsymbol{w}(t))}{\partial \boldsymbol{w}} - \frac{\partial \boldsymbol{u}(\boldsymbol{w}(s))}{\partial \boldsymbol{w}}, \eta \boldsymbol{J}(t)^T \boldsymbol{H}(t)^{-1} \boldsymbol{u}(\boldsymbol{w}(t)) \right\rangle ds \right\|_2 \\
&\leq \int_0^1 \|\boldsymbol{J}(\boldsymbol{w}(t)) - \boldsymbol{J}(\boldsymbol{w}(s))\|_2 \|\eta \boldsymbol{J}(t)^T \boldsymbol{H}(t)^{-1} \boldsymbol{u}(\boldsymbol{w}(t))\|_2 ds \\
&\leq \eta \|\boldsymbol{J}(t)^T \boldsymbol{H}(t)^{-1}\|_2 \|\boldsymbol{u}(\boldsymbol{w}(t))\|_2 \int_0^1 \|\boldsymbol{J}(\boldsymbol{w}(t)) - \boldsymbol{J}(\boldsymbol{w}(s))\|_2 ds \\
&\lesssim \frac{\eta \sqrt{L(t)}}{\sqrt{\lambda_0}} \int_0^1 \|\boldsymbol{J}(\boldsymbol{w}(t)) - \boldsymbol{J}(\boldsymbol{w}(s))\|_2 ds \\
&\lesssim \frac{\eta \sqrt{L(t)}}{\sqrt{\lambda_0}} \int_0^1 (\|\boldsymbol{J}(\boldsymbol{w}(t)) - \boldsymbol{J}(0)\|_2 + \|\boldsymbol{J}(\boldsymbol{w}(s)) - \boldsymbol{J}(0)\|_2) ds \\
&\lesssim \frac{\eta \sqrt{L(t)}}{\sqrt{\lambda_0}} R'',
\end{aligned}
\tag{86}
$$

where the last inequality follows from the fact that

$$\|\boldsymbol{w}_r(s) - \boldsymbol{w}_r(0)\|_2 \leq s\|\boldsymbol{w}_r(t+1) - \boldsymbol{w}_r(0)\|_2 + (1-s)\|\boldsymbol{w}_r(t) - \boldsymbol{w}_r(0)\|_2 \leq R' \leq R$$

and Lemma 4.5.

Plugging (85) into the recursion formula (84) yields that

$$
\begin{aligned}
\|\boldsymbol{u}(t+1)\|_2^2 &= \|(1-\eta)\boldsymbol{u}(t) + \boldsymbol{I}_2(t)\|_2^2 \\
&= (1-\eta)^2 \|\boldsymbol{u}(t)\|_2^2 + \|\boldsymbol{I}_2(t)\|_2^2 + 2\langle (1-\eta)\boldsymbol{u}(t), \boldsymbol{I}_2(t) \rangle \\
&\leq (1-\eta)^2 \|\boldsymbol{u}(t)\|_2^2 + \|\boldsymbol{I}_2(t)\|_2^2 + 2(1-\eta)\|\boldsymbol{u}(t)\|_2 \|\boldsymbol{I}_2(t)\|_2 \\
&\leq \left[ (1-\eta)^2 + \frac{C^2 \eta^2 (R'')^2}{\lambda_0} + 2(1-\eta)\frac{C\eta R''}{\sqrt{\lambda_0}} \right] \|\boldsymbol{u}(t)\|_2^2,
\end{aligned}
$$

where $C$ is a universal constant.

Then we can choose $R''$ such that

$$\|\boldsymbol{I}_2(t)\|_2 \leq \frac{C\eta\sqrt{L(t)}R''}{\sqrt{\lambda_0}} \leq C_1\eta\sqrt{L(t)} = C_1\eta\sqrt{\boldsymbol{u}(t)},$$

where $C$ is a universal constant and $C_1$ is a constant to be determined.

Thus, we can deduce that

$$\begin{aligned}
\|\boldsymbol{u}(t+1)\|_2^2 &\leq \left[(1-\eta)^2 + (C_1\eta)^2 + 2(1-\eta)C_1\eta\right]\|\boldsymbol{u}(t)\|_2^2 \\
&= \left[(1-\eta) + \eta(\eta C_1^2 + 2(1-\eta)C_1 + \eta - 1)\right]\|\boldsymbol{u}(t)\|_2^2 \\
&\leq (1-\eta)\|\boldsymbol{u}(t)\|_2^2,
\end{aligned}$$

where in the last inequality is due to that we can choose $C_1$ such that $\eta C_1^2 + 2(1-\eta)C_1 + \eta - 1 \leq 0$.

Note that since $\eta \in (0,1)$, the quadratic equation $\eta x^2 + 2(1-\eta)x + \eta - 1 = 0$ has one negative root and one positive root, denoted as $x_0$ and $x_1$ respectively. Therefore, the condition $C_1 \leq x_1$ is sufficient to satisfy the requirement. The explicit form of $x_1$ can be written as:

$$x_1 = \frac{2(\eta - 1) + \sqrt{4(1-\eta)^2 - 4\eta(\eta - 1)}}{2\eta} = \frac{\sqrt{1-\eta}}{1 + \sqrt{1-\eta}} \geq \frac{\sqrt{1-\eta}}{2}.$$

Thus, $C_1 = \frac{\sqrt{1-\eta}}{2}$ is sufficient to satisfy that $\eta C_1^2 + 2(1-\eta)C_1 + \eta - 1 \leq 0$.

From this, we can deduce that

$$R'' \lesssim C_1\sqrt{\lambda_0} \lesssim \sqrt{1-\eta}\sqrt{\lambda_0}.$$

Therefore, we can conclude that $\|\boldsymbol{u}(t)\|_2^2 \leq (1-\eta)^t\|\boldsymbol{u}(0)\|_2^2$ holds for $t = 0, \cdots, k$.

$\square$

## C.4 PROOF OF COROLLARY 4.9

*Proof.* In the proof of Theorem 4.7, we have proved that Condition 3 holds for all $t \in \mathbb{N}$. Thus, it is sufficient to prove that Condition 3 can lead to the conclusion in Corollary 4.9.

Setting $\eta = 1$ in (85) yields that

$$\boldsymbol{u}(t+1) = \boldsymbol{I}_2(t).$$

We have that

$$\|\boldsymbol{I}_2(t)\|_2 \lesssim \frac{\sqrt{L(t)}}{\sqrt{\lambda_0}} \int_0^1 \|\boldsymbol{J}(\boldsymbol{w}(t)) - \boldsymbol{J}(\boldsymbol{w}(s))\|_2 ds. \tag{87}$$

Since $\boldsymbol{w}(s) = s\boldsymbol{w}(t+1) + (1-s)\boldsymbol{w}(t)$, then for any $r \in [m]$, we have $\|\boldsymbol{w}_r(s)\|_2 \leq s\|\boldsymbol{w}_r(t+1)\|_2 + (1-s)\|\boldsymbol{w}_r(t)\|_2 \leq B$.

When $\sigma(\cdot)$ is smooth, we can deduce that for any $r \in [m]$,

$$\left\|\frac{\partial s_p(\boldsymbol{w}(s))}{\partial \boldsymbol{w}_r} - \frac{\partial s_p(\boldsymbol{w}(t))}{\partial \boldsymbol{w}_r}\right\|_2 \lesssim \frac{1}{\sqrt{n_1 m}}(B^2+1)\|\boldsymbol{w}_r(s) - \boldsymbol{w}_r(t)\|_2 \leq \frac{1}{\sqrt{n_1 m}}(B^2+1)\|\boldsymbol{w}_r(t+1) - \boldsymbol{w}_r(t)\|_2$$

and

$$\left\|\frac{\partial h_j(\boldsymbol{w}(s))}{\partial \boldsymbol{w}_r} - \frac{\partial h_j(\boldsymbol{w}(t))}{\partial \boldsymbol{w}_r}\right\|_2 \lesssim \frac{1}{\sqrt{n_1 m}}(B+1)\|\boldsymbol{w}_r(s) - \boldsymbol{w}_r(t)\|_2 \leq \frac{1}{\sqrt{n_1 m}}(B+1)\|\boldsymbol{w}_r(t+1) - \boldsymbol{w}_r(t)\|_2.$$

We know that for any $r \in [m]$,

$$\|\boldsymbol{w}_r(t+1) - \boldsymbol{w}_r(t)\|_2 \lesssim \frac{B^2}{\sqrt{m}\lambda_0}\sqrt{L(t)}.$$

Thus for any $s \in [0, 1]$, we have

$$\|\boldsymbol{J}(\boldsymbol{w}(s)) - \boldsymbol{J}(\boldsymbol{w}(t))\|_2^2$$

$$\leq \sum_{r=1}^m \left( \sum_{p=1}^{n_1} \left\| \frac{\partial s_p(\boldsymbol{w}(s))}{\partial \boldsymbol{w}_r} - \frac{\partial s_p(\boldsymbol{w}(t))}{\partial \boldsymbol{w}_r} \right\|_2^2 + \left\| \frac{\partial h_j(\boldsymbol{w}(s))}{\partial \boldsymbol{w}_r} - \frac{\partial h_j(\boldsymbol{w}(t))}{\partial \boldsymbol{w}_r} \right\|_2^2 \right)$$

$$\lesssim \frac{1}{m} \sum_{r=1}^m \left( (B^4 + 1)\|\boldsymbol{w}_r(t+1) - \boldsymbol{w}_r(t)\|_2^2 + (B^2 + 1)\|\boldsymbol{w}_r(t+1) - \boldsymbol{w}_r(t)\|_2^2 \right)$$

$$\lesssim B^4 \left( \frac{B^2}{\sqrt{m}\lambda_0} \sqrt{L(t)} \right)^2 .$$

Plugging this into (87), we have

$$\|\boldsymbol{I}_2(t)\|_2 \lesssim \frac{\sqrt{L(t)}}{\sqrt{\lambda_0}} \int_0^1 \|\boldsymbol{J}(\boldsymbol{w}(t)) - \boldsymbol{J}(\boldsymbol{w}(s))\|_2 ds$$

$$\lesssim \frac{\sqrt{L(t)}}{\sqrt{\lambda_0}} \frac{B^4}{\sqrt{m}\lambda_0} \sqrt{L(t)}$$

$$= \frac{B^4}{\sqrt{m}\lambda_0^3} L(t).$$

Combining with the fact $\boldsymbol{u}(t+1) = \boldsymbol{I}_2(t)$ yields that

$$\left\| \begin{pmatrix} \boldsymbol{s}(t+1) \\ \boldsymbol{h}(t+1) \end{pmatrix} \right\|_2 \leq \frac{CB^4}{\sqrt{m}\lambda_0^3} \left\| \begin{pmatrix} \boldsymbol{s}(t) \\ \boldsymbol{h}(t) \end{pmatrix} \right\|_2^2$$

holds for $t \in \mathbb{N}$, where $C$ is a universal constant.

In the proof above, we only require that $R' \leq R$ and $R'' = CdR \leq \frac{\sqrt{3\lambda_0}}{6}$, leading to the requirement for $m$ that

$$m = \Omega \left( \frac{d^6}{\lambda_0^3} \log^2 \left( \frac{md}{\delta} \right) \log \left( \frac{n_1 + n_2}{\delta} \right) \right).$$

$\square$

## D    AUXILIARY LEMMAS

**Lemma D.1** (Theorem 3.1 in Kuchibhotla & Chakrabortty (2022)). *If $X_1, \cdots, X_n$ are independent mean zero random variables with $\|X_i\|_{\psi_\alpha} < \infty$ for all $1 \leq i \leq n$ and some $\alpha > 0$, then for any vector $a = (a_1, \cdots, a_n) \in \mathbb{R}^n$, the following holds true:*

$$P \left( \left| \sum_{i=1}^n a_i X_i \right| \geq 2eC(\alpha)\|b\|_2\sqrt{t} + 2eL_n^*(\alpha)t^{1/\alpha}\|b\|_{\beta(\alpha)} \right) \leq 2e^{-t}, \ for \ all \ t \geq 0,$$

*where $b = (a_1\|X_1\|_{\psi_\alpha}, \cdots, a_n\|X_n\|_{\psi_\alpha}) \in \mathbb{R}^n$,*

$$C(\alpha) := \max\{\sqrt{2}, 2^{1/\alpha}\} \begin{cases} \sqrt{8}(2\pi)^{1/4}e^{1/24}(e^{2/e}/\alpha)^{1/\alpha}, & if \ \alpha < 1, \\ 4e + 2(\log 2)^{1/\alpha}, & if \ \alpha \geq 1. \end{cases}$$

*and for $\beta(\alpha) = \infty$ when $\alpha \leq 1$ and $\beta(\alpha) = \alpha/(\alpha - 1)$ when $\alpha > 1$,*

$$L_n^*(\alpha) := \frac{4^{1/\alpha}}{\sqrt{2}} \times \begin{cases} C(\alpha), & if \ \alpha < 1, \\ 4e, & if \ \alpha \geq 1. \end{cases}$$

In the following, we will provide some preliminary information about Orlicz norms.

Let $f : [0, \infty) \to [0, \infty)$ be a non-decreasing function with $f(0) = 0$. The $f$-Orlicz norm of a real-valued random variable $X$ is given by

$$\|X\|_f := \inf\{C > 0 : \mathbb{E}\left[ f\left( \frac{|X|}{C} \right) \right] \leq 1\}.$$

If $\|X\|_{\psi_\alpha} < \infty$, we say that $X$ is sub-Weibull of order $\alpha > 0$, where

$$\psi_\alpha(x) := e^{x^\alpha} - 1.$$

Note that when $\alpha \geq 1$, $\|\cdot\|_{\psi_\alpha}$ is a norm and when $0 < \alpha < 1$, $\|\cdot\|_{\psi_\alpha}$ is a quasi-norm. Moreover, since $(|a| + |b|)^\alpha \leq |a|^\alpha + |b|^\alpha$ holds for any $a, b \in \mathbb{R}$ and $0 < \alpha < 1$, we can deduce that

$$\mathbb{E}e^{\frac{|X+Y|^\alpha}{|C|^\alpha}} \leq \mathbb{E}e^{\frac{|X|^\alpha + |Y|^\alpha}{|C|^\alpha}} = \mathbb{E}e^{\frac{|X|^\alpha}{|C|^\alpha}} e^{\frac{|Y|^\alpha}{|C|^\alpha}} \leq \left( \mathbb{E}e^{\frac{2|X|^\alpha}{|C|^\alpha}} \right)^{1/2} \left( \mathbb{E}e^{\frac{2|Y|^\alpha}{|C|^\alpha}} \right)^{1/2}.$$

This implies that

$$\|X + Y\|_{\psi_\alpha} \leq 2^{1/\alpha} \max\{\|X\|_{\psi_\alpha}, \|Y\|_{\psi_\alpha}\} \leq 2^{1/\alpha}(\|X\|_{\psi_\alpha} + \|Y\|_{\psi_\alpha}).$$

Furthermore, for $p, q > 0$, we have $\||X|\|_{\psi_p} = \||X|^{p/q}\|_{\psi_q}^{q/p}$. And in the related proofs, we may frequently use the fact that for real-valued random variable $X \sim \mathcal{N}(0, 1)$, we have $\|X\|_{\psi_2} \leq \sqrt{6}$ and $\|X^2\|_{\psi_1} = \|X\|_{\psi_2}^2 \leq 6$.

**Lemma D.2.** *If $\|X\|_{\psi_\alpha}, \|Y\|_{\psi_\beta} < \infty$ with $\alpha, \beta > 0$, then we have $\|XY\|_{\psi_\gamma} \leq \|X\|_{\psi_\alpha}\|Y\|_{\psi_\beta}$, where $\gamma$ satisfies that*

$$\frac{1}{\gamma} = \frac{1}{\alpha} + \frac{1}{\beta}.$$

*Proof.* Without loss of generality, we can assume that $\|X\|_{\psi_\alpha} = \|Y\|_{\psi_\beta} = 1$. To prove this, let us use Young's inequality, which states that

$$xy \leq \frac{x^p}{p} + \frac{y^q}{q}, \, for \, x, y \geq 0, p, q > 1.$$

Let $p = \alpha/\gamma, q = \beta/\gamma$, then

$$\begin{aligned}
\mathbb{E}[\exp(|XY|^\gamma)] &\leq \mathbb{E}\left[ \exp\left( \frac{|X|^{\gamma p}}{p} + \frac{|Y|^{\gamma q}}{q} \right) \right] \\
&= \mathbb{E}\left[ \exp\left( \frac{|X|^\alpha}{p} \right) \exp\left( \frac{|Y|^\beta}{q} \right) \right] \\
&\leq \mathbb{E}\left[ \frac{\exp(|X|^\alpha)}{p} + \frac{\exp(|Y|^\beta)}{q} \right] \\
&\leq \frac{2}{p} + \frac{2}{q} \\
&= 2,
\end{aligned}$$

where the first and second inequality follow from Young's inequality. From this, we have that $\|XY\|_{\psi_\gamma} \leq \|X\|_{\psi_\alpha}\|Y\|_{\psi_\beta}$.

$\square$

**Lemma D.3** (Bernstein inequality, Theorem 3.1.7 in Giné & Nickl (2021)). *Let $X_i$, $1 \leq i \leq n$ be independent centered random variables a.s. bounded by $c < \infty$ in absolute value. Set $\sigma^2 = 1/n \sum_{i=1}^n \mathbb{E}X_i^2$ and $S_n = 1/n \sum_{i=1}^n X_i$. Then, for all $t \geq 0$,*

$$P\left( S_n \geq \sqrt{\frac{2\sigma^2 t}{n}} + \frac{ct}{3n} \right) \leq e^{-u}.$$

**Lemma D.4.** *For $0 < \delta < 1$, with probability at least $1 - \delta$, we have that when $m \geq \log^2\left( \frac{n_1 + n_2}{\delta} \right)$,*

$$L(0) = \left\| \begin{pmatrix} s(0) \\ h(0) \end{pmatrix} \right\|_2^2 = \mathcal{O}\left( d^2 \log\left( \frac{n_1 + n_2}{\delta} \right) \right).$$

*Proof.* Recall that for $p \in [n_1]$,

$$s_p(0) = \frac{1}{\sqrt{n_1}} \left[ \frac{1}{\sqrt{m}} \sum_{r=1}^{m} a_r \left( \sigma'(\boldsymbol{w}_r(0)^T \boldsymbol{x}_p) w_{r0}(0) - \sigma''(\boldsymbol{w}_r(0)^T \boldsymbol{x}_p) \|\boldsymbol{w}_{r1}(0)\|_2^2 \right) - f(x_p) \right]$$

and for $j \in [n_2]$,

$$h_j(0) = \frac{1}{\sqrt{n_2}} \left[ \frac{1}{\sqrt{m}} \sum_{r=1}^{m} a_r \sigma(\boldsymbol{w}_r(0)^T \boldsymbol{y}_j) - g(\boldsymbol{y}_j) \right].$$

Then

$$L(0) = \sum_{p=1}^{n_1} \frac{1}{2} (s_p(0))^2 + \sum_{j=1}^{n_2} \frac{1}{2} (h_j(0))^2$$

$$\leq \frac{1}{n_1} \sum_{p=1}^{n_1} \left( \frac{1}{\sqrt{m}} \sum_{r=1}^{m} a_r \left( \sigma'(\boldsymbol{w}_r(0)^T \boldsymbol{x}_p) w_{r0}(0) - \sigma''(\boldsymbol{w}_r(0)^T \boldsymbol{x}_p) \|\boldsymbol{w}_{r1}(0)\|_2^2 \right) \right)^2 + \frac{1}{n_1} \sum_{p=1}^{n_1} f^2(x_p)$$

$$+ \frac{1}{n_2} \sum_{j=1}^{n_2} \left( \frac{1}{\sqrt{m}} \sum_{r=1}^{m} a_r \sigma(\boldsymbol{w}_r(0)^T \boldsymbol{y}_j) \right)^2 + \frac{1}{n_2} \sum_{j=1}^{n_2} g^2(\boldsymbol{y}_j).$$

Note that

$$\left| a_r \left( \sigma'(\boldsymbol{w}_r(0)^T \boldsymbol{x}_p) w_{r0} - \sigma''(\boldsymbol{w}_r(0)^T \boldsymbol{x}_p) \|\boldsymbol{w}_{r1}(0)\|_2^2 \right) \right| \lesssim \|\boldsymbol{w}_r(0)\|_2^2 |\boldsymbol{w}_r(0)^T \boldsymbol{x}_p|$$

and $\left| a_r \sigma(\boldsymbol{w}_r(0)^T \boldsymbol{y}_j) \right| \lesssim \|\boldsymbol{w}_r(0)\|_2^2 |\boldsymbol{w}_r(0)^T \boldsymbol{y}_j|$.

Since $\left\| \|\boldsymbol{w}_r(0)\|_2^2 \right\|_{\psi_1} = \mathcal{O}(d)$ and $\|\boldsymbol{w}_r(0)^T \boldsymbol{y}_j\|_{\psi_2}, \|\boldsymbol{w}_r(0)^T \boldsymbol{x}_p\|_{\psi_2} = \mathcal{O}(1)$, from Lemma D.2, we have that

$$\left\| \|\boldsymbol{w}_r(0)\|_2^2 |\boldsymbol{w}_r(0)^T \boldsymbol{x}_p| \right\|_{\psi_{\frac{2}{3}}} = \mathcal{O}(d), \left\| |\boldsymbol{w}_r(0)^T \boldsymbol{y}_j| \right\|_{\psi_{\frac{2}{3}}} = \mathcal{O}(d).$$

Applying Lemma D.1 yields that for fixed $p \in [n_1]$ and $j \in [n_2]$ with probability at least $1 - 2e^{-t}$,

$$\left| \frac{1}{\sqrt{m}} \sum_{r=1}^{m} a_r \left( \sigma'(\boldsymbol{w}_r(0)^T \boldsymbol{x}_p) w_{r0}(0) - \sigma''(\boldsymbol{w}_r(0)^T \boldsymbol{x}_p) \|\boldsymbol{w}_{r1}(0)\|_2^2 \right) \right| \lesssim d\sqrt{t} + \frac{d}{\sqrt{m}} t^{\frac{3}{2}}$$

and with probability at least $1 - 2e^{-t}$,

$$\left| \frac{1}{\sqrt{m}} \sum_{r=1}^{m} a_r \sigma(\boldsymbol{w}_r(0)^T \boldsymbol{y}_j) \right| \lesssim d\sqrt{t} + \frac{d}{\sqrt{m}} t^{\frac{3}{2}}.$$

Then taking a union bound for all $p \in [n_1]$ and $j \in [n_2]$ with $2(n_1 + n_2)e^{-t} = \delta$ yields that

$$L(0) \lesssim \left( d\sqrt{t} + \frac{d}{\sqrt{m}} t^{\frac{3}{2}} \right)^2$$

$$\lesssim d^2 t + \frac{d^2 t^3}{m}$$

$$= d^2 \left( \log\left( \frac{n_1 + n_2}{\delta} \right) + \frac{1}{m} \log^3\left( \frac{n_1 + n_2}{\delta} \right) \right)$$

$$\lesssim d^2 \log\left( \frac{n_1 + n_2}{\delta} \right),$$

since $m \geq \log^2\left( \frac{n_1 + n_2}{\delta} \right)$.

$\square$

