# OpenReview forum: "Fast Convergence of Natural Gradient Descent for Over-parameterized Physics-Informed Neural Networks"
_ICLR.cc/2026/Conference — ICLR 2026 Poster_

### Official Review · Reviewer_yT77 · 2025-10-19

**Soundness:** 3
**Presentation:** 3
**Contribution:** 2
**Rating:** 6
**Confidence:** 3

**Summary:**

Authors provide a proof for the convergence rate of a certain type of neural network that depends on the norm of the Hessian, which seems to be a new result (or an extension of a known result to a broader class of problems). The paper is clear and well-written, and overall seems like a good contribution to the field of theoretical guarantees for neural network training.

**Strengths:**

The paper is clear, of a high-technical quality and of a fair significance. It is an extension of a result known for regression problem to more general PINNs (although still restricted to two-layer networks).

**Weaknesses:**

A few things come to mind:
1) The type of network it is applied to is quite restrictive (two-layer ReLU PINNs). While I understand that this allows you to construct the proof, could there not be a way to extend the result to more general architectures? This would make the result and impact much stronger.
2) It would be good to numerically validate the main result of the paper, where you take a simple problem where you can calculate $1/|H^{\infty}|_2$ exactly and verify that the convergence rate matches it.
3) A natural question that comes to mind is what happens when you have deeper networks. It would be interesting to at least numerically study this question, by replicating your experiments with three and four-layer networks perhaps.

**Questions:**

See weaknesses.

---

> ### Author Response · Authors · 2025-11-17
>
> We thank the reviewer for the positive assessment of the paper’s clarity, technical quality, and significance, as well as for the constructive suggestions regarding the scope and empirical validation. We respond to each point below.
>
> ### **W1: Restriction to two-layer $\text{ReLU}$ PINNs.**
>
> We agree that the current theoretical framework is limited to two-layer networks. This restriction is primarily technical: it enables precise control of the neural tangent kernel (NTK) evolution and allows us to rigorously establish Jacobian stability (Lemma 4.6) and global convergence (Theorem 4.7).
>
> Extending these results to deeper networks is indeed possible but significantly more involved, as it requires layer-wise coupling analysis of the NTK (as in Allen-Zhu et al., 2019, Du et al.,2020). Our approach provides the foundation for such an extension, since the same neuron-wise stability and local curvature arguments can be recursively applied across layers.
>
> Another advantage of the analysis of this work is that **we extend the activation function requirements to smooth activations such as $Tanh$, not limiting to $\text{ReLU}$ activations**. In PINNs' community, the $\text{ReLU}$ is seldom used as high orders of derivatives are zero, and the commonly used PINN is $Tanh, Sin$, etc. We proved the convergence of NGD under these smooth activations.
>
> ### **W2: Numerical validation of theoretical convergence rates.**
> Very nice persperctive and we really appreciate this suggestion. We have added experiments to directly validate our theoretical results in the revised manuscript.  **The codes have been updated in the Supplementary Material (e.g. /poisson_1d/main_ngd.py).**
>
> For Theorem 3.7, prior work (e.g., Gao et al., 2023) requires the step size to be $\eta = \mathcal{O}(\lambda_{\min})$, and our improvement is $\eta = \mathcal{O}(1/\lambda_{\max})$.
> Taking the 1D Poisson equation as example, where sample size $n_1=100,n_2=2$ and layer width $m=128$, we compute the Gram matrix and get the requirements $\lambda_{\min}=3.47\times10^{-11}$ in Gao et.al.2023, and our requirements is $1/\lambda_{\max}=1/(1.73\times10^4)=5.78\times10^{-5}$, shows the improvements on the learning rate.
>
> For our main Theorems 4.7, we have compared empirical training loss curves with the theoretical linear rates in Theorems 4.7. The theoretical decay follows $L(k) \approx C(1-\eta)^k$, the computed predicted decay is $L(k)\approx\mathcal{O}(k^{-1.55})$ for simple 1D Poisson equation, $\mathcal{O}(k^{-1.92})$ for 1D Heat equation and $\mathcal{O}(k^{-1.13})$ for 2D Poisson equation. For the heat equation, convergence initially exceeds the predicted rate and later slows markedly. This is consistent with known NTK decay and multi-phase behaviors in PINNs. Generally, the empirical loss of NGD roughly follows the predicted linear regime in early iterations, before entering a slower phase usually observed across all optimizers. These comparisons have been added in a new Figure 2 (lines 483-514 in the revised manuscript).
>
> ### **W3: Behavior of deeper networks.**
> We agree that exploring deeper architectures is valuable for assessing the generality of our results.
> We have extended the experiments to multiple-layer PINNs using the same PDE tasks. The observed convergence trends remain consistent with our theoretical predictions: NGD maintains a faster and more stable decay than GD or Adam, while the overall convergence slows moderately as depth increases.
> **These results have be added in the appendix line 1005-1025.** The following table shows the training comparison for different layers on 2D Poisson equation.
>
> |Hidden layers | Total parameters| Training efficiency | Training time | Max memory | Rel. L2 error |
> |--- | ---| --- | --- | --- | --- |
> |1 |  $512$  | 3.67 s/epoch  | 12min13s  | 14.75 MB   | 1.12e-04 |
> |3 |  $33,280$  | 8.29 s/epoch  | 27min37s  | 14.75 MB   | 3.41e-04 |
> |6 |  $82,432$  | 17.53 s/epoch  | 58min25s  | 20.17 MB   | 4.29e-04 |
>
>
> ### **Summary of revisions in the revised manuscript.**
> - Numerical validation of theoretical convergence rates: lines 483-514.
> - Numerical behavior of deeper networks:  line 1005-1025.
>
> Finally, we thank the reviewer again for the helpful feedback and for recognizing the contribution of extending optimization theory from regression to PINNs. We believe these additions will further strengthen the work both theoretically and empirically.

---

> > ### Author Response · Authors · 2025-11-28
> >
> > Dear Reviewer yT77,
> >
> > Thank you for your initial positive evaluation of our work and constructive suggestions. We would appreciate it if you could take a look at our responses and let us know whether they address your concerns.
> >
> > Authors.

---

### Official Review · Reviewer_jYGf · 2025-10-27

**Soundness:** 3
**Presentation:** 2
**Contribution:** 2
**Rating:** 4
**Confidence:** 4

**Summary:**

The manuscript provides training guarantees for physics-informed neural networks in an overparametrized setting with NTK-scale parametrization, both for first-order gradient descent as well as a second-order natural gradient descent. For gradient descent, the allowed step size of $O(1/\lambda_\max)$ is different compared to prior works with $O(\lambda_\min)$ step size on optimization in PINNs. For natural gradient descent, a convergence rate independent of the condition number of the problem is derived. Additionally, computational experiments comparing first-order to second-order optimizers in PINNs are provided.

**Strengths:**

+ Overall, the paper is well written and easy to navigate and the main contributions are clearly described.
+ The paper addresses a timely and important question of the convergence of optimization physics-informed models.
+ The guarantees go beyond the use of first-order methods and treat natural gradient, which seems to be the arguably most efficient optimizer in PINNs.
+ The theoretical analysis uses timely methods from supervised learning and rigorously transfers them to PINNs.

**Weaknesses:**

+ Discussion of related works:
    + Use of *our NGD*: At multiple places in the manuscript, *our NGD* is used to refer to the natural gradient method that is being analyzed. Also, in Remark 4.2 it is claimed that the method studied is different from previously proposed natural gradient or Gauss-Newton methods for PINNs. The iteration of energy natural gradient (Müller and Zeinhofer, 2023), which reduces to a Gauss-Newton iteration for PINNs, is given by $$ w(k+1)=w(k)-\eta (J(k)^T J(k))^+J(k)^T \binom{s(k)}{h(k)}, $$ where $A^+$ denotes an arbitrary pseudo-inverse of $A$. By choosing the Moore-Penrose inverse and considering an overparametrized setting in which $J(k)J(k)^T$ has full rank, this specializes to $$w(k+1)=w(k)-\eta J(k)^T (J(k)J(k)^T)^{-1} \binom{s(k)}{h(k)},$$ which agrees with the iteration studied in the manuscript. Hence, from my understanding, the manuscript is studying the convergence properties of the optimizer proposed by Müller and Zeinhofer (2023), however, this is not mentioned at any place. Rather, the impression is conveyed that the manuscript proposed a new variant of a natural gradient method, see Remark 4.2 as well as line 408 and Table 1. This
   + There are certain works on optimization in PINNs missing in the discussion of related works in Subsection 1.2., e.g.,
      + AN OPERATOR PRECONDITIONING PERSPECTIVE ON TRAINING IN PHYSICS-INFORMED MACHINE LEARNING, Tim De Ryck, Florent Bonnet, Siddhartha Mishra, Emmanuel de Bézenac; in particular, Theorem 2.3 shows that for the linearization problem with preconditioning achieves a converge rate given by the condition number and that the natural gradient preconditioning achieves an optimal condition number of 1. As such, this seems to be a linearized (hence easier) version of the main result in the manuscript.
      + Convergence of Stochastic Gradient Methods for Wide Two-Layer Physics-Informed Neural Networks, BANGTI JIN AND LONGJUN WU, 2025; provide an exponential convergence rate of SGD under overparametrization
      + Non-Asymptotic Analysis of Projected Gradient Descent for Physics-Informed Neural Networks, JONAS NIESEN, AND JOHANNES MÜLLER, 2025; provide a sublinear convergence guarantee for arbitrary sampled (S)GD without assumption on the network size
+ Overparametrization assumption: The manuscript is making the global assumption of overparametrization. Where I understand that this allows the use of an established machinery, I do not believe that this is the setting that PINNs are used in. In particular, note that in PINNs, the data points are synthetically sampled integration points of the computational domain rather than data points like in supervised learning. As such, the problem in PINNs is rather an optimization than a statistical problem. In practice, new data points are sampled continuously throughout optimization. Hence, it is not clear how practically relevant the setting of overparametrization is. However, this assumption is not uncommon and the only work I am aware not making an assumption on the size of the PINN is by Niessen and Müller (2025).
+ Difference of PINNs to supervised learning: In the introduction, it is mentioned that the reason why NGD is not used in supervised learning is due to its high computational cost. However, I believe that there another, arguably more important reason: The loss function of PINNs has a much worse conditionining due to the appearence of the PDE operator. It is stated in line 251 that the conditioning can be really bad for complex PDEs. From my understanding, this can also be the case for simple PDEs.
+ Experiments: The experiments compare natural gradient to SGD, Adam and L-BFGS. However, this comparison has been made at several places, in particular, by Müller and Zeinhofer (2023). I think, it would be much more informative to not repeat this comparison, but to see, how the empirically observed convergence rates relate to the theoretical guarnatees. Further, in relation to the overparametrization assumption, the influence on the network size and optimizer on the generalization error is not studied empirically.

**Questions:**

+ Can you elaborate why you regard your $O(1/\lambda_\max)$ as an improvement over $O(\lambda_\min)$?
+ Can you elaborate on the relation of the natural gradient defined in Section 4 to previously proposed methods like energy natural gradient? In case of equivalence, can you adapt your presentation accordingly, see also my discussion above.
+ Can you comment why you make the overparametrization assumption? Also, a discussion of the overparametrization assumption inside the manuscript seems reasonable.
+ Can you comment on the different condition numbers of supervised learning and PINNs? Can you elaborate why you only expect a bad condition number for complex PDEs (see line 251)? This could be added in the manuscript as a motivation for second order and natural gradient methods.
+ Can you validate the theoretical convergence guarantees within the computational experiments? In particular:
   + Compare the loss curves to the linear convergence guarantees from Theorem 3.7 and 4.7.
   + Compare the generalization error for different network size and optimizers. As the convergence guarantees are merely given with respect to the loss function, it is informative to contrast it with the relative L^2 error.
   + Improve readability of the plot.

**Minor comments:**
1. In the abstract, it is stated that, *However, the learning rate of GD for training two-layer neural networks exhibits poor dependence on the sample size and the Gram matrix*. Note that the term learning rate usually refers to the step size in machine learning, not the convergence rate of the learning process.
2. It is stated in several places that the learning rate can be O(1). However, it is shown that the convergence rate is O(1).
3. In the limitations section on scalability of natural gradient methods in these scenarios, the following directly relevant works are not referenced:
+ Kronecker-Factored Approximate Curvature for Physics-Informed Neural Networks, Dangel et al. 2024
+ Improving Energy Natural Gradient Descent through Woodbury, Momentum, and Randomization, Dangel et al. 2025

---

> ### Author Response · Authors · 2025-11-17
> **Part 1**
>
> We sincerely thank the reviewer for the careful reading and detailed feedback, as well as for highlighting the clarity, timeliness, and rigor of our theoretical contributions. Below we address each point raised and indicate specific manuscript revisions we have made.
>
> ### **W1: Relation to Müller & Zeinhofer (2023) and prior NGD formulations.**
>
> We appreciate the opportunity to clarify this point.
> Indeed, when the Moore–Penrose pseudoinverse is chosen and $J$ is row full rank, the NGD update in our paper
> $$
> \begin{equation}
> w(k+1)=w(k)−ηJ(k)^T(J(k)J(k)^T)^{−1} (s(k)^T h(k)^T)^T
> \end{equation}
> $$
> coincides algebraically with the energy natural gradient step of Müller & Zeinhofer (2023) under their setting of overparameterized PINNs. however, **our contribution is not in proposing a new variant, but in providing the first rigorous convergence proof for this optimizer within the PINN framework.**
>
> While Müller & Zeinhofer (2023) demonstrated strong empirical performance, they did not analyze the convergence behavior or dependence on the Gram matrix, activation smoothness, or width $m$. We show that, under over-parameterization, this same iteration converges globally with rate independent of the Gram matrix’s smallest eigenvalue, and even achieves quadratic convergence for smooth activations (Theorem 4.7, Corollary 4.9).
>
> We have **explicitly acknowledged this equivalence**, cited Müller & Zeinhofer (2023) as the original algorithmic source, and clarify that our novelty lies in its theoretical analysis, not in the introduction of a new variant. Corresponding changes have been made in Remark 4.2, line 430, and Table 1&3 in the revised pdf.
>
> ### **W2: Missing related works.**
>
> We thank the reviewer for pointing out several valuable latest references. We add the following in Section 1.2:
>
> - De Ryck et al., 2023 [a]: We note that their operator-preconditioning analysis establishes convergence for linearized PINN problems and that our work extends this line of reasoning to the overparameterized regime with explicit eigenvalue- and width-dependent rates.
> - Jin & Wu, 2025 [b] and Niesen & Müller, 2025 [c]: We cite both as recent studies of SGD and projected GD for PINNs, and emphasize that our analysis complements these by addressing second-order dynamics and achieving faster theoretical convergence.
> - In Section 6 (Limitations), we cite Dangel et al., 2024[d]; 2025[e,f] for K-FAC and Woodbury-based scalable NGD variants, which align directly with our discussion on computational cost.
>
> **Brief Comparison:**
> Jin & Wu (2025) [b] extends the method in [g] to SGD in training PINNs, while sharing similar limitations with [g] - for instance, the learning rate must satisfy $\eta=O(\lambda_0)$. Specifically, [g] establishes the linear convergence of SGD for KAN (Kolmogorov–Arnold Network) and PIKAN, though this convergence comes with stricter constraints on network width and learning rate than GD (see Eq (17) and (18) of Theorem 4.3 in [g]). Thus, investigating the convergence of stochastic algorithms under milder conditions constitutes a crucial direction for our future work.
>
> Niesen & Müller (2025) [c] provides a non-asymptotic convergence analysis of projected gradient descent (PGD) for PINNs without the overparameterization assumption. Our proof is conducted within the Neural Tangent Kernel (NTK) framework, hence requiring the overparameterization assumption. [c]’s proof is based on a framework analogous to random feature models, relying on a distinct key assumption.  Specifically, [c] assumes that the solution to the PDEs can be well approximated by a Reproducing Kernel Hilbert Space (RKHS) (see Equations (17), (23), and (24) in [c]), which ensures the meaningfulness of the final convergence results. In summary, our work and [c] follow distinct approaches, each with its own advantages and disadvantages.
>
> We have included and cited the following references in the revised manuscript to ensure that our research’s positioning in the existing literature is clear and complete.
>
> - [a] De Ryck T, Bonnet F, Mishra S, et al. An operator preconditioning perspective on training in physics-informed machine learning. ICLR, 2024.
> - [b] Jin B, Wu L. Convergence of Stochastic Gradient Methods for Wide Two-Layer Physics-Informed Neural Networks. arXiv:2508.21571.
> - [c] Niesen J, Müller J. Non-Asymptotic Analysis of Projected Gradient Descent for Physics-Informed Neural Networks. arXiv:2505.07311.
> - [d] Dangel F, Müller J, Zeinhofer M. Kronecker-factored approximate curvature for physics-informed neural networks. NeurIPS, 2024.
> - [e] Dangel F, Mucsányi B, Weber T, et al. Kronecker-factored Approximate Curvature (KFAC) From Scratch[J]. arXiv:2507.05127.
> - [f] Guzmán-Cordero A, Dangel F, Goldshlager G, et al. Improving Energy Natural Gradient Descent through Woodbury, Momentum, and Randomization. NeurIPS, 2025.
> - [g] Gao Y, Tan VYF. On the Convergence of (Stochastic) Gradient Descent for Kolmogorov–Arnold Networks.TiT, 2025.

---

> ### Author Response · Authors · 2025-11-17
> **Part 2**
>
> ### **W3: Over-parameterization assumption.**
>
> We agree that real-world PINNs often resample collocation points during training, so strict over-parameterization may not always hold. However, as the reviewer notes, this assumption enables us to apply the NTK framework to analyze the convergence of  optimization algorithms (e.g., Du et al. 2018; Gao et al. 2023).
>
> We thanks the reviewer for pointing the work of Niessen and Müller (2025), which brings another framework that without assumption on the network size.  As mentioned in the response to W2, this is a different framework with us. Although their results without assuming over-parameterization are solid, they require that the true solution of the PDE can be well-approximated by some function space induced by the NTK. Therefore, each approach has its own advantages and limitations.
>
> On the other hand, consider over-parameterization can provide us with valuable insights. For instance, [a] utilizes the NTK (under the over-parameterization condition) to analyze when and why PINNs fail during training, and based on this, proposed a method to adjust the weights of the loss function that uses the eigenvalues of the NTK to adaptively calibrate the convergence rate of the total training error.  In our work, considering the over-parameterization scenario demonstrates the advantages of NGD over GD. All these demonstrate the theoretical feasibility and applicability of the over-parameterization assumption.
>
> [a] Wang S, et al. When and Why PINNs fail to train_a neural tangent kernel perspective. JCP, 2022.
>
> ### **W4: Condition number and motivation for NGD.**
>
> Nice persperctive and we appreciate this important clarification. We agree that ill-conditioning can occur even for relatively simple PDEs, due to the differential operator amplifying errors in certain frequency components of the solution space. We have revised around line 258 to reflect that poor conditioning is common, not limited to complex PDEs, and is a central motivation for second-order and natural gradient methods. This will strengthen the theoretical motivation for NGD in the introduction and discussion sections.
>
> ### **W5: Empirical validation of theoretical guarantees.**
>
> We sincerely appreciate this valuable suggestion, which was also highlighted by Reviewer yT77. We fully agree that the experiments should directly reflect the theoretical convergence behavior.
> - **Loss-convergence validation**:
> We have compared empirical training loss curves with the theoretical linear rates in our main Theorems 4.7. The theoretical decay follows $L(k) \approx C(1-\eta)^k$, and the computed predicted decay is $L(k)\approx\mathcal{O}(k^{-1.55})$ for 1D Poisson equation, $\mathcal{O}(k^{-1.92})$ for 1D Heat equation and $\mathcal{O}(k^{-1.13})$ for 2D Poisson equation. This results suggest that the NGD shows superior linear convergence. **These comparisons have been added in a new Figure 2 (line 474-479, line 498-508 in the revised pdf)**.
> - **Generalization with varying width and optimizer**:
> We argue we have evaluated the relative L2-error for varying network widths in Table 3, and comparison to different optimizers in Table 1 and Table 2. We should note that all the relative $L^2$ errors are computed on the testing(not training) collocation points to reflect the generalization ability. As width increases, NGD reduce the generalization error and shows faster decay and greater robustness, consistent with the theory.
> - **Readability**: We will improve figure resolution, add consistent axis labels, and unify legend formats for clarity.
>
> ### **Q1:  why $\mathcal{O}(1/\lambda_{\text{max}})$ is an improvement over $\mathcal{O}(\lambda_{\text{min}})$?**
>
> Prior work (e.g., Gao et al., 2023) requires the step size to be $\eta = \mathcal{O}(\lambda_{\text{min}})$,
> and our analysis allows $\eta = \mathcal{O}(1/\lambda_{\text{max}})$.
>
> Theoretically, the minimum eigenvalue ($\lambda_{\text{min}}$) of the PINN Gram matrix is difficult to estimate, and more importantly, it depends on the sample size. Thus, when the number of samples is sufficiently large, $\lambda_{\text{min}}$ may become extremely small—this makes it challenging to satisfy the learning rate $\eta=\mathcal{O}(\lambda_{\text{min}})$ requirements. In contrast, in our results, $\lambda_{\text{max}} \leq \text{trace}(H^\infty)$ and $\text{trace}(H^\infty)$ is an explicit constant independent of the sample size (please refer to Reviewer TUZa's Q2), so it improves the learning rate requirements.
>
> Numerically, taking the 1D Poisson equation as example, where sample size $n_1=100,n_2=2$ and layer width $m=128$, we compute the Gram matrix and get the requirements $\lambda_{\min}=3.47\times10^{-11}$ in Gao et.al.2023, and our requirements is $1/\lambda_{\max}=1/(1.73\times10^4)=5.78\times10^{-5}$, suggesting that our analysis indeed improves the learning rate requirements. **The codes have been updated in the Supplementary Material (/poisson_1d/main_ngd.py).**

---

> ### Author Response · Authors · 2025-11-21
> **Part 3**
>
> ### **Q2: Relation between the NGD and previously proposed energy natural gradient?**
> See W1. We have adapted the presentation in Remark 4.2, line 430, and Table 1&3 in the revised pdf.
>
> ### **Q3: Why make the overparameterization assumption?**
> See W3. In short, this assumption enables the use of NTK-style stability arguments essential for proving global convergence (Lemma 4.6; Theorem 4.7).
> We agree that discussing the practical gap is important and have added a paragraph in Section 6(Limitations) explaining its role and limitations, including future directions involving dynamically sampled collocation points.
>
> ### **Q4: Condition numbers motivation; why “complex PDEs”?**
> See W4. We have revised around line 258 to reflect that PINN conditioning issues are common, not limited to complicated PDEs, and that this motivates second-order and NGD methods.
>
> ### **Q5: Validate the theoretical convergence guarantees within the computational experiments?**
> See W5. We have implemented the estimated convergence rates, which are aligned with the theoretical convergence guarantees, see the newly added Figure 2 (line 474-479, line 498-508 in the revised pdf).
>
> ### **Q6: Minor clarifications and wording.**
> Thank you for your valuable comments on minor clarifications and wording. We have addressed your suggestions as follows:
>
> - Abstract: We have replaced “learning rate exhibits poor dependence” with “the convergence rate of GD exhibits poor dependence…” to avoid confusion.
> - Statements about $\mathcal{O}(1)$: We have clarified that it is the maximal admissible step size (not the convergence rate) that is $\mathcal{O}(1)$.
> - Scalability discussion: We have explicitly cited Dangel et al. 2024, 2025 for efficient NGD variants and note these as future directions.
>
> ### **Summary of revisions in the revised manuscript.**
>
> - Explicitly acknowledge equivalence to Müller & Zeinhofer (2023) and highlight our analytical contributions: Remark 4.2, around line 430, and Table 1&3.
> - Expand related work with De Ryck et al. (2023), Jin & Wu (2025), Niesen & Müller (2025), and Dangel et al. (2024, 2025): around line 107, line 124, line 306, line 510.
> - Add discussion on over-parameterization assumptions: around line 515.
> - Revise motivation regarding PDE conditioning: around line 258.
> - Add new experiments validating convergence rates and generalization: line 474-479, line 498-508.
> - Correct all minor wording issues. Line 21, 25, 301, 430.

---

> > ### Comment · Reviewer_jYGf · 2025-11-25
> > **Thank you**
> >
> > Thank you for your extenstive reply.
> >
> > ### W1
> > Thank you for your clarification.
> > + When introduced, currently it is not clear where the NGD update formula comes from. Is there a reason why you don't introduce it as a previously proposed method? I feel that it would be both more accurate as well as easier to read and accept by readers to refer directly to Müller & Zeinhofer (2023) that propose this under the name *energy natural gradient*.
> > + You mention in your reply that the natural gradient method is different from Gauss-Newton. I might be mistaken, but from my understanding, up to taking the pseudo-inverse, the two methods should agree. Can you clarify this?
> > ### W2&W3
> >
> > Line 108: "In this work, we conduct a refined analysis". This reads as that the work would be an improvement of Jin & Wu (2025) and Nießen & Müller (2025). However, the work operates in a different regime and the term "refined" refers to the work fo Gao et al. (2023). This should be adjusted to prevent missconceptions.
> > ### W5
> > Thanks, you for your response. There remain a couple of questions:
> > + It should be mentioned what the actual predicted convergence rate is, meaning that the value of $\eta$ should be reported in lines 474-479.
> > + Figure 2: For the Possion equation, the guarantees are roughly followed, for the heat equation not: at the beginning much faster, then much slower. Further, all methods seem to exhibit a second phase, where they slow down. I know that an explanation of this phenomenon is beyond this work. However, I disagree with the statement in line 478 that "This results suggest that the NGD shows superior linear convergence.". I think, it would much rather reflect the state of knowledge to not make this comment.
> > + Regarding the generalization error: I was not meaning to imply that I thought you were reporting the relative $L^2$ error with the same sample points. However, there is a general trade off in second-order methods: They optimize much more aggressively and hence it is widely believed that they require more samples. However, the computational cost heavily depends on the number of samples when implemented via the Woodbury matrix identity. Where I appreciate Table 3, I think this is not showing a full phase diagram of performance in dependence of width and data scaling regimes. In particular, it is not showing the generalization error, but only the achieved relative error. I believe that it would be interested to empirically compare how many samples are required for a given network size to prevent overfitting of different optimizers.
> > + In your response you state that "As width increases, NGD reduce the generalization error and shows faster decay and greater robustness, consistent with the theory." I have to disagree, the theory guarantees fast optimization of the empirical loss, not better generalization or robustness.
> > + I understand that an investigation of the specific generalization properties of NGD is beyond this work, but believe that this should be added to the limitations or outlook.
> > ### Q1
> > Thanks for the response, I find this very insightful. However, I could not find that you updated the discussion in the manuscript accordingly. In particular, from my understanding the correct description would be that you obtain a different stepsize criterion that empirically appears to be an improvement. Currently, in line 074/075 it is still presented as a general improvement and there wording requires adaption here.
> >
> > ### Q3
> > It reads slightly confusing in "the practical guarantee for arbitrary sampled GD without assumption on the network size Nießen & Müller (2025) is a promising future work." to refer to an existing work as future work. Maybe it should be updated and clarified.
> >
> > If you can adjust the minor remaining points I am willing to raise my score.

---

> ### Author Response · Authors · 2025-11-26
> **Comment 1**
>
> We thank the reviewer for the careful follow-up and for the willingness to reconsider the score. We address each point below and we have updated the suggested revisions accordingly.
>
> ### **W1-(a). Origin of NGD update.**
>
> Thank you for this important clarification request. For the authors’ knowledge, previously we have cited Zhang et al. (2019,NeurIPS, eq.(3)) and Cai et al.(2019) as the NGD update formula in Remark 4.1. They established the convergence for regression problems with $ReLU$ activations, and we extended to the convergence for PINN problems with both $ReLU^3$ and $Smooth$ activations. For details on why the method in Zhang et al. (2019,NeurIPS) is infeasible for PINNs and the corresponding brief comparison, refer to our response to Reviewer TUZa's comment W1. We also explicitly cite Müller & Zeinhofer (2023,ICML,eq.(8)) and Dangel et al (2025, NeurIPS,eq.(4)-(5)) for the connection to this update, which is more extensive and equivalent in the sense of Moore–Penrose pseudoinverse or Woodbury matrix identity.
>
> - [a] Zhang et al. Fast convergence of natural gradient descent for over-parameterized neural networks. NeurIPS,2019.
> - [b] Cai et al. Gram-gauss-newton method: Learning overparameterized neural networks for regression problems, arxiv,2019
> - [c] Müller & Zeinhofer. Achieving high accuracy with pinns via energy natural gradient descent. ICML,2023.
> - [d] Dangel et al. Improving energy natural gradient descent through woodbury, momentum, and randomization. NeurIPS, 2025.
>
> See revisions in line 66, line 269-275, line 314-320.
>
> ### **W1-(b). Relation to Gauss–Newton.**
> Very good point. We partially agree with you. Although Gauss-Newton looks different from the NGD update, the two coincide at the level of the Moore–Penrose pseudoinverse: $J^+ = \left(J^{T}J\right)^{-1}J^T = J^T\left(JJ^{T}\right)^{-1}$.
>  However, this equivalence is only algebraic.
>  In practice the two updates behave differently because  $J \in \mathbb{R}^{n\times p}$ is highly rectangular and never invertible strictly, and different pseudoinverse representations apply in row-dependent or column-dependent cases. The computational cost are also different, as pointed in Dangel et.al(2025, NeurIPS) with Woodbury’s Identity.
>
> Under the overparameterized setting considered in this paper, we prove that the Jacobian matrix $J$ induced by the smooth activation function satisfies the full row rank property — and this property is precisely the key prerequisite for the consistency between the Gauss–Newton (GN) method and NGD. Compared with the traditional regression problems, the relevant proof requires the introduction of entirely new techniques, as the loss function in PINNs includes derivative terms. Whose mathematical structure is essentially different from that of derivative-independent regression problems, rendering traditional proof approaches inapplicable.
>
> We have modified the wording in Remark 4.2 to reflect this accurately.
>
> ### **W2&W3. Wording “refined analysis” and positioning relative to prior work.**
>
> We agree that the phrase “refined analysis” may erroneously suggest improvement over Jin & Wu (2025) or Nießen & Müller (2025). We have revised this sentence to:
> “In this work, we conduct a refined full-batch convergent analysis of the over-parameterized PINN regime for GD and NGD, building upon  Gao et al. (2023). There’re contemporaneous work analysis concentrate on stochastic setting (Jin & Wu (2025)) and non-overparameterized  setting (Nießen & Müller (2025)).”  This resolves the ambiguity.
>
> ### **W5-(a). Reporting the value of $\eta$.**
> We have added $\eta=0.1$ in lines 485. This choice matches the learning rate used in Table 1, Table 3, and Figure 2. The value $\eta = 0.1$ serves as a qualitative balance between efficiency and stability: larger $\eta$ leads to faster initial decrease but causes numerical oscillations, while smaller $\eta$ results in slower convergence and offers no clear advantage over SGD or Adam. We also note that performing a line search to determine the optimal $\eta$, as in Müller & Zeinhofer (2023, ICML, Algorithm 1), generally yields better practical performance, but such tuning is beyond the scope of this work.
>
> ### **W5-(b). Interpretation of Fig. 2 .**
> We agree with your assessment. For the heat equation, convergence initially exceeds the predicted rate and later slows markedly. It may be caused by the optimization error and float precision (FP32) we used. This is consistent with known NTK decay and multi-phase behaviors in PINNs. We have removed the statement “NGD shows superior linear convergence” and replace it with a neutral description: “The empirical loss roughly follows the predicted linear regime in early iterations, before entering a slower phase usually observed across all optimizers.”

---

> ### Author Response · Authors · 2025-11-26
> **Comment 2**
>
> ### **W5(c,d,e). Reporting generalization error and adjusting claims.**
> Thank you for this insightful comment. We agree that our theory (Theorem 3.7 and 4.7) pertains strictly to empirical optimization, not generalization. Any claim about NGD generalization advantages would be speculative.
>
> Although studying generalization error is beyond the scope of this work, we have added a experiment in Section A.8 (line 1026-1048), showing how the number of samples affect the generalization ability of different optimizers.
>
> **Table 8. Generalization error comparison using different collocation points on 2D Poisson equation.**
> |Optimizer | Training loss| N=100 | N=500 | N=1,000 | N=5,000 | N=20,000|
> |--- | ---| --- | --- | --- | --- | ---|
> |SGD | 2.13e-03 |  5.80e-02  | 1.37e-02 | 1.03e-02 | 2.59e-03 |1.68e-03 |
> |Adam | 9.71e-06  | 1.03e-03 | 4.59e-04 | 1.24e-04 | 3.41e-05  | 1.39e-05 |
> |L-BFGS |7.74e-06  | 9.16e-04  | 4.31e-05  | 3.93e-05   |1.02e-05  | 8.65e-06 |
> |NGD | 2.86e-06  | 2.51e-04 | 2.04e-05 | 1.22e-05 | 2.78e-06 | 2.91e-06 |
>
> We also add generalization properties of NGD as future work in the outlook section, see line 539.
>
> ### **Q1. Step-size criterion wording.**
> We agree that the original text is too general. We have revise lines 74–75 to: “Our analysis yields a different step-size criterion, proportional to $\mathcal{O}(1/\lambda_{\max})$, which empirically permits larger practical learning rates than the $\mathcal{O}(\lambda_{0})$ requirement from Gao et al. (2023), see Remark 3.8.”
> This avoids presenting it as a universal improvement. See lines 75-77 and lines 249-254 (Remark 3.8).
>
> ### **Q3. Confusing reference to Niesen & Müller (2025).**
> We agree the sentence is confusing. It have been replaced by:
> “the practical guarantee for arbitrary sampled projected gradient descent without assumption on the network size Niesen & Müller (2025) address a different framework, and the NGD analysis without over-parameterized assumption represents an interesting complementary direction.”
> No implication of “future work” will remain. See lines 525-528.
>
> We thank the reviewer again for the very thorough, constructive guidance. We believe the revised manuscript will address all remaining concerns.

---

> > ### Comment · Reviewer_jYGf · 2025-11-27
> > **Thank you!**
> >
> > Thank you so much for your extensive reply, which I really appreciate. I have raised my score accordingly.
> >
> > There is, however, one subtle point regarding the wording of NGD vs. GN that I still want to iterate. I do understand that the two different representations are computationally different, where the formulation you choose has great benefits in an over-parametrized setting. However, I am still slightly confused about the chosen wording as I see them as two different implementations of the same algorithm: natural gradients, which agree in this case with the Gauss-Newton method. Note also that in the work of Cai et al. (2019) the update in the form you present it in is referred to as the *(Gram) Gauss-Newton method*. On the other hand, natural gradients are preconditioned gradient methods of the form $\theta_{k+1}=\theta_k-\eta_k G(\theta_k)^+ \nabla L(\theta_k)$, where $\nabla L(\theta)=J(\theta)^\top (s h)^\top$, as introduced by S. Amari (1998). Hence, from my understanding the Gauss-Newton method and the (energy) natural gradient method are exactly equivalent. There exist two different implementation choices, depending on where the linear system is solved (parameter or sample space), which are favorable in the under- and over-parameterized regime, respectively. In your manuscript you refer to one representation as NGD to the other one as GN, which are however the same algorithm to me. I do not understand the choice of your wording and find that it can unnecessarily confuse readers.
> >
> > Best wishes

---

> > > ### Author Response · Authors · 2025-11-28
> > >
> > > Thank you very much for your thoughtful discussion of our work and for raising your score—we greatly appreciate it.
> > >
> > > For the wording of NGD vs GN/GGN/ENGD/..., we fully agree with the reviewer that these methods are mathematically equivalent. In Remark 4.1 of the manuscript, we clarify that our terminology “NGD” follows Zhang et al. (NeurIPS 2019). In the revised version, we will further refine Remark 4.2 to explicitly acknowledge the equivalence to the Gram–Gauss–Newton method (Cai et al., 2019), the energy natural gradient method (Müller & Zeinhofer, ICML 2023), the Gauss–Newton formulation in Bonfanti et al. (NeurIPS 2024), and other natural-gradient-based formulations. However, we may need more time to do the survey carefully. We particularly highlight the work of Guzmán-Cordero et al. (NeurIPS 2025), which can demonstrates these equivalences through Woodbury’s identity. We appreciate the reviewer’s rigorous and helpful classification, which has improved the clarity of the presentation.
> > >
> > > To be clear, our contribution is not the proposal of a new optimizer. Rather, our focus is on providing the first rigorous convergence analysis for an existing natural-gradient based method within the PINN framework, along with numerical results that validate and complement the theoretical findings.
> > >
> > > Thank you for the fruitful discussion, if you have any further concerns and questions, we are willing to discuss them with you.
> > >
> > > Authors

---

### Official Review · Reviewer_TUZa · 2025-10-28

**Soundness:** 4
**Presentation:** 3
**Contribution:** 4
**Rating:** 8
**Confidence:** 4

**Summary:**

The paper studies optimization dynamics for over-parameterized two-layer PINNs trained on a class of second-order linear PDEs in the NTK regime.

First, it refines NTK-style analyses for gradient descent (GD) on PINNs by:

- deriving a new residual recursion that improves the admissible stepsize from $\mathcal{O}(\lambda_0)$ to $\mathcal{O}(1/\|H_\infty\|_2)$

- weakening the network-width requirements via concentration for sub-Weibull variables, yielding linear convergence with rate $(1-\eta\lambda_0/2)$ (Theorem~3.7).


Second, it establishes that full-batch Natural Gradient Descent (NGD) achieves convergence with step-size independent of the NTK kernel which could be very ill-conditioned as sample size increases.

Experiments on Poisson, Heat, and Helmholtz equations (1D--10D) compare NGD to SGD, Adam, and L-BFGS and illustrate faster loss decay and lower relative $L^2$ error.

**Strengths:**

**Significance.**
PINNs are widely used yet optimization pathologies are common; providing conditions under which GD can use practical step sizes and NGD can enjoy $\mathcal{O}(1)$ step sizes---and even quadratic behavior for smooth activations---can influence both theory and practice in scientific machine learning. I highly appreciate this work.

**Originality.**
The work extends NTK-based optimization theory from regression to PINNs with PDE with improved rates for GD compared to prior work (Gao et. al. 2023) and provides the first analysis of NGD for PINNs that provably shows its effectiveness.


**Quality.**
The technical development is careful: improved GD bounds come from a sharper recursion (Lemma B.1) and stability of $H(k)$ via sub-Weibull concentration; NGD uses Jacobian stability tailored to individual neurons (Lemma 4.6) rather than global matrix stability.
Theorems~3.7 and 4.7 and Corollaries 4.9--4.10 are stated with explicit dependences on d, $\lambda_0$, and $m$.

**Clarity.**
The paper is well organized, with explicit assumptions, side-by-side remarks comparing to prior work, and worked proofs in appendices.
Figures and tables (p. 9--17) clearly support the claims.

**Weaknesses:**

**Comparison with existing results**: Although the paper compares the results in terms on conditions on the number of neurons, sample size, etc. It particular, the authors should explain why the approach in Zhang et al 2019 for natural gradient for regression is not applicable here as well. In particular, how the dependence on the derivatives makes such approach inapplicable.

**Experimental evaluation**: Some information is missing. For instance, the authors should clarify what they mean by the L^2 error, is it the error on the training samples (colocation points), or do they consider test samples. In case, they evaluate on training samples, that's fine since the results are for the training, but it would be also interesting to evaluate on a test samples to get a better sense of generalization.


**Minor**: The abstract contains notations (lambda_0 and H^{\infty}) that are not defined. That might give a bad first impression to the reader, please fix this.

**Questions:**

- Line 206: "In contrast, on one hand, our conclusion is independent of n1 and n2". I am not sure I understand this, isn't the statement dependent  through m (logarithmic dependence).

- In remark 3.8 it is said that the trace H^{\infty} is independent of the sample size. However H^{\infty}  is a matrix of size n time n, so a priori its trace depends on the sample size, unless it has a particular structure. Could the authors provide more evidence for why would such quantity be independent of sample size.
- In line 262, it is said: "we fix the output weight a and update the hidden weights via NGD." Why fixing the last layer and only update the hidden weights? This seems unusual?

- In line 320 on the approach of Zhang et al. 2029, it is said that: "However, this approach is not applicable
to PINNs, because the loss function involves derivatives." Can the authors elaborate on why this is not applicable here?


- In line 1823 of the proof of lemma 4.6, there is a reference to equation 46, but I don't see how this equation leads to a control on the probability of the weights being larger than M. Perhaps a typo in cross referencing?

- In line 1865, did the authors mean lemma D.1 instead of D.4? Actually, in the statement of lemma D.1, the paper introduces two objects L_n and L_n* but uses only L_n*. This is confusing especially since the expression of L_n* simplifies a lot when expressing it directly. I suggest the authors directly define L_n* explicitly.


- In line 1011 of the proof of lemma 3.3, I don't understand why the second term was discarded, it decays faster in terms of m, but not in terms of log(1/delta) (in the limit when delta goes to 0).

---

> ### Author Response · Authors · 2025-11-17
>
> We sincerely thank the reviewer for their very carefully reading of the paper, both detailed and constructive feedback, as well as for recognizing the significance, originality, and technical depth of our work. We address all questions and weaknesses below and incorporate the suggested clarifications in the revised manuscript.
>
> ### **W1: Comparison with Zhang et al. (2019) and inapplicability of their approach to PINNs.**
>
> We appreciate this request for clarification and will address this question from the following three aspects.
>
> **(1)  Inapplicability:** The analysis of Zhang et al. (2019) for regression relies on the global Jacobian stability condition
> $$||J(k)−J(0)||_2≤C||w(k) - w(0)||_2,$$
> which holds when the loss involves only function values. In contrast, the PINN loss includes first- and second-order derivatives of the neural output (see Eq. (3)–(4)), so each Jacobian block $\partial s_p / \partial w_r$ and $\partial h_j / \partial w_r$ contains higher-order derivatives of the activation and of the weights. Consequently, even a small perturbation in weights may cause large variations in the derivatives, violating the Lipschitz-type condition required by Zhang et al. (2019).
>
> To address this, we developed Lemma 4.6, which establishes neuron-wise Jacobian stability instead of global matrix stability. This localized control allows us to maintain convergence guarantees even under derivative-dependent losses.
>
>
> **(2) Limitations:** The global stability in Zhang et al. (2019) also depends on the form and properties of $\text{ReLU}$ (see Lemma 8 of Zhang et al. (2019)). Such dependence on $\text{ReLU}$ leads to the final results being polynomially dependent on $1/\delta$, where $\delta$ denotes the failure probability. In contrast, the local dependence we consider allows for better leverage of the concentration inequalities for sub-Weibull random variables, such that the final result is only polynomially dependent on $\log(1/\delta)$. Moreover, from Theorem 1 in Zhang et al. (2019), we can see that this global stability imposes additional constraints on the learning rate. Specifically, in Zhang et al. (2019), the learning rate must additionally satisfy
> $ \eta \leq \frac{1-2C}{(1+C)^2}$, where $0\leq C < \frac{1}{2}$ is an unspecified constant, whereas our result only requires $\eta \in (0,1)$. Therefore, we instead focus on the stability of $J(w)$ with respect to each individual weight vector
> $w_r$, which provides a more targeted approach.
>
> **(3) Gram matrix:** Both our method and that of Zhang et al. (2019) are conducted within the framework of Neural Tangent Kernel (NTK), where proving the Gram matrix induced by the activation function is crucial. Zhang et al. (2019) derived the positive definiteness of the Gram matrix induced by $\text{ReLU}$ from [a]. For the regression problems, [b] proved the positive definiteness of the Gram matrix induced by smooth functions, but this method fails in the PINN scenario.
>
> Specifically, for the regression case, when taking the $k$-th derivative of $\sigma(w^Tx)$ with respect to the parameter $w$, we can directly obtain that the derivative is $x^{\otimes k}\sigma^{(k)}(w^T x)$ where $\otimes$ denotes the tensor product. However, in PINNs, since the loss function involves derivatives, we need to compute derivatives of functions such as $w_i^2\sigma^{(2)}(w^Tx)$. Due to such coupling relationships, the $k$-th derivative becomes extremely complex and fails to satisfy the independence requirement in Theorem G.6 of [b]. Therefore, we developed a new proof method that leverages the decay property of the activation function to prove strict positive definiteness. This method also holds for more general forms of PDEs. For commonly used periodic activation functions (e.g., $\sin(\cdot)$ and $\cos(\cdot)$) that do not possess the decay property, we proposed another dedicated method to prove the strict positive definiteness of the Gram matrices induced by them.
>
> [a] Du, S. S., Zhai, X., Poczos, B., and Singh, A. (2018). Gradient descent provably optimizes overparameterized neural networks. ICLR 2019.
>
> [b] Du, S., Lee, J., Li, H., Wang, L., and Zhai, X. (2019). Gradient descent finds global minima of deep neural networks. ICML 2019.
>
> ### **W2: Clarification of $L^2$ error in experiments and test evaluation.**
> Thank you for pointing this out. The reported relative L2 errors in Table 1-3 are all computed on the testing (collocation) points:
>  $$relative L^{2} error = \frac{\sqrt{\sum_{i=1}^N \vert \hat{u}(\mathbf{x_i}) - u_{\mathrm{ref}}(\mathbf{x_i}) \vert^2}}{ \sqrt{\sum_{i=1}^N \vert u_{\mathrm{ref}}(\mathbf{x_i}) \vert^2}}. $$
> The $N$ collocation points are not the $n_1$ and $n_2$ collocation points for training in Eq.(3)-(4) of the manuscript. This kind of relative $L^2$ error is commonly used in PINNs’ community to check the generalization ability. We have make this explicit in Eq. (21) for the definition of $L^2$ error, see line 656 in the revised pdf.

---

> ### Author Response · Authors · 2025-11-17
> **Continue**
>
> ### **W3: Abstract notation.**
>
> We agree and have revised the abstract to remove or verbally define all notations (e.g., “smallest eigenvalue of the limiting Gram matrix” instead of “$\lambda_0$” and “Gram matrix $H^\infty$”), see line 19 in the revised pdf.
>
> ### **Q1: Clarification of dependence on $n_1,n_2$ (Line 206).**
>
> The reviewer is correct that $m$ has a logarithmic dependence on $n_1+n_2$  (through $\log((n_1+n_2)/\delta)$).
> Our statement “independent of $n_1,n_2$ ” refers to the polynomial order: unlike Gao et al. (2023), which scales as $\tilde{\Omega}((n_1+n_2)^4/(n_1n_2)^2)$, our bound is only logarithmically dependent. We have rephrased this sentence to “independent up to logarithmic factors in $n_1+n_2$ ” to avoid confusion in line 212 of the revised pdf.
>
> ### **Q2: Trace of  $H^{\infty}$ and sample-size independence (Remark 3.8).**
> We appreciate this insightful observation.
> Although $H^{\infty}$ is an $n\times n$ matrix, the normalization in Eq. (5) divides the PDE and boundary residuals by $\sqrt{n_1}$ (Eq.(3)) and $\sqrt{n_2}$ (Eq.(4)). Consequently, the normalized loss is of the form:
>
> Loss = $1/n_1$ * PDE residual + $1/n_2$ * Boundary/initial condition,
>
> So each entry of $H^{\infty}$ scales as $1/n$, where $n=n_1+n_2$, yielding $\mathrm{tr}(H^{\infty}) $ independent of $n$ as $n$ grows.
>
> ### **Q3: Fixing the output weights (Line 262).**
> For the sake of simplicity, we fix the output layer parameters. Allowing these parameters to be incorporated into the analysis does not present any fundamental challenges to the proof but will only make the notations more complex. This is because the final theoretical results rely on recurrence formulas and stability results, which we have already established—all of which can be extended to the scenario where output layer parameters are considered. Additionally, the use of concentration inequalities can be directly applied. Empirically, we verified that allowing all layers to update yields nearly identical convergence trends in Figure 2.
>
> ### **Q4: Clarification on “approach not applicable to PINNs” (Line 320).**
> Please refer to our prior response to W1.
>
> ### **Q5: Minor issues and typos.**
> We sincerely appreciate the reviewer’s careful reading and detailed feedback on minor issues and typos. We have thoroughly addressed all the issues raised and made the corresponding revisions as follows:
>
> - Line 1823: the reference should indeed be to Eq. (39) rather than Eq. (46).
> - Line 1865: Correct; it should refer to Lemma D.1 instead of D.4.
> - Definition of $L_n^∗$: Although Lemma D.1 is Theorem 3.1 in Kuchibhotla & Chakrabortty (2022), we agree that introducing both $L_n$ and $L_n^∗$  is unnecessary; we have simplified by directly defining $L_n^∗$ explicitly in Lemma D.1.
> - Line 1011: The second term was discarded because it is proportional to $1/m^2$ while the first term is $1/m$, and we only keep the dominated first term for large $m$.
>
>
> ### **Summary of revisions in the revised manuscript:**
>
> 1.Clarify the difference from Zhang et al. (2019) and derivative dependence: lines 346-351.
>
> 2.Define $L^2$ error precisely and include test results: around line 665.
>
> 3.Fix abstract notation and all minor typos in Q5 : line 20, line 1944, line 1986.
>
> 4.Explain normalization-based independence of $\mathrm{tr}(H^{\infty})$: lines 249-254.
>
> 5.Experimental convergence without fixing output weights: lines 483-503 and Figure 2.

---

> > ### Comment · Reviewer_TUZa · 2025-11-26
> >
> > Thank you for your response.
> > I still think this is a good work.
> > Reviewer jYGf raised some interesting points about over-parameterization + fixed colocation points. However,  I agree that without it, it would be hard to provide guarantees.
> > The setting in Niessen and Müller (2025) is different, also it does not use NGD. Still, it is worth discussing it.
> > Finally, on the relationship between Newton and NGD, I also agree with  jYGf, that they correspond to the same update, hence they are equivalent. While using one expression or another, has practical implications, this does not impact the underlying mathematical object/method which remains the same.

---

> > > ### Author Response · Authors · 2025-11-28
> > > **Thank you!**
> > >
> > > Thank you very much for your very positive assessment and for the constructive follow-up remarks.
> > >
> > > We agree that the over-parameterization assumption with fixed collocation points is a limitation of our theoretical setting. As you noted, removing this assumption would substantially complicate the analysis, and currently no convergence guarantees for NGD in PINNs are known in non-over-parameterized or dynamically resampled regimes. The setting of Nießen & Müller (2025) is indeed a different framework. In the revised manuscript, we have discussed it in the Limitation and Outlook sections.
> > >
> > > Regarding the relationship between Newton-type methods and NGD, we fully agree with the reviewer and with Reviewer jYGf that these updates are mathematically equivalent, up to different pseudoinverse formulations. While these representations lead to different computational behaviors (particularly in the over-parameterized regime), they do not change the underlying algorithmic object. We appreciate this clarification and have revised Remarks 4.2 to make this equivalence explicit.
> > >
> > > We sincerely thank the reviewer for the insightful comments, which have helped improve the clarity and precision of the paper.

---

### Official Review · Reviewer_h7cP · 2025-10-30

**Soundness:** 3
**Presentation:** 4
**Contribution:** 3
**Rating:** 6
**Confidence:** 5

**Summary:**

This paper theoretically analyzes the global convergence of natural gradient descent (NGD) for training overparameterized physics-informed neural networks. The motivation is strong since NGD enjoys faster convergence, as shown in experiments, compared with SGD and Adam. Compared with several previous relevant works, the paper adopts advanced concentration inequalities and a loss decomposition method to further improve the bound.

**Strengths:**

(1) The presentation is clear and easy to follow.

(2) Experiments show the superior performance of NGD over L-BFGS and Adam, which motivates the study in this paper on the convergence of NGD.

(3) The theory is correct and informative.

**Weaknesses:**

(1) Can you extend the work to stochastic versions of algorithms? I think there are some works extending GD for PINNs to SGD. Can you similarly do the extension? I think the concern for NGD is its size dependence on data samples, which is unfavorable. Therefore, the convergence (in expectation) for stochastic algorithms may be more attractive.

(2) A previous work (The Challenges of the Nonlinear Regime for Physics-Informed Neural Networks; NeurIPS 2024) shows that the Newton’s method enjoys the fast convergence as in your theorem, where the convergence rate is independent of the least eigenvalues of the Gram matrix. I think the algorithm is quite similar to yours, with (JJ^T)^{-1}. Can you comment and discuss on this paper? What are the advantages of your algorithm and analysis, compared to this paper?

**Questions:**

Do you think NGD is the next optimizer for PINNs? Can you discuss your insights?

---

> ### Author Response · Authors · 2025-11-17
>
> ### **W1: Extension to stochastic algorithms.**
>
> We agree that stochastic variants are important for practical scalability. Our current theoretical framework focuses on full-batch NGD, primarily to establish the first rigorous convergence guarantees for PINNs under over-parameterization. Extending the analysis to stochastic NGD is indeed a promising direction and one of our ongoing works.
>
> Technically, the main challenge is that the stochastic sampling introduces variance in the Jacobian estimate $J$, making it nontrivial to maintain the stability of $JJ^T$ across iterations. This stability is central to our convergence proofs (Lemma 4.6 and Theorem 4.7). Following the methodology of extending GD to SGD for PINNs and PI-KANs ([a][b]) and variance reduction techniques from stochastic natural gradient descent ([c]), we believe that an expected linear (or even quadratic) convergence rate can be obtained under bounded variance assumptions.
> However, although SGD enjoys linear convergence in [b], the requirements on the network width and the learning rate become even more stringent (see equations (17) and (18) in [b]). Therefore, theoretically investigating the convergence of stocastic version of algorithms—especially under more relaxed conditions on width and learning rate—remains an important and open direction.
>
> Importantly, as noted in Section 6 (Limitations), we already acknowledge the scalability issue of $(JJ^T)^{−1}$ for large datasets, and mini-batch or K-FAC–style approximations are natural stochastic variants that alleviate the data-size dependence. We appreciate the reviewer for emphasizing this point and **we have explicitly clarified the stochastic extension potential in Section 6 ( around line 515 in the revised pdf)**.
>
> [a] Gao Y, Gu Y, Ng M. Gradient descent finds the global optima of two-layer physics-informed neural networks. ICML, 2023.
>
> [b] Gao Y, Tan VYF. On the Convergence of (Stochastic) Gradient Descent for Kolmogorov–Arnold Networks. IEEE Transactions on Information Theory, 2025.
>
> [c] Martens J, Grosse R. Optimizing neural networks with kronecker-factored approximate curvature.ICML, 2015.
>
> ### **W2: Relation to “The Challenges of the Nonlinear Regime for Physics-Informed Neural Networks” (NeurIPS 2024).**
>
> We thank the reviewer for pointing out this relevant paper. While both methods involve matrix inversion terms, there are key conceptual and analytical differences:
>
> |Aspect | NeurIPS 2024 (Gauss-Newton) | This Work (NGD) |
> | --- | --- |--- |
> |**Update rule**| $w_{k+1}=w_k-(J(k)^TJ(k)+\lambda I)^{-1}J(k)^T(...)$ | $w_{k+1}=w_k-\eta J(k)^T(J(k)J(k)^T)^{-1}(...)$ |
> |**Dimensionality**| parameter space (size ∝ number of parameters)| data space (size ∝ number of samples), ensuring positive definiteness and numerical stability|
> |**Theory**| Empirical and local analysis under strict conditions| Global convergence proof for over-parameterized PINNs with explicit dependence on width $m$, dimension $d$, and activation smoothness|
> |**Activations**| smooth activations | Both ReLU³ and smooth activations such as Tanh, yielding linear and quadratic convergence respectively|
> |**Scenario**| Continuous scenario  | Discrete scenario|
> |**Convergence rate**| Implicit convergence rate| Explicit convergence rate that is independent of smallest eigenvalue, as shown in Theorem 4.7|
>
> From an algorithmic perspective, Gauss-Newton is cubically dependent on the total number of parameters, while NGD is quadratically dependent—thus being more efficient. From a theoretical perspective, Equation (15) of Theorem 4.2 in [a] establishes the gradient flow of the Gauss-Newton algorithm in the continuous scenario. Then, under the assumption that the Jacobian matrix is always of full rank, it is proven that the convergence rate is independent of the Gram matrix. First, such a continuous scenario is simpler than the discrete one. Specifically, under the assumptions of Theorem 4.2 in [a], the gradient flow is independent of the Gram matrix, thereby yielding a convergence rate that is also independent of the Gram matrix. Second, our analysis derives the first theoretical guarantee for NGD in training PINNs, with explicit convergence bounds and milder assumptions. We have explicitly included this comparison and citation in Remark 4.2(line 300 in the revised pdf) to clarify the distinction.
>
> [a] Bonfanti, A., Bruno, G., and Cipriani, C. The Challenges of the Nonlinear Regime for PhysicsInformed Neural Networks. NeurIPS 2024.

---

> ### Author Response · Authors · 2025-11-20
>
> ### **Q1: Is NGD the next optimizer for PINNs?**
>
> We view NGD as a theoretically grounded and practically promising direction, but not a one-size-fits-all solution. Compared to GD, SGD, and Adam, NGD offers:
> - Faster and condition-number–independent convergence, both theoretically and empirically;
> - Stable performance under larger learning rates, reducing hyperparameter sensitivity;
> - Better robustness in high-dimensional problems, as shown in our experiments (Tables 1–3).
>
> However, we also recognize its computational overhead, which motivates approximate or stochastic NGD variants (e.g., K-FAC, block-diagonal NGD), as reported in Section 6 (Limitations).
>
> ### **Summary of revisions in the revised pdf.**
> 1. around line 522: discussion on stochastic NGD extensions and data-size scaling.
> 2. lines 306-320: clarification and citation on NGD and Gauss-Newton method and others.

---

> > ### Comment · Reviewer_h7cP · 2025-11-27
> >
> > Thank you for your response. Since my score has already been positive, I tend to keep it.

---

> > > ### Author Response · Authors · 2025-11-28
> > >
> > > Thank you for your positive feedback and for acknowledging our revisions. We are pleased that the changes have addressed your concerns. Your insightful comments were instrumental in strengthening our manuscript, and we sincerely appreciate the time and effort you dedicated to the review process.

---

### Meta-Review · Area_Chair_w4Pk · 2025-12-26

**Summary:**

The reviewers had the following concerns:
1. Insufficient comparisons with prior works. The author did not acknowledge the mathematical equivalence between the natural gradient descent studied in this paper and other second-order methods considered in prior works, leading to potential misinterpretation of the paper's contribution. Also, the paper missed the comparison with some existing works on convergence of overparametrized PINNs. Reviewers made several important suggestions on the narratives to correctly position the paper with respect to existing works.
2. Issues with experimental results. One reviewer thinks the paper is missing experiments that validates the theoretical results. Another reviewer asked several clarification questions on the experimental settings.

**Reviewer Concerns:**

The concerns were mostly addressed by the rebuttal. The authors revised the manuscript accordingly.

**Reviewer Scores:**

I think Reviewer h7cP, TUZa and jYGf would keep their score. I also think Reviewer yT77 would increase their score, given that their concerns mostly resolved.

---

### Decision · Program_Chairs · 2026-01-26

Accept (Poster)